# Quantifying Point-Prediction Uncertainty in Neural Networks via Residual Estimation with an I/O Kernel

**Xin Qiu**
Cognizant
qiuxin.nju@gmail.com

**Elliot Meyerson**
Cognizant
elliot.meyerson@cognizant.com

**Risto Miikkulainen**
Cognizant
The University of Texas at Austin
risto@cognizant.com

## Abstract

Neural Networks (NNs) have been extensively used for a wide spectrum of real-world regression tasks, where the goal is to predict a numerical outcome such as revenue, effectiveness, or a quantitative result. In many such tasks, the point prediction is not enough: the uncertainty (i.e. risk or confidence) of that prediction must also be estimated. Standard NNs, which are most often used in such tasks, do not provide uncertainty information. Existing approaches address this issue by combining Bayesian models with NNs, but these models are hard to implement, more expensive to train, and usually do not predict as accurately as standard NNs. In this paper, a new framework (RIO) is developed that makes it possible to estimate uncertainty in any pretrained standard NN. The behavior of the NN is captured by modeling its prediction residuals with a Gaussian Process, whose kernel includes both the NN's input and its output. The framework is evaluated in twelve real-world datasets, where it is found to (1) provide reliable estimates of uncertainty, (2) reduce the error of the point predictions, and (3) scale well to large datasets. Given that RIO can be applied to any standard NN without modifications to model architecture or training pipeline, it provides an important ingredient for building real-world NN applications.

## 1 Introduction

Nowadays, Neural Networks (NNs) are arguably the most popular machine learning tool among Artificial Intelligence (AI) community. Researchers and practitioners have applied NNs to a wide variety of fields, including manufacturing (Bergmann et al., 2014), bioinformatics (LeCun et al., 2015), physics (Baldi et al., 2014), finance (Niaki & Hoseinzade, 2013), chemistry (Anjos et al., 2015), healthcare (Shahid et al., 2019), etc. Although standard NNs are good at making a point prediction (a single outcome) for supervised learning tasks, they are unable to provide uncertainty information about predictions. For real-world decision makers, representing prediction uncertainty is of crucial importance (Krzywinski & Altman, 2013; Ghahramani, 2015). For example, in the case of regression, providing a $95\%$ confidence interval around the prediction allows the decision maker to anticipate the possible outcomes with explicit probability. In contrast, simply returning a single point prediction imposes increased risks on decision making, e.g., a predictively good but actually risky medical treatment may be overconfidently interpreted without uncertainty information.

Conventional Bayesian models such as Gaussian Processes (GP) (Rasmussen & Williams, 2006) offer a mathematically grounded approach to reason about the predictive uncertainty, but often come with a prohibitive computational cost and lower prediction performance compared to NNs (Gal & Ghahramani, 2016). As a potential solution, considerable research has been devoted to the combination of Bayesian models and NNs (see Section 2 for a detailed review of such approaches),

aiming to overcome the downsides of each. However, from the classical Bayesian Neural Network (Neal, 1996), in which a distribution of weights is learned, to the recent Neural Processes (Garnelo et al., 2018a;b; Kim et al., 2019), in which a distribution over functions is defined, all such methods require significant modifications to the model infrastructure and training pipeline. Compared to standard (non-Bayesian) NNs, these new models are often computationally slower to train and harder to implement (Gal & Ghahramani, 2016; Lakshminarayanan et al., 2017), creating tremendous difficulty for practical uses. Gal & Ghahramani (2016) derived a theoretical tool to extract uncertainty information from dropout training, however, the method can only be applied to dropout models, and requires changes to their internal inference pipeline. Quantifying point-prediction uncertainty in standard NNs, which are overwhelmingly popular in practical applications, still remains a challenging problem with significant potential impact.

To circumvent the above issues, this paper presents a new framework that can quantitatively estimate the point-prediction uncertainty of standard NNs without any modifications to the model structure or training pipeline. The proposed approach works as a supporting tool that can augment any pretrained NN without retraining them. The idea is to capture the behavior of the NN by estimating its prediction residuals using a modified GP, which uses a new composite (I/O) kernel that makes use of both inputs and outputs of the NNs. The framework is referred to as RIO (for Residual estimation with an I/O kernel). In addition to providing valuable uncertainty estimation, RIO has an interesting side-effect: It provides a way to reduce the error of the NN predictions. Moreover, with the help of recent sparse GP models, RIO scales well to large datasets. Since classification problems can be treated as regression on class labels (Lee et al., 2018), this paper focuses on regression tasks. In this setting, empirical studies are conducted with twelve real-world datasets, and an initial theoretical investigation is presented that characterizes the benefits of RIO. The results show that RIO exhibits reliable uncertainty estimations, more accurate point predictions, and better scalability compared to alternative approaches.

## 2 RELATED WORK

There has been significant interest in combining NNs with probabilistic Bayesian models. An early approach was Bayesian Neural Networks, in which a prior distribution is defined on the weights and biases of a NN, and a posterior distribution is then inferred from the training data (MacKay, 1992; Neal, 1996). Notice that unlike these methods, RIO is concerned only with uncertainty over NN predictions, not over NN weights. Traditional variational inference techniques have been applied to the learning procedure of Bayesian NN, but with limited success (Hinton & van Camp, 1993; Barber & Bishop, 1998; Graves, 2011). By using a more advanced variational inference method, new approximations for Bayesian NNs were achieved that provided similar prediction performance as dropout NNs (Blundell et al., 2015). However, the main drawbacks of Bayesian NNs remain: prohibitive computational cost and difficult implementation procedure compared to standard NNs.

Alternatives to Bayesian NNs have been developed recently. One such approach introduces a training pipeline that incorporates ensembles of NNs and adversarial training (Lakshminarayanan et al., 2017). Another approach, NNGP, considers a theoretical connection between NNs and GP to develop a model approximating the Bayesian inference process of wide deep neural networks (Lee et al., 2018). Deep kernel learning (DKL) combines NNs with GP by using a deep NN embedding as the input to the GP kernel (Wilson et al., 2016). In Iwata & Ghahramani (2017), NNs are used for the mean functions of GPs, and parameters of both NNs and GP kernels are simultaneously estimated by stochastic gradient descent methods. Conditional Neural Processes (CNPs) combine the benefits of NNs and GP, by defining conditional distributions over functions given data, and parameterizing this dependence with a NN (Garnelo et al., 2018a). Neural Processes (NPs) generalize deterministic CNPs by incorporating a latent variable, strengthening the connection to approximate Bayesian and latent variable approaches (Garnelo et al., 2018b). Attentive Neural Processes (ANPs) further extends NPs by incorporating attention to overcome underfitting issues (Kim et al., 2019). The above models all require significant modifications to the original NN model and training pipeline. Compared to standard NNs, they are also less computationally efficient and more difficult for practitioners to implement. In the approach that shares the most motivation with RIO, Monte Carlo dropout was used to estimate the predictive uncertainty of dropout NNs (Gal & Ghahramani, 2016). However, this method is restricted to dropout NNs, and also requires modifications to the NN inference process. Different from all above-mentioned methods, which are independent alternatives to standard NNs,

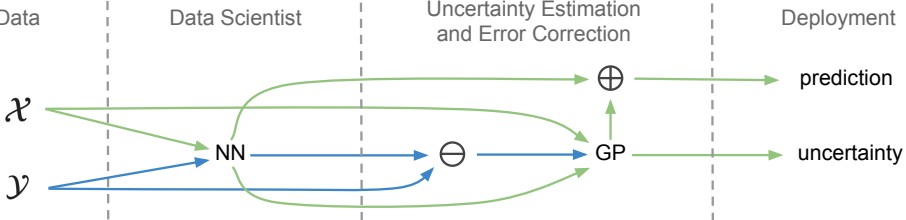

Figure 1: **Complete model-building process.** Given a dataset, first a standard NN model is constructed and trained by a data scientist. The RIO method takes this pretrained model and trains a GP to estimate the residuals of the NN using both the output of the NN and the original input. Blue pathways are only active during the training phase. In the deployment phase, the GP provides uncertainty estimates for the predictions, while calibrating them, i.e., making point predictions more accurate. Overall, RIO transforms the standard NN regression model into a more practical probabilistic estimator.

RIO is designed as a supporting tool that can be applied on top of any pretrained NN. RIO augments standard NNs without retraining or modifying any component of them.

## 3 THE RIO FRAMEWORK

This section gives the general problem statement, develops the RIO framework, and discusses its scalability. For background introductions of NNs, GP, and its more efficient approximation, SVGP (Hensman et al., 2013; 2015), see Appendix B.

### 3.1 PROBLEM STATEMENT

Consider a training dataset $\mathcal{D} = (\mathcal{X}, \mathbf{y}) = \{(\mathbf{x}_i, y_i)\}_{i=1}^n$, and a pretrained standard NN that outputs a point prediction $\hat{y}_i$ given $\mathbf{x}_i$. The problem is two-fold: (1) Quantify the uncertainty in the predictions of the NN, and (2) calibrate the point predictions of the NN (i.e. make them more accurate).

### 3.2 FRAMEWORK OVERVIEW

RIO solves this problem by modeling the residuals between observed outcomes $y$ and NN predictions $\hat{y}$ using GP with a composite kernel. The framework can be divided into two phases: the training phase and the deployment phase.

In the training phase, the residuals between observed outcomes and NN predictions on the training dataset are calculated as

$$r_i = y_i - \hat{y}_i, \ \ \text{for } i = 1, 2, \ldots, n \,. \tag{1}$$

Let $\mathbf{r}$ denote the vector of all residuals and $\hat{\mathbf{y}}$ denote the vector of all NN predictions. A GP with a composite kernel is trained assuming $\mathbf{r} \sim \mathcal{N}(0, \mathbf{K}_c((\mathcal{X}, \hat{\mathbf{y}}), (\mathcal{X}, \hat{\mathbf{y}})) + \sigma_n^2 \mathbf{I})$, where $\mathbf{K}_c((\mathcal{X}, \hat{\mathbf{y}}), (\mathcal{X}, \hat{\mathbf{y}}))$ denotes an $n \times n$ covariance matrix at all pairs of training points based on a composite kernel

$$k_c((\mathbf{x}_i, \hat{y}_i), (\mathbf{x}_j, \hat{y}_j)) = k_{\text{in}}(\mathbf{x}_i, \mathbf{x}_j) + k_{\text{out}}(\hat{y}_i, \hat{y}_j), \ \ \text{for } i, j = 1, 2, \ldots, n \,. \tag{2}$$

Suppose a radial basis function (RBF) kernel is used for both $k_{\text{in}}$ and $k_{\text{out}}$. Then,

$$k_c((\mathbf{x}_i, \hat{y}_i), (\mathbf{x}_j, \hat{y}_j)) = \sigma_{\text{in}}^2 \exp(-\frac{1}{2l_{\text{in}}^2} \|\mathbf{x}_i - \mathbf{x}_j\|^2) + \sigma_{\text{out}}^2 \exp(-\frac{1}{2l_{\text{out}}^2} \|\hat{y}_i - \hat{y}_j\|^2) \,. \tag{3}$$

The training process of GP learns the hyperparameters $\sigma_{\text{in}}^2$, $l_{\text{in}}$, $\sigma_{\text{out}}^2$, $l_{\text{out}}$, and $\sigma_n^2$ by maximizing the log marginal likelihood $\log p(\mathbf{r}|\mathcal{X}, \hat{\mathbf{y}})$ given by

$$\log p(\mathbf{r}|\mathcal{X}, \hat{\mathbf{y}}) = -\frac{1}{2}\mathbf{r}^\top (\mathbf{K}_c((\mathcal{X}, \hat{\mathbf{y}}), (\mathcal{X}, \hat{\mathbf{y}})) + \sigma_n^2 \mathbf{I})\mathbf{r} - \frac{1}{2}\log |\mathbf{K}_c((\mathcal{X}, \hat{\mathbf{y}}), (\mathcal{X}, \hat{\mathbf{y}})) + \sigma_n^2 \mathbf{I}| - \frac{n}{2}\log 2\pi \,. \tag{4}$$

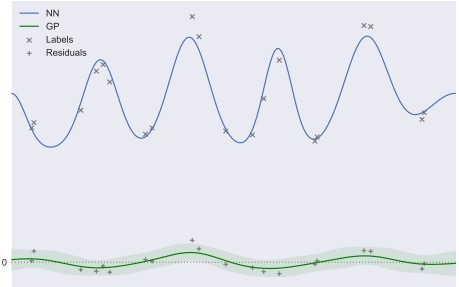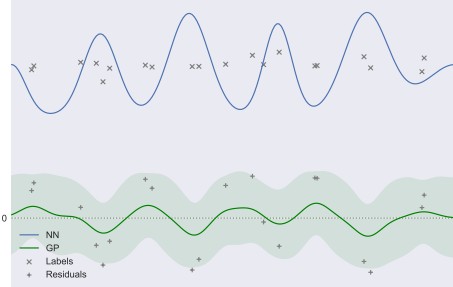

Figure 2: **Capturing uncertainty of more and less accurate NNs.** These figures illustrate the behavior of RIO in two cases: (left) The neural network has discovered true complex structure in the labels, so the residuals have low variance and are easy for the GP to fit with high confidence; (right) The ineffective neural network has introduced unnecessary complexity, so the residuals are modeled with high uncertainty. In both cases, RIO matches the intuition for how uncertain the NN really is.

In the deployment phase, a test point $\mathbf{x}_*$ is input to the NN to get an output $\hat{y}_*$. The trained GP predicts the distribution of the residual as $\hat{r}_*|\mathcal{X}, \hat{\mathbf{y}}, \mathbf{r}, \mathbf{x}_*, \hat{y}_* \sim \mathcal{N}(\bar{\hat{r}}_*, \text{var}(\hat{r}_*))$, where

$$\bar{\hat{r}}_* = \mathbf{k}_*^\top (\mathbf{K}_c((\mathcal{X}, \hat{\mathbf{y}}), (\mathcal{X}, \hat{\mathbf{y}})) + \sigma_n^2 \mathbf{I})^{-1} \mathbf{r} , \tag{5}$$

$$\text{var}(\hat{r}_*) = k_c((\mathbf{x}_*, \hat{y}_*), (\mathbf{x}_*, \hat{y}_*)) - \mathbf{k}_*^\top (\mathbf{K}_c((\mathcal{X}, \hat{\mathbf{y}}), (\mathcal{X}, \hat{\mathbf{y}})) + \sigma_n^2 \mathbf{I})^{-1} \mathbf{k}_* , \tag{6}$$

where $\mathbf{k}_*$ denotes the vector of kernel-based covariances (i.e., $k_c((\mathbf{x}_*, \hat{y}_*), (\mathbf{x}_i, \hat{y}_i))$) between $(\mathbf{x}_*, \hat{y}_*)$ and the training points.

Interestingly, note that the predicted residuals can also be used to calibrate the point predictions of the NN, so that the final calibrated prediction with uncertainty information is given by

$$\hat{y}'_* \sim \mathcal{N}(\hat{y}_* + \bar{\hat{r}}_*, \text{var}(\hat{r}_*)) . \tag{7}$$

In other words, RIO not only adds uncertainty estimation to a standard NN—it also provides a way to calibrate NN predictions, without any modification to its architecture or training. Figure 1 shows the overall procedure when applying the proposed framework in real-world applications. Figure 2 shows example behavior of RIO that illustrates the intuition of the approach. An algorithmic description of RIO is provided in Appendix C.

### 3.3 SCALABILITY

RIO is scalable to large datasets (No. of data points×No. of features>1M) by applying sparse GP methods, e.g., SVGP (Hensman et al., 2013; 2015). All the conclusions in previous sections still remain valid since sparse GP is simply an approximation of the original GP. In the case of applying SVGP with a traditional optimizer, e.g., L-BFGS-B (Byrd et al., 1995; Zhu et al., 1997), the computational complexity is $\mathcal{O}(nm^2)$, and space complexity is $\mathcal{O}(nm)$, where $n$ is the number of data points and $m$ is the number of inducing variables, compared to $\mathcal{O}(n^3)$ and $\mathcal{O}(n^2)$ for traditional GP. Experiments verify that the computational cost of RIO with SVGP is significantly cheaper than other state-of-the-art approaches.

## 4 EMPIRICAL EVALUATION

Experiments in this section compare nine algorithms on 12 real-world datasets. The algorithms include standard NN, the proposed RIO framework, four ablated variants of RIO, and three state-of-the-art models that provide predictive uncertainty: SVGP (Hensman et al., 2013), Neural Network Gaussian Process (NNGP) (Lee et al., 2018), and Attentive Neural Processes (ANP) (Kim et al., 2019). In naming the RIO variants,"R" means estimating NN residuals then correcting NN outputs,"Y" means directly estimating outcomes, "I" means only using input kernel, "O" means only using output kernel, and "IO" means using I/O kernel. For all RIO variants (including full RIO), SVGP is used as the GP component, but using the appropriate kernel and prediction target. Therefore, "Y+I" amounts to original SVGP, and it is denoted as "SVGP" in all the experimental results. All 12 datasets

Table 1: Summary of experimental results

| Dataset n × d | Method | RMSE mean±std | NLPD mean±std | Noise Variance | Time (sec) | Dataset n × d | Method | RMSE mean±std | NLPD mean±std | Noise Variance | Time (sec) |
|---|---|---|---|---|---|---|---|---|---|---|---|
| yacht | NN | 2.30±0.93†‡ | - | - | 4.02 | ENB/h | NN | 1.03±0.51†‡ | - | - | 6.65 |
|  | RIO | **1.46±0.49** | 2.039±0.762 | 0.82 | 7.16 |  | RIO | **0.70±0.38** | **1.038±0.355** | 0.46 | 8.18 |
|  | R+I | 2.03±0.73†‡ | 2.341±0.516†‡ | 2.54 | 4.30 |  | R+I | 0.79±0.46†‡ | 1.147±0.405†‡ | 0.63 | 7.52 |
| 308 | R+O | 1.88±0.66†‡ | 2.305±0.614†‡ | 1.60 | 6.27 | 768 | R+O | 0.80±0.43†‡ | 1.169±0.388†‡ | 0.59 | 7.61 |
| × | Y+O | 1.86±0.64†‡ | 2.305±0.639†‡ | 1.89 | 9.93 | × | Y+O | 0.88±0.48†‡ | 1.248±0.405†‡ | 0.75 | 11.06 |
| 6 | Y+IO | 1.58±0.52†‡ | 2.160±0.773†‡ | 1.18 | 9.44 | 8 | Y+IO | 0.76±0.41†‡ | 1.124±0.368†‡ | 0.56 | 10.64 |
|  | SVGP | 4.42±0.62†‡ | 2.888±0.102†‡ | 18.56 | 8.96 |  | SVGP | 2.13±0.18†‡ | 2.200±0.074†‡ | 4.70 | 10.16 |
|  | NNGP | 12.40±1.45†‡ | 35.18±0.534†‡ | - | 7347 |  | NNGP | 4.97±0.29†‡ | 32.40±0.638†‡ | - | 7374 |
|  | ANP | 7.59±3.20†‡ | **1.793±0.887†‡** | - | 40.82 |  | ANP | 4.08±2.27†‡ | 2.475±0.559†‡ | - | 102.3 |
| ENB/c | NN | 1.88±0.44†‡ | - | - | 6.45 | airfoil | NN | 4.82±0.43†‡ | - | - | 6.48 |
|  | RIO | **1.48±0.33** | **1.816±0.191** | 1.58 | 8.07 |  | RIO | **3.07±0.18** | **2.554±0.053** | 9.48 | 17.63 |
|  | R+I | 1.71±0.44†‡ | 1.969±0.236†‡ | 2.22 | 5.02 |  | R+I | 3.16±0.18†‡ | 2.583±0.051†‡ | 10.07 | 15.90 |
| 768 | R+O | 1.75±0.43†‡ | 2.000±0.229†‡ | 2.25 | 4.57 | 1505 | R+O | 4.17±0.26†‡ | 2.849±0.066†‡ | 16.64 | 9.97 |
| × | Y+O | 1.76±0.43†‡ | 2.000±0.231†‡ | 2.32 | 10.99 | × | Y+O | 4.24±0.28†‡ | 2.869±0.075†‡ | 17.81 | 22.72 |
| 8 | Y+IO | 1.64±0.36†‡ | 1.936±0.210†‡ | 1.96 | 10.56 | 5 | Y+IO | 3.64±0.53†‡ | 2.712±0.150†‡ | 14.40 | 24.51 |
|  | SVGP | 2.63±0.23†‡ | 2.403±0.078†‡ | 6.81 | 10.28 |  | SVGP | 3.59±0.20†‡ | 2.699±0.053†‡ | 12.67 | 21.74 |
|  | NNGP | 4.91±0.32†‡ | 30.14±0.886†‡ | - | 7704 |  | NNGP | 6.54±0.23†‡ | 33.60±0.420†‡ | - | 3355 |
|  | ANP | 4.81±2.15†‡ | 2.698±0.548†‡ | - | 64.11 |  | ANP | 21.17±30.72†‡ | 5.399±6.316†‡ | - | 231.7 |
| CCS | NN | 6.23±0.53†‡ | - | - | 9.46 | wine/r | NN | 0.691±0.041†‡ | - | - | 3.61 |
|  | RIO | **5.97±0.48** | **3.241±0.109** | 24.74 | 13.71 |  | RIO | 0.672±0.036 | 1.094±0.100 | 0.28 | 9.25 |
| 1030 | R+I | 6.01±0.50†‡ | 3.248±0.112†‡ | 25.40 | 9.52 | 1599 | R+I | 0.669±0.036†‡ | 1.085±0.097†‡ | 0.28 | 8.34 |
| × | R+O | 6.17±0.54†‡ | 3.283±0.120†‡ | 26.31 | 9.54 | × | R+O | 0.676±0.035†‡ | 1.099±0.094‡ | 0.29 | 5.02 |
| 8 | Y+O | 6.15±0.52†‡ | 3.279±0.117†‡ | 26.53 | 21.35 | 11 | Y+O | 0.676±0.034†‡ | 1.096±0.092 | 0.29 | 12.71 |
|  | Y+IO | 6.06±0.49†‡ | 3.261±0.110†‡ | 25.82 | 23.15 |  | Y+IO | 0.672±0.036†‡ | 1.094±0.098 | 0.28 | 12.48 |
|  | SVGP | 6.87±0.39†‡ | 3.336±0.048†‡ | 44.55 | 19.85 |  | SVGP | **0.642±0.028†‡** | **0.974±0.042†‡** | 0.40 | 12.17 |
| wine/w | NN | 0.721±0.023†‡ | - | - | 7.17 | CCPP | NN | 4.96±0.53†‡ | - | - | 14.52 |
|  | RIO | 0.704±0.018 | 1.090±0.038 | 0.37 | 16.74 |  | RIO | **4.05±0.128** | **2.818±0.031** | 16.30 | 42.65 |
| 4898 | R+I | **0.699±0.018†‡** | **1.081±0.037†‡** | 0.38 | 13.5 | 9568 | R+I | 4.06±0.13†‡ | 2.822±0.031†‡ | 16.39 | 39.88 |
| × | R+O | 0.710±0.019†‡ | 1.098±0.038†‡ | 0.39 | 6.19 | × | R+O | 4.32±0.15†‡ | 2.883±0.035†‡ | 18.50 | 18.48 |
| 11 | Y+O | 0.710±0.019†‡ | 1.096±0.038†‡ | 0.39 | 18.39 | 4 | Y+O | 4.37±0.20†‡ | 2.914±0.122†‡ | 23.98 | 48.27 |
|  | Y+IO | 0.705±0.019†‡ | 1.090±0.038 | 0.38 | 20.06 |  | Y+IO | 4.56±1.00†‡ | 2.958±0.216†‡ | 31.06 | 46.8 |
|  | SVGP | 0.719±0.018†‡ | **1.081±0.022†‡** | 0.50 | 18.18 |  | SVGP | 4.36±0.13†‡ | 2.893±0.031†‡ | 19.04 | 46.43 |
| protein | NN | 4.21±0.07†‡ | - | - | 151.8 | SC | NN | 12.23±0.77†‡ | - | - | 51.9 |
|  | RIO | **4.08±0.06** | **2.826±0.014** | 15.71 | 149.4 |  | RIO | **11.28±0.46** | **3.853±0.042** | 105.83 | 53.39 |
| 45730 | R+I | 4.11±0.06†‡ | 2.834±0.037†‡ | 15.99 | 141.2 | 21263 | R+I | 11.33±0.45†‡ | 3.858±0.041†‡ | 107.35 | 47.72 |
| × | R+O | 4.14±0.06†‡ | 2.840±0.015†‡ | 16.18 | 115.1 | × | R+O | 11.63±0.52†‡ | 3.881±0.046†‡ | 112.91 | 30.47 |
| 9 | Y+O | 4.14±0.06†‡ | 2.840±0.015†‡ | 16.17 | 138.4 | 81 | Y+O | 11.64±0.53†‡ | 3.882±0.046†‡ | 113.61 | 45.35 |
|  | Y+IO | **4.08±0.06** | **2.826±0.014** | 15.72 | 155.5 |  | Y+IO | 11.32±0.45†‡ | 3.856±0.041†‡ | 106.93 | 57.74 |
|  | SVGP | 4.68±0.04†‡ | 2.963±0.007†‡ | 22.54 | 149.5 |  | SVGP | 14.66±0.25†‡ | 4.136±0.014†‡ | 239.28 | 50.89 |
| CT | NN | 1.17±0.34†‡ | - | - | 194.5 | MSD | NN | 12.42±2.97†‡ | - | - | 1136 |
|  | RIO | **0.88±0.15** | 1.284±0.219 | 1.02 | 516.4 |  | RIO | **9.26±0.21** | **3.639±0.022** | 84.28 | 1993 |
| 53500 | R+I | 1.17±0.34†‡ | 1.538±0.289†‡ | 1.71 | 19.80 | 515345 | R+I | 10.92±1.30†‡ | 3.811±0.128†‡ | 135.34 | 282.0 |
| × | R+O | **0.88±0.15** | 1.283±0.219†‡ | 1.02 | 159.4 | × | R+O | **9.25±0.20** | **3.638±0.021** | 84.05 | 1518 |
| 384 | Y+O | 0.99±0.42†‡ | 1.365±0.385†‡ | 2.45 | 166.3 | 90 | Y+O | 10.00±0.86†‡ | 3.768±0.148†‡ | 169.90 | 1080 |
|  | Y+IO | 0.91±0.16†‡ | **1.280±0.177‡** | 0.62 | 578.6 |  | Y+IO | 9.43±0.52‡ | 3.644±0.025†‡ | 85.66 | 2605 |
|  | SVGP | 52.07±0.19†‡ | 5.372±0.004†‡ | 2712 | 27.56 |  | SVGP | 9.57±0.00†‡ | 3.677±0.000†‡ | 92.21 | 2276 |

The symbols † and ‡ indicate that the difference between the marked entry and RIO is statistically significant at the 5% significance level using paired $t$-test and Wilcoxon test, respectively. The best entries that are significantly better than all the others under at least one statistical test are marked in boldface (ties are allowed).

are real-world regression problems (Dua & Graff, 2017), and cover a wide variety of dataset sizes and feature dimensionalities. Except for the "MSD" dataset, all other datasets are tested for 100 independent runs. During each run, the dataset is randomly split into training set, validation set, and test set, and all algorithms are trained on the same split. All RIO variants that involve an output kernel or residual estimation are based on the trained NN in the same run. For "MSD", since the dataset split is strictly predefined by the provider, only 10 independent runs are conducted. NNGP and ANP are only tested on the four smallest dataset (based on the product of dataset size and feature dimensionality) because they do not scale well to larger datasets. It is notable that for all the RIO variants, no extensive hyperparameter tuning is conducted; the same default setup is used for all experiments, i.e., standard RBF kernel and 50 inducing points. See Appendix D.1 for additional details of experimental setups and a link to downloadable source code. Table 1 summarizes the numerical results from these experiments. The main conclusions in terms of point-prediction error, uncertainty estimation, computational requirements, and ablation studies are summarized below.

**Point-Prediction Error** The errors between point predictions of models and true outcomes of test points are measured using Root Mean Square Error (RMSE); the mean and standard deviation of RMSEs over multiple experimental runs are shown in Table 1. For models that return a probabilistic distribution, the mean of the distribution is the point prediction. Although the main motivation of RIO is to enhance pretrained NN rather than construct a new state-of-the-art prediction model from scratch, RIO performs the best or equals the best method (based on statistical tests) in 10 out of 12 datasets. RIO significantly outperforms original NN in all 12 datasets, while original SVGP performs significantly worse than NN in 7 datasets. For the CT dataset, which has 386 input features, SVGP fails severely since the input kernel cannot capture the implicit correlation information. ANP is unstable on the airfoil dataset because it scales poorly with dataset size. Figure 3 compares NN, RIO

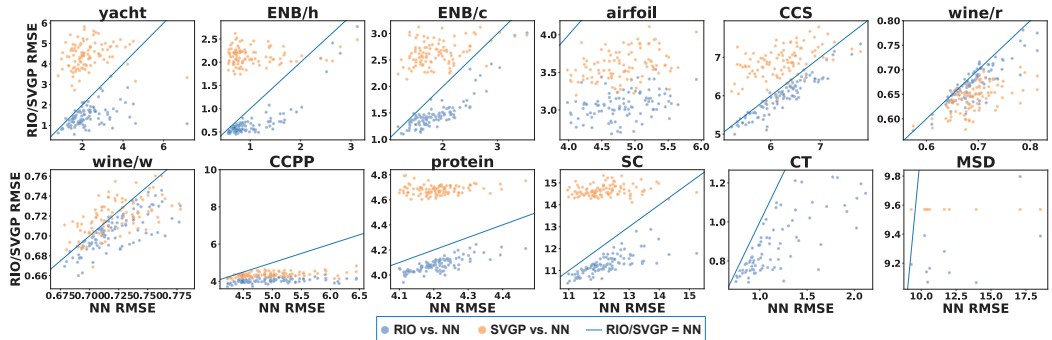

Figure 3: **Comparison among NN, RIO, and SVGP.** The horizontal axis denotes the prediction RMSE of the NN, and the vertical axis the prediction RMSE of RIO (blue dots) and SVGP (yellow dots). Each dot represents an independent experimental run. Since the scales are different, the solid blue line indicates where NN and RIO/SVGP have same prediction RMSE. Thus, a dot below the line means that the method (RIO or SVGP) performs better than the NN, and vice versa. Results of SVGP on the CT dataset are not plotted because its prediction RMSE exceeded the visible scale (i.e. they were >50). RIO consistently reduces the error of the NN, and outperforms SVGP in most cases.

and SVGP in terms of prediction RMSE. RIO is able to improve the NN predictions consistently regardless of how the dataset is split and how well the NN is trained. Even in the situations where original NN is much worse than SVGP, RIO still successfully improves the NN performance into a level that is comparable or better than SVGP. Appendix D.3 provides additional investigation into the behavior of RIO output corrections. RIO exhibits diverse behaviors that generally move the original NN predictions closer to the ground truth. To conclude, applying RIO to NNs not only provides additional uncertainty information, but also reliably reduces the point-prediction error.

**Uncertainty Estimation** Average negative log predictive density (NLPD) is used to measure the quality of uncertainty estimation, which favours conservative models, but also effectively penalizes both over- and under-confident predictions (Quiñonero-Candela et al., 2006). The mean and standard deviation of NLPDs over multiple experimental runs are shown in Table 1 (lower is better). RIO performs the best or equals the best method (based on statistical tests) in 8 out of 12 datasets. NNGP always yields a high NLPD; it returns over-confident predictions, because it does not include noise estimation in its original implementation. For the yacht dataset, ANP achieves the best NLPD, but with high RMSE. This is because ANP is able to correctly return high predictive variance when its prediction error is high. For all other tested datasets, RIO variants consistently outperform ANP. Among all RIO variants, the full RIO provides the most reliable overall predictive uncertainty. The conclusion is that RIO successfully extracts useful uncertainty information from NN predictions. Appendix D.2 evaluates uncertainty estimates on an additional metric: the true coverage probabilities of estimated confidence intervals. Although RIO also provides reasonable estimates for this metric in most cases, comparison among algorithms using this metric is discouraged due to its unreliability (see Appendix D.2 for examples).

**Computation Time** Table 1 shows the average wall clock time of each algorithm. All algorithms are implemented using Tensorflow under the same running environment (see Appendix D.1 for implementation details). The RIO variants scale well to increasing dataset sizes and feature dimensionalities. L-BFGS-B converges especially quickly for R+I on the three highest dimensional datasets, presumably because the residuals are very well-behaved compared to the raw targets or NN output. In contrast, ANP's computation time increases significantly with the scale of the dataset, and NNGP always needs very expensive computational budgets due to its costly grid search of hyperparameters.

**Ablation Study** The RIO variants with residual estimation generally perform better than its counterparts in both point-prediction error and uncertainty estimation. This result confirms the effectiveness of residual estimation, as suggested in Section 5.1. Another important result is that Y+IO outperforms both Y+I (SVGP) and Y+O in most cases across all performance metrics, and RIO generally provides better performance than R+I and R+O in all respects. This result, in turn, confirms that the I/O kernel provides additional robustness, as suggested by the analysis in Section 5.2. In sum, both residual estimation and the I/O kernel contribute substantially to the performance of the framework.

Table 2: Spearman's correlation between RMSE and $\sigma_n^2$ across RIO variants (including SVGP)

| | yacht | ENB/h | ENB/c | airfoil | CCS | wine/r | wine/w | CCPP | protein | SC | CT | MSD |
|---|---|---|---|---|---|---|---|---|---|---|---|---|
| correlation | **0.943** | **0.943** | **1.0** | **1.0** | **0.943** | -0.09 | **0.886** | **1.0** | **0.943** | **1.0** | 0.771 | **0.943** |
| $p$-value | **0.005** | **0.005** | **0.0** | **0.0** | **0.005** | 0.872 | **0.02** | **0.0** | **0.005** | **0.0** | 0.072 | **0.005** |

Entries that are considered to indicate very strong positive monotonic correlation are marked in boldface.

**Correlation between error and noise variance**   Table 2 shows the results of Spearman's Rank Correlation between RMSE and noise variance $\sigma_n^2$. For each dataset, there are six pairs of data points, each of which contains the mean of RMSE and noise variance of a RIO variant. A correlation value larger than 0.8 and $p$-value less than 0.05 indicate a very strong positive monotonic correlation. For 10 out of 12 datasets, very strong positive monotonic correlation between RMSE and noise variance was observed. This empirical result is in accordance with the theoretical prediction in Section 5.1.

**Application to a large-scale vision problem**   To show RIO's off-the-shelf applicability to modern deep convolutional architectures, it was applied to a recent pretrained NN for age estimation (Yang et al., 2018). The pretrained NN and all data preprocessing were taken exactly from the official code release. The model is a variant of DenseNet-121 (Huang et al., 2017), and uses the IMDB age estimation dataset, which contains $\approx$ 172K RGB images (Rothe et al., 2016). The goal is to predict the age of the individual in each image. The features for the GP input kernel were simply a global max pool of the CNN's first stage output. From Table 3, RIO substantially improves upon the mean absolute error (MAE) of the pretrained NN, outperforms SVGP in terms of both MAE and NLPD, and yields realistic confidence intervals (CIs). SVGP effectively learns nothing, so it estimates almost all variance as noise, while RIO effectively augments the pretrained model.

Table 3:  IMDB Age Estimation Results with DenseNet

| Method | MAE | NLPD | 95% CI Coverage | 90% CI Coverage | 68% CI Coverage |
|---|---|---|---|---|---|
| Pretrained DenseNet (Yang et al., 2018) | 7.43 | - | - | - | - |
| SVGP | 36.45 | 5.06 | 0.99 | 0.96 | 0.62 |
| RIO | **6.35** | **3.59** | 0.94 | 0.91 | 0.75 |

CI coverage means the percentage of testing outcomes that are within the estimated CI.

# 5   ANALYSIS OF RIO

This section presents an initial theoretical investigation into what makes RIO work, including why it is useful to fit residuals, and why it is beneficial to use an I/O kernel.

## 5.1   BENEFITS OF FITTING NEURAL NETWORK RESIDUALS

Given a pretrained NN, this section provides a theoretical perspective on how fitting its residuals with a GP can yield useful uncertainty information, while leading to prediction error lower than the NN or GP alone. The idea is that, due to its high expressivity, a pretrained NN may have learned complex structure that a GP would model poorly, when selecting a kernel that would capture this complexity is infeasible. Fitting a GP to the residuals of the NN is easier, since this complex structure has been removed, and takes advantage of the predictive power of the NN, while providing useful uncertainty estimates.

Given a pretrained NN, when is fitting its residuals with a GP a good idea? To produce a model with uncertainty information, one could simply discard the NN and train a GP directly on $\mathcal{D}$. However, for problems with enough complexity, GP model selection (i.e., specifying an appropriate class of kernels) is challenging. Fortunately, the pretrained NN may have learned a useful representation of this complexity, which can then be exploited. Suppose instead that a GP tries to model this complexity, and this attempt leads to worse generalization. This poor performance could come from many dimensions of the kernel selection process. As an initial theoretical investigation, an ideal case is considered first: suppose the GP optimizer optimally avoids modeling this complexity incorrectly by estimating the variance of such complexity as noise. Such an estimate leads to the following decomposition:

$$y_i = h(\mathbf{x}_i) + \xi_i = f(\mathbf{x}_i) + g(\mathbf{x}_i) + \xi_i, \tag{8}$$

where $h(\cdot)$ is the true signal, $f(\cdot)$ is the apparent signal of the GP, $g(\cdot)$ is its apparent (spurious) noise, and $\xi_i \sim \mathcal{N}(0, \sigma_n^2)$ is real noise. Intuitively, due to its high expressivity, it is possible for a pretrained NN $\bar{h}_{\text{NN}}$ to correctly model part of $g$, along with part of $f$, resulting in the residuals

$$y_i - \bar{h}_{\text{NN}}(\mathbf{x}_i) = r(\mathbf{x}_i) + \xi_i = r_f(\mathbf{x}_i) + r_g(\mathbf{x}_i) + \xi_i, \tag{9}$$

where $r_f(\cdot)$ is now the apparent signal of the GP, and $r_g(\cdot)$ is now its apparent noise. In such a case, it will be easier for the GP to learn the the residuals than to learn $h$ directly. The uncertainty estimates of the GP will also be more precise, since the NN removes spurious noise. Thus, Eq. 8 immediately predicts that the noise estimates for GP will be higher when fitting $h$ directly than when fitting NN residuals. This prediction is confirmed in experiments (Table 1).

To analyze these behaviors, the above decomposition is used to identify a class of scenarios for which fitting residuals leads to improvements (see Appendix A for additional details and proofs of following theorems). For simplicity, suppose GP kernels are parameterized only by their signal variance $\beta$, i.e., kernels are of the form $\beta k(\cdot, \cdot)$ for some kernel $k$. Then, the following two assumptions make Eq. 8 concrete: (1) GP is well-suited for $f$, i.e., $f(\cdot) \sim \mathcal{GP}(0, k(\cdot, \cdot))$; (2) GP is ill-suited for $g$, i.e., $g$ is independent of $f$ and $\varepsilon$-indistinguishable for GP on $\mathcal{D}$ (see Lemma A.2 for an example of such a function). Intuitively, this means the predictions of GP change by no more than $\varepsilon$ in the presence of $g$.

Consider the residuals $y_i - \bar{h}_{\text{NN}}(\mathbf{x}_i) = r(\mathbf{x}_i) + \xi_i = r_f(\mathbf{x}_i) + r_g(\mathbf{x}_i) + \xi_i$, where $r_f$ is the remaining GP component, and $r_g$ is the remainder of $g$. The following assumption ensures that $r_g$ captures any spurious complexity added by an imperfectly trained NN: $r_f \sim \mathcal{GP}(0, \alpha k(\cdot, \cdot))$, for some $\alpha \in (0, 1]$.

Let $\bar{h}_{\text{GP}}$ be the GP predictor trained directly on $\mathcal{D}$, and $\bar{h}_{\text{GP+NN}} = \bar{h}_{\text{NN}} + \bar{r}_{\text{GP}}$ be the final predictor after fitting residuals; let $E_{\text{GP}}^h$, $E_{\text{NN}}^h$, and $E_{\text{GP+NN}}^h$ be the expected generalization errors of $\bar{h}_{\text{GP}}$, $\bar{h}_{\text{NN}}$, and $\bar{h}_{\text{GP+NN}}$, resp. The advantage of $\bar{h}_{\text{GP+NN}}$ becomes clear as $g$ becomes complex:

**Theorem 5.1** (Advantage of fitting residuals).

$$\lim_{\varepsilon \to 0} \left( E_{\text{GP}}^h - E_{\text{GP+NN}}^h \right) \geq \mathbb{E}[g^2(\mathbf{x})] - \mathbb{E}[r_g^2(\mathbf{x})] \quad \text{and} \quad \lim_{\varepsilon \to 0} \left( E_{\text{NN}}^h - E_{\text{GP+NN}}^h \right) > 0.$$

The experimental results in Section 4 support this result (Table 1). First, in all experiments, GP+NN does indeed improve over the underlying NN. Second, notice that the improvement of GP+NN over GP is only guaranteed when $\mathbb{E}[g^2(\mathbf{x})] - \mathbb{E}[r_g^2(\mathbf{x})]$ is positive, i.e., when the NN successfully captures some underlying complexity. In experiments, when the NN is deficient, GP+NN often outperforms GP, but not always. For example, if the NN is overfit to zero training loss, and uses the same training data as the GP, the GP cannot provide benefit. However, when the NN is well-trained, using proper overfitting prevention, the improvements are consistent and significant.

The prediction of Theorem 5.1 that lower apparent noise corresponds to improved error is confirmed in experiments (Table 2). Note that an optimal model would learn $h$ exactly, and have a predictive variance of $\sigma_n^2$. So, the improvement in error of $\bar{h}_{\text{GP+NN}}$ over $\bar{h}_{\text{GP}}$ also corresponds to a predictive variance that is closer to that of the optimal model, since some spurious noise has been removed by $\bar{h}_{\text{NN}}$. Complementing this theoretical improvement in uncertainty estimation, the above setting also leads to a key practical property, which is illustrated in Figure 2:

**Theorem 5.2.** *The uncertainty of $\bar{r}_{\text{GP}}$ is positively correlated with the variance of NN residuals.*

This property matches the intuition that the GP's variance should generally be higher for unstable NNs, i.e., NNs with high residual variance, than for stable NNs. Such a property is crucial to measuring the confidence of NN predictions in practice.

Overall, the initial theoretical results in this section (and the appendix) show that if a problem is well-suited to be learned by an NN, and a practitioner has access to a pretrained NN for that problem, then training a GP to fit the residuals of the NN can provide potential benefits in uncertainty estimates, without sacrificing prediction performance. Further theoretical investigation regarding more complicated situations, e.g., misspecification of GP kernels, is considered as one interesting direction for future work.

## 5.2 ROBUSTNESS OF THE I/O KERNEL

This section provides a justification for why a GP using the proposed I/O kernel is more robust than the standard GP, i.e., using the input kernel alone. The key assumption is that the output of an NN can

contain valuable information about its behavior, and, consequently, the structure of the target function. Consider the setup in Section 5.1, but now with $y_i = f_{\text{in}}(\mathbf{x}_i) + f_{\text{out}}(\mathbf{x}_i) + \xi_i$, where $f_{\text{in}} \sim GP(0, k_{\text{in}})$ and $f_{\text{out}} \sim GP(0, k_{\text{out}})$, with non-trivial RBF kernels $k_{\text{in}}$, $k_{\text{out}}$ (as in Equation 3). Let $\bar{h}_{\text{NN}}$ be an NN of sufficient complexity to be nontrivially non-affine, in that there exists a positive-measure set of triples $(\mathbf{x}, \mathbf{x}', \mathbf{x}'')$ s.t. $\|\mathbf{x} - \mathbf{x}'\| = \|\mathbf{x} - \mathbf{x}''\|$, but $\|\bar{h}_{\text{NN}}(\mathbf{x}) - \bar{h}_{\text{NN}}(\mathbf{x}')\| \neq \|\bar{h}_{\text{NN}}(\mathbf{x}) - \bar{h}_{\text{NN}}(\mathbf{x}'')\|$. Denote the generalization errors of the standard GP, GP with output kernel only, and GP with I/O kernel by $E_{\text{I}}^h$, $E_{\text{O}}^h$, and $E_{\text{I/O}}^h$, respectively. The expected result follows (proof in Appendix A).

**Theorem 5.3** (Advantage of I/O kernel). $E_{\text{I/O}}^h < E_{\text{I}}^h$ and $E_{\text{I/O}}^h < E_{\text{O}}^h$.

The optimizer associated with the GP simultaneously optimizes the hyperparameters of both kernels, so the less useful kernel usually receives a smaller signal variance. As a result, the I/O kernel is resilient to failures of either kernel. In particular, the GP using I/O kernel improves performance even in the case where the problem is so complex that Euclidean distance in the input space provides no useful correlation information or when the input space contains some noisy features. Conversely, when the NN is a bad predictor, and $h_{\text{NN}}$ is no better noise, the standard GP with input kernel alone is recovered. In other words, the I/O kernel is never worse than using the input kernel alone, and in practice it is often better. This conclusion is confirmed in the experiments in Section 4.

# 6 DISCUSSION AND FUTURE DIRECTIONS

In addition to the reliable uncertainty estimation, accurate point prediction, and good scalability, demonstrated in Section 4, RIO provides other important benefits.

RIO can be directly applied to any standard NN without modification to the model architecture or training pipeline. Moreover, retraining of the NN or change of inference process are not required. The framework simply requires the outputs of an NN; it does not need to access any internal structure. This feature makes the framework more accessible to practitioners in real-world applications, e.g., data scientists can train NNs using traditional pipelines, then directly apply RIO to the trained NNs.

RIO also provides robustness to a type of adversarial attack. Consider a worst-case scenario, in which an adversary can arbitrarily alter the output of the NN with minuscule changes to the input. It is well-known that there are NNs for which this is possible (Goodfellow et al., 2015). In this case, with the help of the I/O kernel, the model becomes highly uncertain with respect to the output kernel. A confident prediction then requires both input and output to be reasonable. In the real world, a high degree of uncertainty may meet a threshold for disqualifying the prediction as outside the scope of the model's ability.

There are several promising future directions for extending RIO: First, applying RIO to reinforcement learning (RL) algorithms, which usually use standard NNs for reward predictions, would allow uncertainty estimation of the future rewards. Agents can then directly employ efficient exploration strategies, e.g., bandit algorithms (Thompson, 1933), rather than traditional stochastic approaches like $\varepsilon$-greedy. Second, RIO applied to Bayesian optimization (BO) (Močkus, 1975) would make it possible to use standard NNs in surrogate modeling. This approach can potentially improve the expressivity of the surrogate model and the scalability of BO. Third, since RIO only requires access to the inputs and outputs of NNs, it could be directly applied to any existing prediction models, including hybrid and ensemble models. For example, experiments in Appendix D.6 show that RIO achieves good results when directly applied to random forests. This general applicability makes RIO a more general tool for real-world practitioners.

# 7 CONCLUSION

The RIO framework both provides estimates of predictive uncertainty of neural networks, and reduces their point-prediction errors. The approach captures NN behavior by estimating their residuals with an I/O kernel. RIO is theoretically-grounded, performs well on several real-world problems, and, by using a sparse GP approximation, scales well to large datasets. Remarkably, it can be applied directly to any pretrained NNs without modifications to model architecture or training pipeline. Thus, RIO can be used to make NN regression practical and powerful in many real-world applications.

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

## A   ADDITIONAL DETAILS AND PROOFS FOR SECTIONS 5.1 AND 5.2

Section 5.1 provides a theoretical perspective on how using GP to fit NN residuals can provide advantages over GP or NN alone. The idea is to decompose the target function into two independent parts: one that is well-suited for a GP to learn, and the other that is difficult for it to capture. The following definition provides a formalization of this notion of difficulty.

**Definition A.1** ($\varepsilon$-indistinguishable). Let $\bar{f}_A$ be the predictor of a learner $A$ trained on $\{(\mathbf{x}_i, y_i)\}_{i=1}^n$, and $\bar{h}_A$ that of $A$ trained on $\mathcal{D} = \{(\mathbf{x}_i, y_i + g(\mathbf{x}_i))\}_{i=1}^n$. Then, if $|\bar{f}_A(\mathbf{x}) - \bar{h}_A(\mathbf{x})| < \varepsilon \, \forall \, \mathbf{x}$, we say $g$ *is $\varepsilon$-indistinguishable for A on $\mathcal{D}$.*[1]

In other words, the predictions of $A$ change by no more than $\varepsilon$ in the presence of $g$. If $\varepsilon$ is small compared to $\mathbb{E}[g^2(\mathbf{x})]$, then $g$ contains substantial complexity that $A$ cannot capture.

For notation, let $\bar{\psi}_A$ denote the predictor of a learner $A$ trained on $\{(\mathbf{x}_i, \psi(\mathbf{x}_i) + \xi_i)\}_{i=1}^n$, with $\xi_i \sim \mathcal{N}(0, \sigma_n^2)$. Let $E_A^\psi = \mathbb{E}[(\psi(\mathbf{x}) - \bar{\psi}_A(\mathbf{x}))^2]$ denote the expected generalization error of $A$ on $\psi$ given training locations $\{\mathbf{x}_i\}_{i=1}^n$. Now, consider a dataset $\mathcal{D} = \{(\mathbf{x}_i, y_i)\}_{i=1}^n$ with

$$y_i = h(\mathbf{x}_i) + \xi_i = f(\mathbf{x}_i) + g(\mathbf{x}_i) + \xi_i,$$

where $f(\cdot) \sim \mathcal{GP}(0, k(\cdot, \cdot))$, $\xi_i \sim \mathcal{N}(0, \sigma_n^2)$, and $\mathbb{E}[g^2(\mathbf{x})] = \sigma_g^2$. For simplicity, assume the GP learner selects optimal hyperparameters for $k$ and $\sigma_n^2$. Suppose $g$ is independent of $f$ and $\varepsilon$-indistinguishable for GP on $\mathcal{D}$, but a neural network $\bar{h}_{\mathrm{NN}}$ may have successfully learned some of $g$'s structure. These assumptions induce a concrete class of examples of the decomposition in Eq. 8. To demonstrate the existence of such decompositions, Lemma A.2 provides a constructive example of such a function $g$.

**Lemma A.2** (Construction of independent $\varepsilon$-indistinguishable functions for GP). *Given a dataset $\{(\mathbf{x}_i, f(\mathbf{x}_i) + \xi_i)\}_{i=1}^n$, for any $\sigma_g^2$ and $\varepsilon > 0$, there exists $g$ with $\mathbb{E}[g^2(\mathbf{x})] = \sigma_g^2$, and $\{\mathbf{x}_i\}_{i=n+1}^{2n}$, s.t. $g$ is $\varepsilon$-indistinguishable on $\{(\mathbf{x}_i, f(\mathbf{x}_i) + g(\mathbf{x}_i) + \xi_i)\}_{i=1}^{2n}$ for GP with a continuous kernel $k$.*

*Proof.* Let $\mathbf{x}_{n+i} = \mathbf{x}_i + \mathbf{v} \, \forall \, i = 1, \ldots, n$, for a fixed vector $\mathbf{v}$. Consider the linear smoother view of GP prediction (Rasmussen & Williams, 2006): $\bar{f}_{\mathrm{GP}}(\mathbf{x}_*) = \mathbf{w}(\mathbf{x}_*)^\top \mathbf{y}$, where $\mathbf{w}(\mathbf{x}_*) = (K + \sigma_n^2 I)^{-1} \mathbf{k}(\mathbf{x}_*)$ is the *weight function* (Silverman et al., 1984), and $\mathbf{y}$ is the target vector. Since $k$ and $\mathbf{w}(\mathbf{x}_*)$ are continuous around each $\mathbf{x}_i$, we can choose $\mathbf{v}$ such that $\forall \, \mathbf{x}_*$

$$|\mathbf{w}(\mathbf{x}_*)_i - \mathbf{w}(\mathbf{x}_*)_{n+i}| < \frac{\varepsilon}{n\sigma_g^2} \, \forall \, i = 1, \ldots, n.$$

Let $\bar{f}_{\mathrm{GP}}$ be the predictor of GP trained on $\{(\mathbf{x}_i, f(\mathbf{x}_i) + \xi_i)\}_{i=1}^{2n}$, and $\bar{h}_{\mathrm{GP}}$ the predictor trained on $\{(\mathbf{x}_i, f(\mathbf{x}_i) + g(\mathbf{x}_i) + \xi_i)\}_{i=1}^{2n} = \mathcal{D}$. Now, choose $g$ so that

$$g(\mathbf{x}_i) = \begin{cases} \sigma_g^2 & \text{if } i \leq n, \\ -\sigma_g^2 & \text{if } i > n. \end{cases}$$

Then,

$$\bar{h}_{\mathrm{GP}}(\mathbf{x}_*) = \mathbf{w}(\mathbf{x}_*)^\top (f(\mathbf{x}_i) + g(\mathbf{x}_i) + \xi_i)_{i=1}^{2n} = \mathbf{w}(\mathbf{x}_*)^\top (f(\mathbf{x}_i) + \xi_i)_{i=1}^{2n} + \mathbf{w}(\mathbf{x}_*)^\top (g(\mathbf{x}_i))_{i=1}^{2n}$$

$$= \bar{f}_{\mathrm{GP}}(\mathbf{x}_*) + \sum_{i=1}^n \sigma_g^2 \mathbf{w}(\mathbf{x}_*)_i - \sum_{i=1}^n \sigma_g^2 \mathbf{w}(\mathbf{x}_*)_{n+i} = \bar{f}_{\mathrm{GP}}(\mathbf{x}_*) + \sigma_g^2 \sum_{i=1}^n \left(\mathbf{w}(\mathbf{x}_*)_i - \mathbf{w}(\mathbf{x}_*)_{n+i}\right).$$

---

[1]Note that a related information theoretic notion of indistinguishability has recently been defined for analyzing the limits of deep learning (Abbe & Sandon, 2018).

$$\left| \sigma_g^2 \sum_{i=1}^{n} \left( \mathbf{w}(\mathbf{x}_*)_i - \mathbf{w}(\mathbf{x}_*)_{n+i} \right) \right| < \sigma_g^2 n \left( \frac{\varepsilon}{n\sigma_g^2} \right) = \varepsilon \implies |\bar{f}_{\mathrm{GP}}(\mathbf{x}_*) - \bar{h}_{\mathrm{GP}}(\mathbf{x}_*)| < \varepsilon.$$

$\square$

Consider the residuals $y_i - \bar{h}_{\mathrm{NN}}(\mathbf{x}_i) = r(\mathbf{x}_i) + \xi_i = r_f(\mathbf{x}_i) + r_g(\mathbf{x}_i) + \xi_i$. Here, $r_f$ is the remaining GP component, i.e., $r_f \sim \mathcal{GP}(0, \alpha k(\cdot, \cdot))$, for $0 < \alpha \le 1$. Similarly, $r_g$ is the remainder of $g$ $\varepsilon$-indistinguishable for GP on $\{(\mathbf{x}_i, r(\mathbf{x}_i) + \xi_i)\}_{i=1}^{n}$, i.e., $\sigma_g^2 - \mathbb{E}[r_g^2(\mathbf{x})] = \delta$. These two assumptions about $r_f$ and $r_g$ ensure that any new spurious complexity added by the NN is collected in $r_g$, since the biggest advantage of the NN (its expressivity) is also its greatest source of risk. The final predictor after fitting residuals is then $\bar{h}_{\mathrm{GP+NN}} = \bar{h}_{\mathrm{NN}} + \bar{r}_{\mathrm{GP}}$. The following sequence of results captures the improvement due to residual estimation.

**Lemma A.3** (Generalization of GP on $h$).

$$E_{\mathrm{GP}}^f + \sigma_g^2 - 2\varepsilon(E_{\mathrm{GP}}^f)^{\frac{1}{2}} - 2\varepsilon\sigma_g < E_{\mathrm{GP}}^h < E_{\mathrm{GP}}^f + \sigma_g^2 + 2\varepsilon(E_{\mathrm{GP}}^f)^{\frac{1}{2}} + 2\varepsilon\sigma_g + \varepsilon^2.$$

*Proof.* Let $\Delta \bar{f}_{\mathrm{GP}}(\mathbf{x}) = \bar{h}_{\mathrm{GP}}(\mathbf{x}) - \bar{f}_{\mathrm{GP}}(\mathbf{x})$ denote the change in GP prediction due to $g$. Then,

$$
\begin{aligned}
E_{\mathrm{GP}}^h &= \mathbb{E}[(h(\mathbf{x}) - \bar{h}_{\mathrm{GP}}(\mathbf{x}))^2] = \mathbb{E}[((f(\mathbf{x}) + g(\mathbf{x})) - (\bar{f}_{\mathrm{GP}}(\mathbf{x}) + \Delta\bar{f}_{\mathrm{GP}}(\mathbf{x})))^2] \\
&= \mathbb{E}[((f(\mathbf{x}) - \bar{f}_{\mathrm{GP}}(\mathbf{x})) + (g(\mathbf{x}) - \Delta\bar{f}_{\mathrm{GP}}(\mathbf{x})))^2] \\
&= E_{\mathrm{GP}}^f + 2\mathbb{E}[(f(\mathbf{x}) - \bar{f}_{\mathrm{GP}}(\mathbf{x}))(g(\mathbf{x}) - \Delta\bar{f}_{\mathrm{GP}}(\mathbf{x}))] + \mathbb{E}[(g(\mathbf{x}) - \Delta\bar{f}_{\mathrm{GP}}(\mathbf{x}))^2] \\
&= E_{\mathrm{GP}}^f - 2\mathbb{E}[(f(\mathbf{x}) - \bar{f}_{\mathrm{GP}}(\mathbf{x}))\Delta\bar{f}_{\mathrm{GP}}(\mathbf{x})] + \mathbb{E}[(g(\mathbf{x}) - \Delta\bar{f}_{\mathrm{GP}}(\mathbf{x}))^2],
\end{aligned}
$$

where the last line makes use of the fact that $f$ and $g$ are independent. Now,

$$\left| 2\mathbb{E}[(f(\mathbf{x}) - \bar{f}_{\mathrm{GP}}(\mathbf{x}))\Delta\bar{f}_{\mathrm{GP}}(\mathbf{x}))] \right| < 2\varepsilon(E_{\mathrm{GP}}^f)^{\frac{1}{2}},$$

and

$$\mathbb{E}[(g(\mathbf{x}) - \Delta\bar{f}_{\mathrm{GP}}(\mathbf{x}))^2] = \sigma_g^2 - 2\mathbb{E}[g(\mathbf{x})\Delta\bar{f}_{\mathrm{GP}}(\mathbf{x})] + \mathbb{E}[\Delta\bar{f}_{\mathrm{GP}}(\mathbf{x})^2],$$

where

$$\left| 2\mathbb{E}[g(\mathbf{x})\Delta\bar{f}_{\mathrm{GP}}(\mathbf{x})] \right| < 2\varepsilon\sigma_g \quad \text{and} \quad 0 \le \mathbb{E}[\Delta\bar{f}_{\mathrm{GP}}(\mathbf{x})^2] < \varepsilon^2.$$

So, $E_{\mathrm{GP}}^f + \sigma_g^2 - 2\varepsilon(E_{\mathrm{GP}}^f)^{\frac{1}{2}} - 2\varepsilon\sigma_g < E_{\mathrm{GP}}^h < E_{\mathrm{GP}}^f + \sigma_g^2 + 2\varepsilon(E_{\mathrm{GP}}^f)^{\frac{1}{2}} + 2\varepsilon\sigma_g + \varepsilon^2.$ $\square$

**Lemma A.4** (Generalization of NN). $E_{\mathrm{NN}}^h = \alpha\mathbb{E}[f^2(\mathbf{x})] + \sigma_g^2 - \delta.$

*Proof.* Making use of the fact that $\mathbb{E}[r_f] = 0$,

$$E_{\mathrm{NN}}^h = \mathbb{E}[r^2(x)] = \mathbb{E}[(r_f(x) + r_g(x))^2] = \mathbb{E}[r_f^2(x)] + \mathbb{E}[r_g^2(x)].$$

Now, $r_f \sim \mathrm{GP}(0, \alpha k) \implies \mathbb{E}[r_f^2(x)] = \alpha\mathbb{E}[f^2(\mathbf{x})]$, and $\sigma_g^2 - \mathbb{E}[r_g^2(\mathbf{x})] = \delta \implies \mathbb{E}[r_g^2(\mathbf{x})] = \sigma_g^2 - \delta$. So, $E_{\mathrm{NN}}^h = \alpha\mathbb{E}[f^2(\mathbf{x})] + \sigma_g^2 - \delta$. $\square$

**Lemma A.5** (Generalization of GP fitting residuals).

$$E_{\mathrm{GP}}^{r_f} + \sigma_g^2 - \delta - 2\varepsilon(E_{\mathrm{GP}}^{r_f})^{\frac{1}{2}} - 2\varepsilon(\sigma_g^2 - \delta)^{\frac{1}{2}} < E_{\mathrm{GP+NN}}^h < E_{\mathrm{GP}}^{r_f} + \sigma_g^2 - \delta + 2\varepsilon(E_{\mathrm{GP}}^{r_f})^{\frac{1}{2}} + 2\varepsilon(\sigma_g^2 - \delta)^{\frac{1}{2}} + \varepsilon^2.$$

*Proof.* Let $\Delta\bar{r}_{f\mathrm{GP}}(\mathbf{x}) = \bar{r}_{\mathrm{GP}}(\mathbf{x}) - \bar{r}_{f\mathrm{GP}}(\mathbf{x})$ denote the change in GP prediction due to $r_g$. Then,

$$
\begin{aligned}
E_{\mathrm{GP+NN}}^h &= \mathbb{E}[(h(\mathbf{x}) - \bar{h}_{\mathrm{GP+NN}}(\mathbf{x}))^2] = \mathbb{E}[(h(\mathbf{x}) - \bar{h}_{\mathrm{NN}}(\mathbf{x}) - \bar{r}_{\mathrm{GP}}(\mathbf{x}))^2] \\
&= \mathbb{E}[(r_f(\mathbf{x}) + r_g(\mathbf{x})) - (\bar{r}_{f\mathrm{GP}}(\mathbf{x}) + \Delta\bar{r}_{f\mathrm{GP}}(\mathbf{x})))^2] \\
&= \mathbb{E}[((r_f(\mathbf{x}) - \bar{r}_{f\mathrm{GP}}(\mathbf{x})) + (r_g(\mathbf{x}) - \Delta\bar{r}_{f\mathrm{GP}}(\mathbf{x})))^2] \\
&= E_{\mathrm{GP}}^{r_f} + 2\mathbb{E}[(r_f(\mathbf{x}) - \bar{r}_{\mathrm{GP}}(\mathbf{x}))(r_g(\mathbf{x}) - \Delta\bar{r}_{f\mathrm{GP}}(\mathbf{x}))] + \mathbb{E}[(r_g(\mathbf{x}) - \Delta\bar{r}_{f\mathrm{GP}}(\mathbf{x}))^2] \\
&= E_{\mathrm{GP}}^{r_f} - 2\mathbb{E}[(r_f(\mathbf{x}) - \bar{r}_{\mathrm{GP}}(\mathbf{x})\Delta\bar{r}_{f\mathrm{GP}}(\mathbf{x})] + \mathbb{E}[(r_g(\mathbf{x}) - \Delta\bar{r}_{f\mathrm{GP}}(\mathbf{x}))^2].
\end{aligned}
$$

Similar to the case of Lemma A.3,

$$\left| 2\mathbb{E}[(r_f(\mathbf{x}) - \bar{r}_{\mathrm{GP}}(\mathbf{x}))(r_g(\mathbf{x}) - \Delta\bar{r}_{f\mathrm{GP}}(\mathbf{x}))] \right| < 2\varepsilon(E_{\mathrm{GP}}^{r_f})^{\frac{1}{2}},$$

and

$$\mathbb{E}[(r_g(\mathbf{x}) - \Delta\bar{r}_{f\mathrm{GP}}(\mathbf{x}))^2] = \sigma_g^2 - \delta - 2\mathbb{E}[r_g(\mathbf{x})\Delta\bar{r}_{f\mathrm{GP}}(\mathbf{x})] + \mathbb{E}[\Delta\bar{r}_{f\mathrm{GP}}(\mathbf{x})^2],$$

where

$$\left|2\mathbb{E}[r_g(\mathbf{x})\Delta\bar{r}_{f\mathrm{GP}}(\mathbf{x})]\right| < 2\varepsilon(\sigma_g^2 - \delta)^{\frac{1}{2}} \quad \text{and} \quad 0 \leq \mathbb{E}[\Delta\bar{r}_{f\mathrm{GP}}(\mathbf{x})^2] < \varepsilon^2.$$

So,

$$E_{\mathrm{GP}}^{r_f} + \sigma_g^2 - \delta - 2\varepsilon(E_{\mathrm{GP}}^{r_f})^{\frac{1}{2}} - 2\varepsilon(\sigma_g^2-\delta)^{\frac{1}{2}} < E_{\mathrm{GP+NN}}^h < E_{\mathrm{GP}}^{r_f} + \sigma_g^2 - \delta + 2\varepsilon(E_{\mathrm{GP}}^{r_f})^{\frac{1}{2}} + 2\varepsilon(\sigma_g^2-\delta)^{\frac{1}{2}} + \varepsilon^2.$$
$$\square$$

**Lemma A.6** (Generalization of GP on $f$ and $r_f$). *From the classic result (Opper & Vivarelli, 1999; Sollich, 1999; Rasmussen & Williams, 2006): Consider the eigenfunction expansion $k(\mathbf{x}, \mathbf{x}') = \sum_j \lambda_j \phi_j(\mathbf{x})\phi_j(\mathbf{x}')$ and $\int k(\mathbf{x},\mathbf{x}')\phi_i(\mathbf{x})p(x)d\mathbf{x} = \lambda_i\phi_i(\mathbf{x}')$. Let $\Lambda$ be the diagonal matrix of the eigenvalues $\lambda_j$, and $\Phi$ be the design matrix, i.e., $\Phi_{ji} = \phi_j(\mathbf{x}_i)$. Then,*

$$E_{\mathrm{GP}}^f = \mathrm{tr}(\Lambda^{-1} + \sigma_n^{-2}\Phi\Phi^\top)^{-1} \quad \text{and} \quad E_{\mathrm{GP}}^{r_f} = \mathrm{tr}(\alpha^{-1}\Lambda^{-1} + \sigma_n^{-2}\Phi\Phi^\top)^{-1}.$$

**Theorem 5.1.** $\lim_{\varepsilon\to 0}\left(E_{\mathrm{GP}}^h - E_{\mathrm{GP+NN}}^h\right) \geq \delta$ and $\lim_{\varepsilon\to 0}\left(E_{\mathrm{NN}}^h - E_{\mathrm{GP+NN}}^h\right) > 0.$

*Proof.* From Lemmas A.3, A.4, and A.5 we have, resp.:

$$\lim_{\varepsilon\to 0} E_{\mathrm{GP}}^h = E_{\mathrm{GP}}^f + \sigma_g^2, \quad \lim_{\varepsilon\to 0} E_{\mathrm{NN}}^h = \alpha\mathbb{E}[f^2(\mathbf{x})] + \sigma_g^2 - \delta, \quad \text{and} \quad \lim_{\varepsilon\to 0} E_{\mathrm{GP+NN}}^h = E_{\mathrm{GP}}^{r_f} + \sigma_g^2 - \delta.$$

From Lemma A.6, we have

$$E_{\mathrm{GP}}^f - E_{\mathrm{GP}}^{r_f} = \mathrm{tr}(\Lambda^{-1} + \sigma_n^{-2}\Phi\Phi^\top)^{-1} - \mathrm{tr}(\alpha^{-1}\Lambda^{-1} + \sigma_n^{-2}\Phi\Phi^\top)^{-1} \geq 0,$$

and

$$\alpha\mathbb{E}[f^2(\mathbf{x})] - E_{\mathrm{GP}}^{r_f} = \alpha\mathbb{E}[f^2(\mathbf{x})] - \mathrm{tr}(\alpha^{-1}\Lambda^{-1} + \sigma_n^{-2}\Phi\Phi^\top)^{-1} > 0.$$

So,

$$\lim_{\varepsilon\to 0}\left(E_{\mathrm{GP}}^h - E_{\mathrm{GP+NN}}^h\right) \geq \delta \quad \text{and} \quad \lim_{\varepsilon\to 0}\left(E_{\mathrm{NN}}^h - E_{\mathrm{GP+NN}}^h\right) > 0.$$
$$\square$$

**Theorem 5.2.** The variance of NN residuals is positively correlated with the uncertainty of $r_{\mathrm{GP}}$.

*Proof.* Increases in $\mathbb{E}[r_f^2(x)]$ lead to increases in $\alpha$; increases in $\mathbb{E}[r_g^2(x)]$ lead to decreases in $\delta$, and thus increases in the estimated noise level $\hat{\sigma}_n^2$. So, an increase in either $\mathbb{E}[r_f^2(x)]$ or $\mathbb{E}[r_g^2(x)]$ leads to an increase in $\alpha k((\mathbf{x}_*, \hat{y}_*), (\mathbf{x}_*, \hat{y}_*)) - \alpha\mathbf{k}_*^\top(\alpha\mathbf{K}((\mathcal{X}, \hat{\mathbf{y}}), (\mathcal{X}, \hat{\mathbf{y}})) + \hat{\sigma}_n^2\mathbf{I})^{-1}\alpha\mathbf{k}_*$, which is the predictive variance of $r_{\mathrm{GP}}$. $\square$

**Theorem 5.3.** $E_{\mathrm{I/O}}^h < E_{\mathrm{I}}^h$ and $E_{\mathrm{I/O}}^h < E_{\mathrm{O}}^h.$

*Proof.*
$$\|\mathbf{x} - \mathbf{x}'\| = \|\mathbf{x} - \mathbf{x}''\| \implies k_{\mathrm{in}}(\mathbf{x}, \mathbf{x}') = k_{\mathrm{in}}(\mathbf{x}, \mathbf{x}'').$$
$$\|\bar{h}_{\mathrm{NN}}(\mathbf{x}) - \bar{h}_{\mathrm{NN}}(\mathbf{x}')\| \neq \|\bar{h}_{\mathrm{NN}}(\mathbf{x}) - \bar{h}_{\mathrm{NN}}(\mathbf{x}'')\| \implies k_{\mathrm{out}}(\mathbf{x}, \mathbf{x}') \neq k_{\mathrm{out}}(\mathbf{x}, \mathbf{x}'').$$
These are true for all hyperparameter settings of $k_{\mathrm{in}}$ or $k_{\mathrm{out}}$, since both are RBF kernels. So, there is no hyperparameter setting of $k_{\mathrm{in}}$ that yields $k_{\mathrm{in}}'(\mathbf{x}_1, \mathbf{x}_2) = k_{\mathrm{in}}(\mathbf{x}_1, \mathbf{x}_2) + k_{\mathrm{out}}(\mathbf{x}_1, \mathbf{x}_2) \ \forall \ \mathbf{x}_1, \mathbf{x}_2$. Similarly, there is no hyperparameter setting of $k_{\mathrm{out}}$ that yields $k_{\mathrm{out}}'(\mathbf{x}_1, \mathbf{x}_2) = k_{\mathrm{in}}(\mathbf{x}_1, \mathbf{x}_2) + k_{\mathrm{out}}(\mathbf{x}_1, \mathbf{x}_2) \ \forall \ \mathbf{x}_1, \mathbf{x}_2$. Since, neither the input nor output kernel alone can correctly specify the kernel over all sets of positive measure, their generalization error is greater than the Bayes error, which is achieved by the correct kernel (Sollich, 2002), i.e., $k_{\mathrm{in}}(\mathbf{x}_1, \mathbf{x}_2) + k_{\mathrm{out}}(\mathbf{x}_1, \mathbf{x}_2)$. $\square$

# B  BACKGROUND

This section reviews notation for Neural Networks, Gaussian Process, and its more efficient approximation, SVGP. The RIO method, introduced in Section 3 of the main paper, uses Gaussian Processes to estimate the uncertainty in neural network predictions and reduces their point-prediction errors.

## B.1   Neural Networks

Neural Networks (NNs) learn a nonlinear transformation from input to output space based on a number of training examples. Let $\mathcal{D} \subseteq \mathbb{R}^{d_{\text{in}}} \times \mathbb{R}^{d_{\text{out}}}$ denote the training dataset with size $n$, and $\mathcal{X} = \{\mathbf{x}_i : (\mathbf{x}_i, \mathbf{y}_i) \in \mathcal{D}, \mathbf{x}_i = [x_i^1, x_i^2, \ldots, x_i^{d_{\text{in}}}] \mid i = 1, 2, \ldots, n\}$ and $\mathcal{Y} = \{\mathbf{y}_i : (\mathbf{x}_i, \mathbf{y}_i) \in \mathcal{D}, \mathbf{y}_i = [y_i^1, y_i^2, \ldots, y_i^{d_{\text{out}}}] \mid i = 1, 2, \ldots, n\}$ denote the inputs and outputs (i.e., targets). A fully-connected feed-forward neural network with $L$ hidden layers of width $N_l$ (for layer $l = 1, 2, \ldots, L$) performs the following computations: Let $z_l^j$ denote the output value of $j$th node in $l$th hidden layer given input $\mathbf{x}_i$, then $z_l^j = \phi(\sum_{k=1}^{d_{\text{in}}} w_l^{j,k} x_i^k + b_l^j)$, for $l = 1$ and $z_l^j = \phi(\sum_{k=1}^{N_{l-1}} w_l^{j,k} z_{l-1}^k + b_l^j)$, for $l = 2, \ldots, L$, where $w_l^{j,k}$ denotes the weight on the connection from $k$th node in previous layer to $j$th node in $l$th hidden layer, $b_l^j$ denotes the bias of $j$th node in $l$th hidden layer, and $\phi$ is a nonlinear activation function. The output value of $j$th node in output layer is then given by $\hat{y}_i^j = \sum_{k=1}^{N_L} w_{\text{out}}^{j,k} z_L^k + b_{\text{out}}^j$, where $w_{\text{out}}^{j,k}$ denotes the weight on the connection from $k$th node in last hidden layer to $j$th node in output layer, and $b_{\text{out}}^j$ denotes the bias of $j$th node in output layer.

A gradient-based optimizer is usually used to learn the weights and bias given a pre-defined loss function, e.g., a squared loss function $\mathcal{L} = \frac{1}{n} \sum_{i=1}^n (\mathbf{y}_i - \hat{\mathbf{y}}_i)^2$. For a standard NN, the learned parameters are fixed, so the NN output $\hat{\mathbf{y}}_i$ is also a fixed point. For a Bayesian NN, a distribution of the parameters is learned, so the NN output is a distribution of $\hat{\mathbf{y}}_i$. However, a pretrained standard NN needs to be augmented, e.g., with a Gaussian Process, to achieve the same result.

## B.2   Gaussian Process

A Gaussian Process (GP) is a collection of random variables, such that any finite collection of these variables follows a joint multivariate Gaussian distribution (Rasmussen & Williams, 2006). Given a training dataset $\mathcal{X} = \{\mathbf{x}_i \mid i = 1, 2, \ldots, n\}$ and $\mathbf{y} = \{y_i = f(\mathbf{x}_i) + \epsilon \mid i = 1, 2, \ldots, n\}$, where $\epsilon$ denotes additive independent identically distributed Gaussian noise, the first step for GP is to fit itself to these training data assuming $\mathbf{y} \sim \mathcal{N}(0, \mathbf{K}(\mathcal{X}, \mathcal{X}) + \sigma_n^2 \mathbf{I})$, where $\mathcal{N}$ denotes a multivariate Gaussian distribution with mean 0 and covariance matrix $\mathbf{K}(\mathcal{X}, \mathcal{X}) + \sigma_n^2 \mathbf{I}$. $\mathbf{K}(\mathcal{X}, \mathcal{X})$ denotes the kernel-based covariance matrix at all pairs of training points with each entry $k_{i,j} = k(\mathbf{x}_i, \mathbf{x}_j)$, and $\sigma_n^2$ denotes the noise variance of observations. One commonly used kernel is the radial basis function (RBF) kernel, which is defined as $k(\mathbf{x}_i, \mathbf{x}_j) = \sigma_f^2 \exp(-\frac{1}{2l_f^2} \|\mathbf{x}_i - \mathbf{x}_j\|^2)$. The signal variance $\sigma_f^2$, length scale $l_f$ and noise variance $\sigma_n^2$ are trainable hyperparameters. The hyperparameters of the covariance function are optimized during the learning process to maximize the log marginal likelihood $\log p(\mathbf{y}|\mathcal{X})$.

After fitting phase, the GP is utilized to predict the distribution of label $y_*$ given a test point $\mathbf{x}_*$. This prediction is given by $y_*|\mathcal{X}, \mathbf{y}, \mathbf{x}_* \sim \mathcal{N}(\bar{y}_*, \text{var}(y_*))$ with $\bar{y}_* = \mathbf{k}_*^\top (\mathbf{K}(\mathcal{X}, \mathcal{X}) + \sigma_n^2 \mathbf{I})^{-1} \mathbf{y}$ and $\text{var}(y_*) = k(\mathbf{x}_*, \mathbf{x}_*) - \mathbf{k}_*^\top (\mathbf{K}(\mathcal{X}, \mathcal{X}) + \sigma_n^2 \mathbf{I})^{-1} \mathbf{k}_*$, where $\mathbf{k}_*$ denotes the vector of kernel-based covariances (i.e., $k(\mathbf{x}_*, \mathbf{x}_i)$) between $\mathbf{x}_*$ and all the training points, and $\mathbf{y}$ denotes the vector of all training labels. Unlike with NN, the uncertainty of the prediction of a GP is therefore explicitly quantified.

## B.3   SVGP

The main limitation of the standard GP, as defined above, is that it is excessively expensive in both computational and storage cost. For a dataset with $n$ data points, the inference of standard GP has time complexity $\mathcal{O}(n^3)$ and space complexity $\mathcal{O}(n^2)$. To circumvent this issue, sparse GP methods were developed to approximate the original GP by introducing inducing variables (Csató & Opper, 2002; Seeger et al., 2003; Quiñonero Candela & Rasmussen, 2005; Titsias, 2009). These approximation approaches lead to a computational complexity of $\mathcal{O}(nm^2)$ and space complexity of $\mathcal{O}(nm)$, where $m$ is the number of inducing variables. Following this line of work, SVGP (Hensman et al., 2013; 2015) further improves the scalability of the approach by applying Stochastic Variational Inference (SVI) technique, as follows:

Consider the same training dataset and GP as in section B.2, and assume a set of inducing variables as $\mathcal{Z} = \{\mathbf{z}_i \mid i = 1, 2, \ldots, m\}$ and $\mathcal{U} = \{u_i = f(\mathbf{z}_i) + \epsilon \mid i = 1, 2, \ldots, m\}$ ($f(\cdot)$ and $\epsilon$ are unknown).

---

**Algorithm 1** Procedure of RIO

---

**Require:**
$(\mathcal{X}, \mathbf{y}) = \{(\mathbf{x}_i, y_i)\}_{i=1}^n$: training data
$\hat{\mathbf{y}} = \{\hat{y}_i\}_{i=1}^n$: NN predictions on training data
$\mathbf{x}_*$: data to be predicted
$\hat{y}_*$: NN prediction on $\mathbf{x}_*$

**Ensure:**
$\hat{y}'_* \sim \mathcal{N}(\hat{y}_* + \bar{\hat{r}}_*, \mathrm{var}(\hat{r}_*))$: a distribution of calibrated prediction

**Training Phase:**
1: calculate residuals $\mathbf{r} = \{r_i = y_i - \hat{y}_i\}_{i=1}^n$
2: **for** each optimizer step **do**
3:     calculate covariance matrix $\mathbf{K}_c((\mathcal{X}, \hat{\mathbf{y}}), (\mathcal{X}, \hat{\mathbf{y}}))$, where each entry is given by $k_c((\mathbf{x}_i, \hat{y}_i), (\mathbf{x}_j, \hat{y}_j)) = k_{\mathrm{in}}(\mathbf{x}_i, \mathbf{x}_j) + k_{\mathrm{out}}(\hat{y}_i, \hat{y}_j),$ for $i, j = 1, 2, \ldots, n$
4:     optimize GP hyperparameters by maximizing log marginal likelihood $\log p(\mathbf{r}|\mathcal{X}, \hat{\mathbf{y}}) = -\frac{1}{2}\mathbf{r}^\top(\mathbf{K}_c((\mathcal{X}, \hat{\mathbf{y}}), (\mathcal{X}, \hat{\mathbf{y}})) + \sigma_n^2\mathbf{I})\mathbf{r} - \frac{1}{2}\log|\mathbf{K}_c((\mathcal{X}, \hat{\mathbf{y}}), (\mathcal{X}, \hat{\mathbf{y}})) + \sigma_n^2\mathbf{I}| - \frac{n}{2}\log 2\pi$

**Deployment Phase:**
5: calculate residual mean $\bar{\hat{r}}_* = \mathbf{k}_*^\top(\mathbf{K}_c((\mathcal{X}, \hat{\mathbf{y}}), (\mathcal{X}, \hat{\mathbf{y}})) + \sigma_n^2\mathbf{I})^{-1}\mathbf{r}$ and residual variance $\mathrm{var}(\hat{r}_*) = k_c((\mathbf{x}_*, \hat{y}_*), (\mathbf{x}_*, \hat{y}_*)) - \mathbf{k}_*^\top(\mathbf{K}_c((\mathcal{X}, \hat{\mathbf{y}}), (\mathcal{X}, \hat{\mathbf{y}})) + \sigma_n^2\mathbf{I})^{-1}\mathbf{k}_*$
6: return distribution of calibrated prediction $\hat{y}'_* \sim \mathcal{N}(\hat{y}_* + \hat{r}_*, \mathrm{var}(\hat{r}_*))$

---

SVGP learns a variational distribution $q(\mathcal{U})$ by maximizing a lower bound of $\log p(\mathbf{y}|\mathcal{X})$, where $\log p(\mathbf{y}|\mathcal{X}) = \log \int p(\mathbf{y}|\mathcal{U}, \mathcal{X})p(\mathcal{U})\mathrm{d}\mathcal{U}$ and $p(\cdot)$ denotes the probability density under original GP. Trainable hyperparameters during the learning process include values of $\mathbf{z}_i$ and hyperparameters of the covariance function of original GP. Given a test point $\mathbf{x}_*$, the predictive distribution is then given by $p(y_*|\mathbf{x}_*) = \int p(y_*|\mathcal{U}, \mathbf{x}_*)q(\mathcal{U})\mathrm{d}\mathcal{U}$, which still follows a Gaussian distribution. One advantage of SVGP is that minibatch training methods (Le et al., 2011) can be applied in case of very large dataset. Suppose the minibatch size is $m'$ and $m \ll m\prime$, then for each training step/iteration, the computational complexity is $\mathcal{O}(m'm^2)$, and the space complexity is $\mathcal{O}(m'm)$. For full details about SVGP, see Hensman et al. (2013). Since NNs typically are based on training with relatively large datasets, SVGP makes it practical to implement uncertainty estimates on NNs.

## C   Procedure of RIO

This section provides an algorithmic description of RIO (see Algorithm 1).

## D   Empirical Study

### D.1   Experimental Setups

**Dataset Description**   In total, 12 real-world regression datasets from UCI machine learning repository (Dua & Graff, 2017) are tested. Table 4 summarizes the basic information of these datasets. For all the datasets except MSD, 20% of the whole dataset is used as test dataset and 80% is used as training dataset, and this split is randomly generated in each independent run. For MSD, the first 463715 samples are used as training dataset and the last 51630 samples are used as testing dataset according to the provider's guideline. During the experiments, all the datasets except for MSD are tested for 100 independent runs, and MSD datasets are tested for 10 independent runs. For each independent run, the dataset is randomly split into training set, validation set, and test set (except for MSD, in which the dataset split is strictly predefined by the provider), and the same random dataset split are used by all the tested algorithms to ensure fair comparisons. All source codes for reproducing the experimental results are provided at: (https://github.com/leaf-ai/rio-paper).

**Parametric Setup for Algorithms**

- NN: For SC dataset, a fully connected feed-forward NN with 2 hidden layers, each with 128 hidden neurons, is used. For CT dataset, a fully connected feed-forward NN with 2 hidden layers, each with 256 hidden neurons, is used. For MSD dataset, a fully connected

Table 4: Summary of testing dataset

| abbreviation | full name in UCI ML repository | dataset size | dimension | note |
|---|---|---|---|---|
| yacht | Yacht Hydrodynamics Data Set | 308 | 6 | - |
| ENB/h | Energy efficiency | 768 | 8 | Heating Load as target |
| ENB/c | Energy efficiency | 768 | 8 | Cooling Load as target |
| airfoil | Airfoil Self-Noise | 1505 | 5 | - |
| CCS | Concrete Compressive Strength | 1030 | 8 | - |
| wine/r | Wine Quality | 1599 | 11 | only use winequality-red data |
| wine/w | Wine Quality | 4898 | 11 | only use winequality-white data |
| CCPP | Combined Cycle Power Plant | 9568 | 4 | - |
| CASP | Physicochemical Properties of Protein Tertiary Structure | 54730 | 9 | - |
| SC | Superconductivty Data | 21263 | 81 | - |
| CT | Relative location of CT slices on axial axis | 53500 | 384 | - |
| MSD | YearPredictionMSD | 515345 | 90 | train: first 463715, test: last 51630 |

feed-forward NN with 4 hidden layers, each with 64 hidden neurons, is used. For all the remaining datasets, a fully connected feed-forward NN with 2 hidden layers, each with 64 hidden neurons, is used. The inputs to the NN are normalized to have mean 0 and standard deviation 1. The activation function is ReLU for all the hidden layers. The maximum number of epochs for training is 1000. 20% of the training data is used as validation data, and the split is random at each independent run. An early stop is triggered if the loss on validation data has not be improved for 10 epochs. The optimizer is RMSprop with learning rate 0.001, and the loss function is mean squared error (MSE).

- RIO, RIO variants and SVGP (Hensman et al., 2013): SVGP is used as an approximator to original GP in RIO and all the RIO variants. For RIO, RIO variants and SVGP, the number of inducing points are 50 for all the experiments. RBF kernel is used for both input and output kernel. For RIO, RIO variants and SVGP, the signal variances and length scales of all the kernels plus the noise variance are the trainable hyperparameters. The optimizer is L-BFGS-B with default parameters as in Scipy.optimize documentation (https://docs.scipy.org/doc/scipy/reference/optimize.minimize-lbfgsb.html), and the maximum number of iterations is set as 1000. The training process runs until the L-BFGS-B optimizer decides to stop.

- NNGP (Lee et al., 2018): For NNGP kernel, the depth is 2, and the activation function is ReLU. $n_g = 101$, $n_v = 151$, and $n_c = 131$. Following the learning process in original paper, a grid search is performed to search for the best values of $\sigma_w^2$ and $\sigma_b^2$. Same as in the original paper, a grid of 30 points evenly spaced from 0.1 to 5.0 (for $\sigma_w^2$) and 30 points evenly spaced from 0 to 2.0 (for $\sigma_b^2$) was evaluated. The noise variance $\sigma_\epsilon^2$ is fixed as 0.01. The grid search process stops when Cholesky decomposition fails or all the 900 points are evaluated. The best values found during the grid search will be used in the experiments. No pre-computed lookup tables are utilized.

- ANP (Kim et al., 2019): The parametric setups of ANP are following the recommendations in the original paper. The attention type is multihead, the hidden size is 64, the max number of context points is 50, the context ratio is 0.8, the random kernel hyperparameters option is on. The size of latent encoder is $64 \times 64 \times 64 \times 64$, the number of latents is 64, the size of deterministic encoder is $64 \times 64 \times 64 \times 64$, the size of decoder is $64 \times 64 \times 64 \times 64 \times 2$, and the deterministic path option is on. Adam optimizer with learning rate $10^{-4}$ is used, and the maximum number of training iterations is 2000.

**Performance Metrics**

- To measure the point-prediction error, the Root Mean Square Error (RMSE) between the method predictions and true outcomes on test datasets are calculated for each independent experimental run. After that, the mean and standard deviations of these RMSEs are used to measure the performance of the algorithms.

- To quantitatively measure the quality of uncertainty estimation, average negative log predictive density (NLPD) (Quiñonero-Candela et al., 2006) is used to measure the quality of

uncertainty estimation. NLPD is given by

$$L = -\frac{1}{n} \sum_{i=1}^{n} \log p(\hat{\mathbf{y}}_i = \mathbf{y}_i | \mathbf{x}_i) \tag{10}$$

where $\hat{\mathbf{y}}_i$ indicates the prediction results, $\mathbf{x}_i$ is the input with true associated outcome $\mathbf{y}_i$, $p(\cdot)$ is the probability density function (PDF) of the returned distribution based on input $\mathbf{x}_i$.

- To investigate the behaviors of RIO variants during learning, the mean of estimated noise variance $\sigma_n^2$ over all the independent runs are calculated.

- To compare the computation time of the algorithms, the training time (wall clock time) of NN, RIO, all the ablated RIO variants, SVGP and ANP are averaged over all the independent runs as the computation time. It is notable that the computation time of all RIO variants does not include the training time of associated NN, because the NN is considered to be pretrained. For NNGP, the wall clock time for the grid search is used. In case that the grid search stops due to Cholesky decomposition failures, the computation time of NNGP will be estimated as the average running time of all the successful evaluations $\times$ 900, which is the supposed number of evaluations. All the algorithms are implemented using Tensorflow, and tested in the exactly same python environment. All the experiments are running on a machine with 16 Intel(R) Xeon(R) CPU E5-2623 v4@2.60GHz and 128GB memory.

## D.2 RESULTS ON ESTIMATED CONFIDENCE INTERVALS

Confidence interval (CI) is a useful tool to estimate the distribution of outcomes with explicit probabilities in real-world applications. In order to provide more insights for practitioners, the percentages of testing outcomes that are within the 95%/90%/68% CIs as estimated by each algorithm are calculated. Ideally, these percentages should be as close to the estimated confidence levels as possible, e.g., a perfect uncertainty estimator would have exactly 95% of testing outcomes within its estimated 95% CIs. In practice, when two models have similar quality of CI estimations, the more conservative one is favoured. Figure 4 and 5 show the distribution of the percentages that testing outcomes are within the estimated 95%/90%/68% CIs over all the independent runs for all the datasets and algorithms. Figure 6, 7 and 8 compare the distributions of coverage percentages for 1%-99% CIs for RIO and SVGP. The experimental data are extracted from the experiments that generate Table 1 in main paper. In Figure 4, 5, 6, 7 and 8, the box extends from the lower to upper quartile values of the data (each data point represents an independent experimental run), with a line at the median. The whiskers extend from the box to show the range of the data. Flier points are those past the end of the whiskers, indicating the outliers.

According to the results, RIO is able to provide reasonable CI estimations in most cases. However, using this metric to compare algorithms regarding uncertainty estimation quality may be noisy and misleading. A method that simply learns the distribution of the labels would perform well in CI coverage metric, but it cannot make any meaningful point-wise prediction. More detailed examples are given as follows.

For "CT" dataset, SVGP has an extremely high RMSE of around 52 while RIO variants only have RMSEs of 1. However, SVGP still shows acceptable 95% and 90% CI coverage, and even has over-confident coverage for 68% CI. After investigation, it was found that this happened only by chance, and was not due to the accurate CI estimations of SVGP. Since SVGP is not able to extract any useful information from the high-dimensional input space, it simply treated all outcomes as noise. As a result, SVGP shows a very large RMSE compared to other algorithms, and the mean of its predicted outcome distribution is always around 0. Since SVGP treats everything as noise, the estimated noise variance is very high, and the estimated 95% CI based on this noise variance is overly high and covers all the test outcomes in most cases. When the estimated 90% CI is evaluated, the large error in mean estimation and large error in noise variance estimation cancel mostly cancel each other out by chance, i.e., the estimated 90% CI is mistakenly shifted by erroneous mean then the overly wide noise variance fortunately covers slightly more than 90% test outcomes. The case of the estimated 68% CI is similar, but now the error in noise variance cannot fully cover the error in mean, so the coverage percentages are now below 68%, indicating over-confidence.

One interesting observation is that SVGP tends to be more "conservative" for high confidence levels (>90%), even in cases where they are "optimistic" for low confidence levels. After investigation,

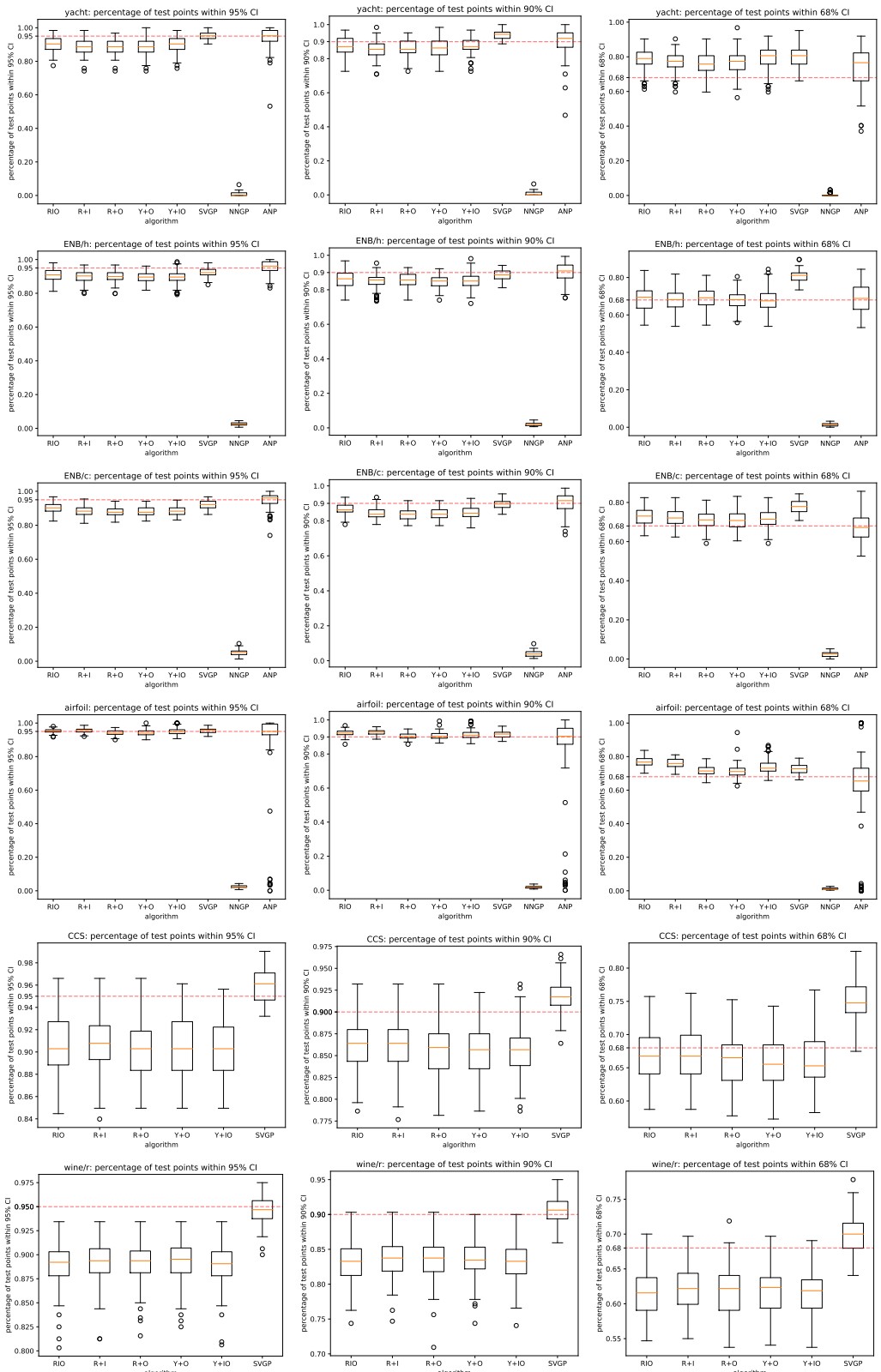

Figure 4: **Quality of estimated CIs.** These figures show the distribution of the percentages that testing outcomes are within the estimated 95%/90%/68% CIs over all the independent runs.

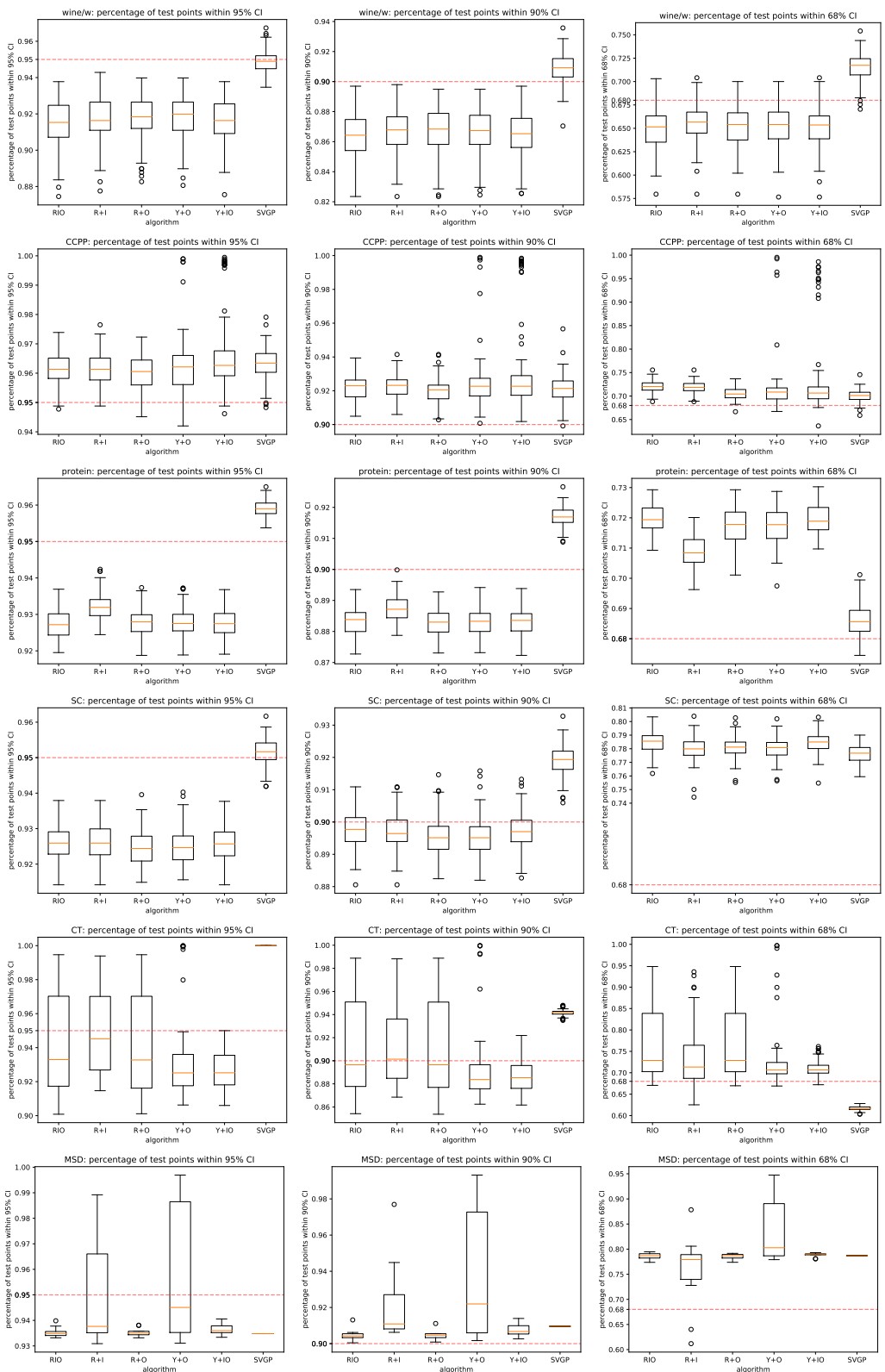

Figure 5: **Quality of estimated CIs.** These figures show the distribution of the percentages that testing outcomes are within the estimated 95%/90%/68% CIs over all the independent runs.

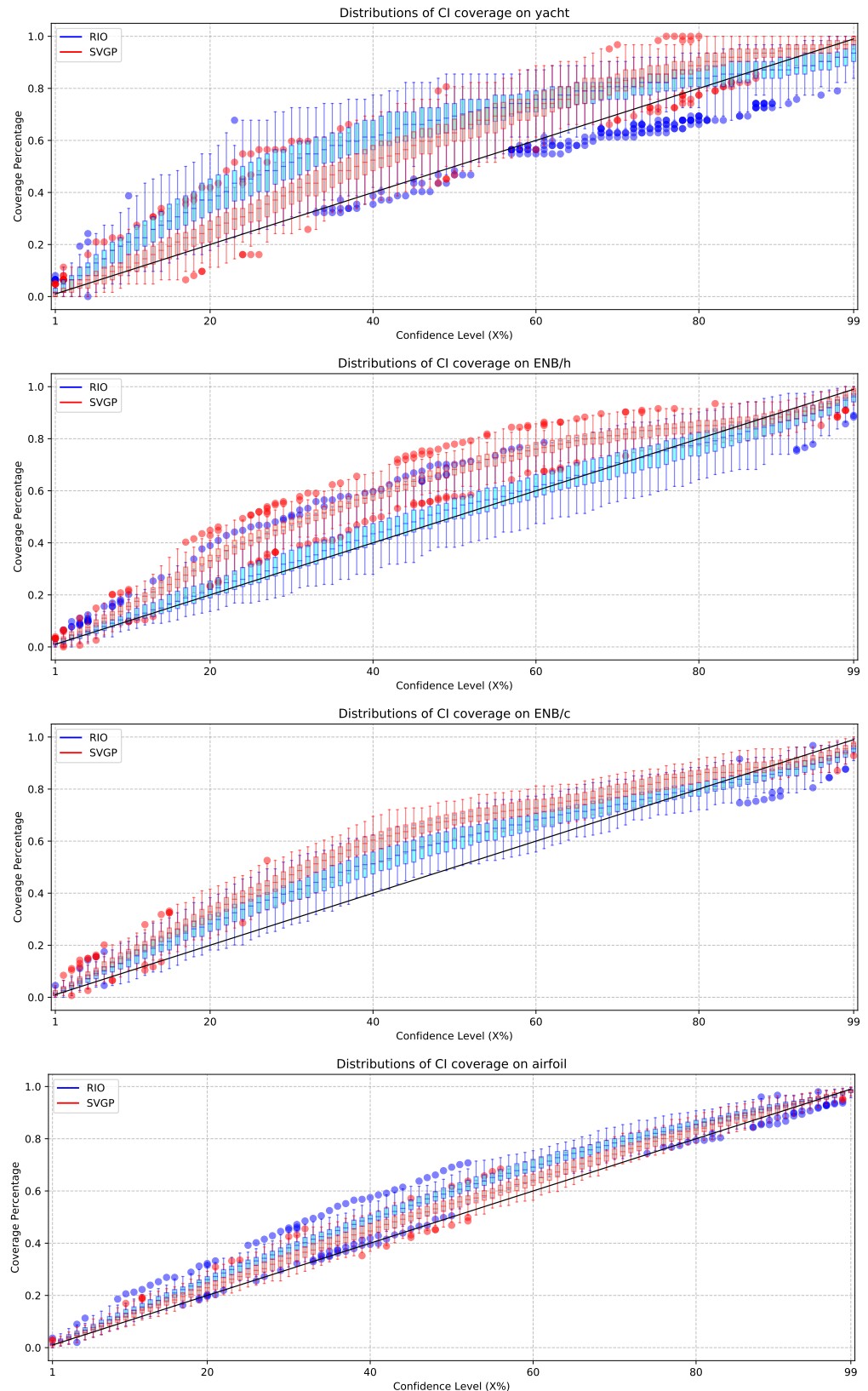

Figure 6: **Distribution of CI Coverage.** These figures compare the distributions of empirical coverage percentages of estimated 1%-99% CIs over all the independent runs for RIO and SVGP.

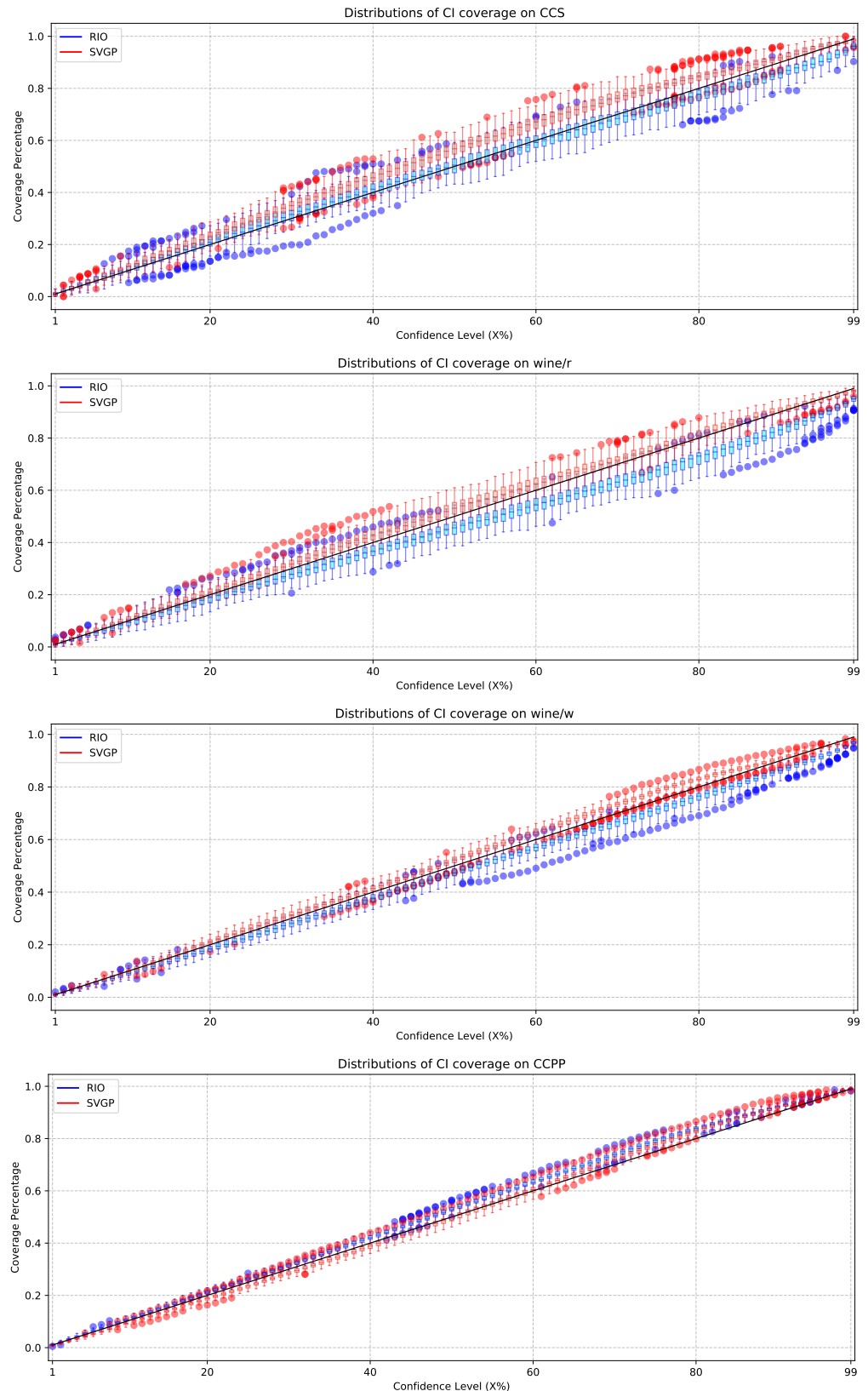

Figure 7: **Distribution of CI Coverage.** These figures compare the distributions of empirical coverage percentages of estimated 1%-99% CIs over all the independent runs for RIO and SVGP.

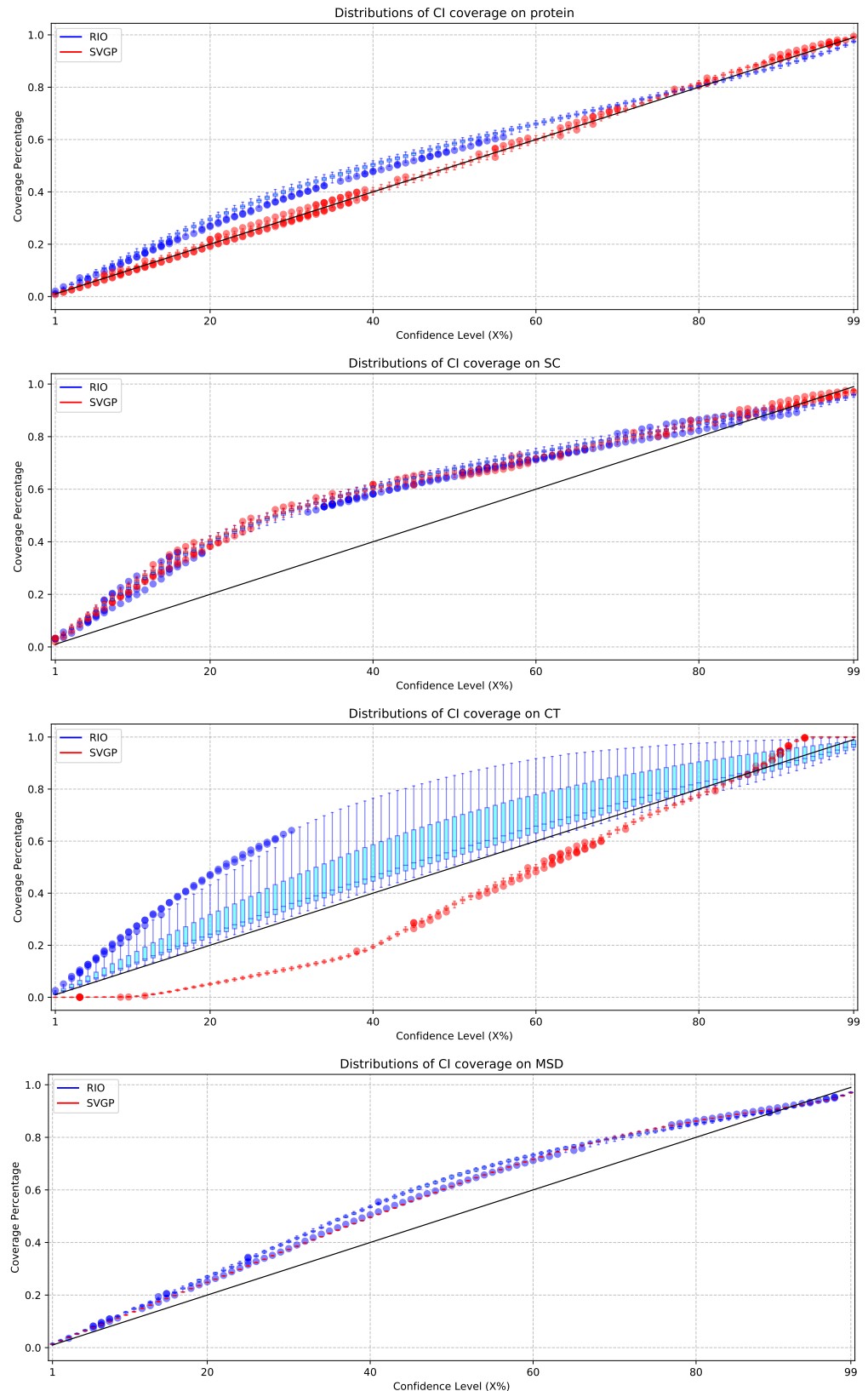

Figure 8: **Distribution of CI Coverage.** These figures compare the distributions of empirical coverage percentages of estimated 1%-99% CIs over all the independent runs for RIO and SVGP.

this is because SVGP normally has an overly high noise variance estimation (also comes with a higher prediction RMSE in most cases), so it has a higher chance of covering more points when the increase in CI width (influenced by noise variance) surpasses the erroneous shift of mean (depending on prediction errors).

### D.3 ADDITIONAL RESULTS ON OUTPUT CORRECTION

This section shows additional demonstrations of the detailed behaviors of RIO output correction. The experimental data are extracted from the experiments that generate Table 1 in main paper.

Figure 9 plots the distributions of ground truth labels (outcomes), original NN predictions and predictions corrected after RIO for a randomly picked run for each dataset. Figure 10 shows the point-wise comparisons between NN outputs and RIO-corrected outputs for the same experimental runs as in Figure 9. Based on the results, it is clear that RIO tends to calibrate each NN prediction accordingly. The distribution of outputs after RIO calibration may be a shift, or shrinkage, or expansion, or even more complex modifications of the original NN predictions, depending on how different are NN predictions from ground truths. As a result, the distribution of RIO calibrated outputs are closer to the distribution of the ground truth. One interesting behavior can be observed in Figure 9 for "protein" dataset: after applying RIO, the range of whiskers shrunk and the outliers disappeared, but the box (indicating 25 to 75 percentile of the data) expanded. This behavior shows that RIO can customize its calibration to each point. Another interesting behavior is that for "wine/r" dataset (see both Figure 9 and 10), RIO shifts all the original NN outputs to lower values, which are closer to the distribution of the ground truth.

To better study the behaviors of RIO, a new performance metric is defined, called improvement ratio (IR), which is the ratio between number of successful corrections (successfully reducing the prediction error) and total number of data points. For each run on each dataset, this IR value is calculated, and the distribution of IR values over 100 independent runs (random dataset split except for MSD, random NN initialization and training) on each dataset is plotted in Figure 11. According to the results, the IR values for RIO are above 0.5 in most cases. For 7 datasets, IR values are above 0.5 in all 100 independent runs. For some runs in yacht, ENB/h, CT, and MSD, the IR values are above 0.8 or even above 0.9. All these observations show that RIO is making meaningful corrections instead of random perturbations. Results in Figure 11 also provides useful information for practitioners: Although not all RIO calibrations improve the result, most of them do.

Same empirical analysis is also conducted for two RIO variants, namely R+I (predicting residuals with only input kernel) and R+O (predicting residuals with only output kernel). Figure 12, 13 and 14 show results for R+I, and Figure 15, 16 and 17 show results for R+O. From the results, the output kernel is helpful in problems where input kernel does not work well (CT and MSD), and it also shows more robust performance in terms of improvement ratio (IR) in most datasets. However it is still generally worse than full RIO. More specifically, R+I shows an extremely low IR in CT dataset (see Figure 14). After investigation, this is because the input kernel itself is not able to learn anything from the complex high-dimensional input space, so it treats everything as noise. As a result, it keeps the NN output unchanged during correction in most cases. Applying output kernel instead solves the issue. After comparing Figure 10, 13 and 16, it can be observed that the behaviors of RIO are either a mixture or selection between R+I and R+O. This means RIO with I/O kernel is able to choose the best kernel among these two or combines both if needed.

### D.4 EXPERIMENTAL RESULTS WITH ARD

The experimental results shown in the main paper is based on the setup that all RIO variants are using standard RBF kernel without Automatic Relevance Determination (ARD). To investigate the performance of RIO under more sophisticated setups, same experiments are run for all RIO variants with ARD mechanism turned on (all other experimental setups are identical with section D.1, except that the NN depth for MSD dataset is reduced from 4 hidden layers to 2, due to computation failure issue during Cholesky decomposition). Table 5 shows the summarized experimental results. From the results, RIO still performs the best or equals the best method (based on statistical tests) in 9 out of 12 datasets for both RMSE and NLPD metrics. This clearly shows the effectiveness and robustness of RIO.

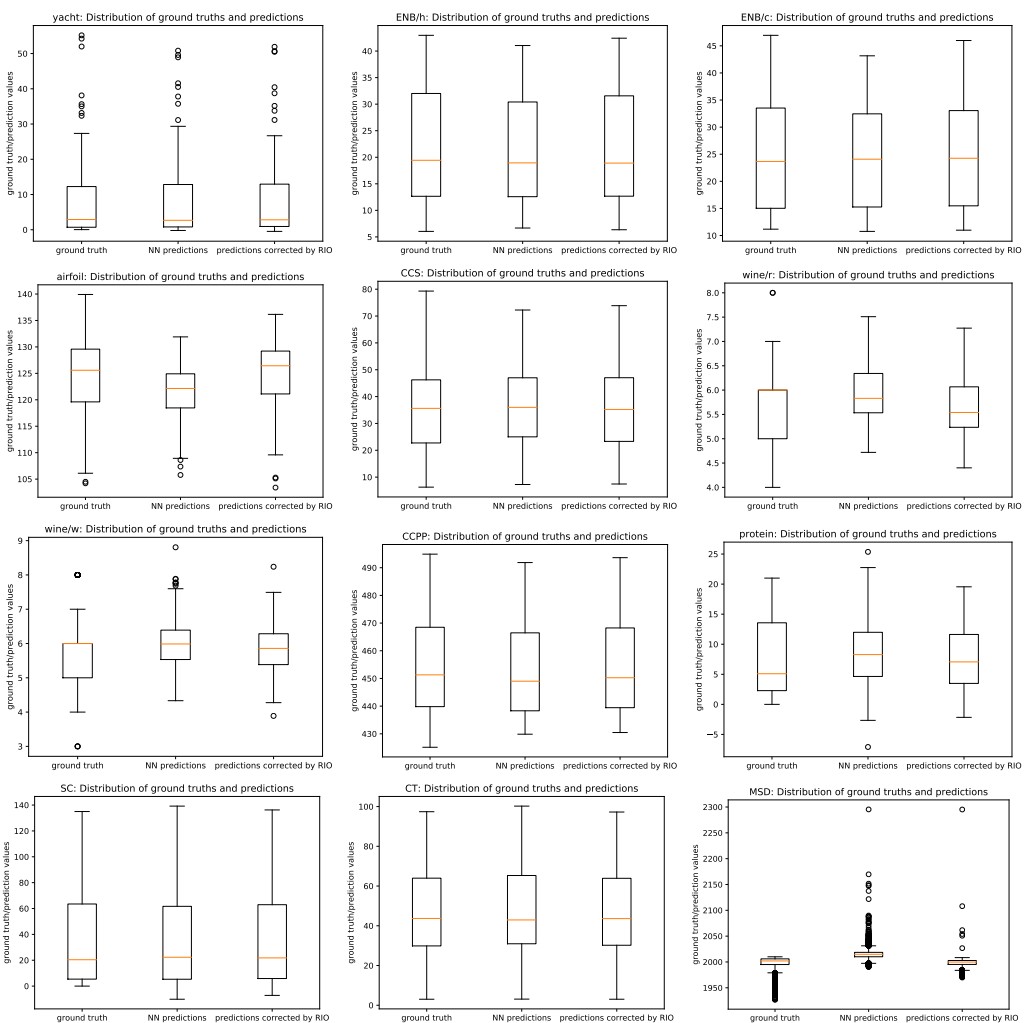

Figure 9: **Distribution of Ground Truths and NN/RIO-corrected predictions.** The box extends from the lower to upper quartile values of the data points, with a line at the median. The whiskers extend from the box to show the range of the data. Flier points are those past the end of the whiskers, indicating the outliers.

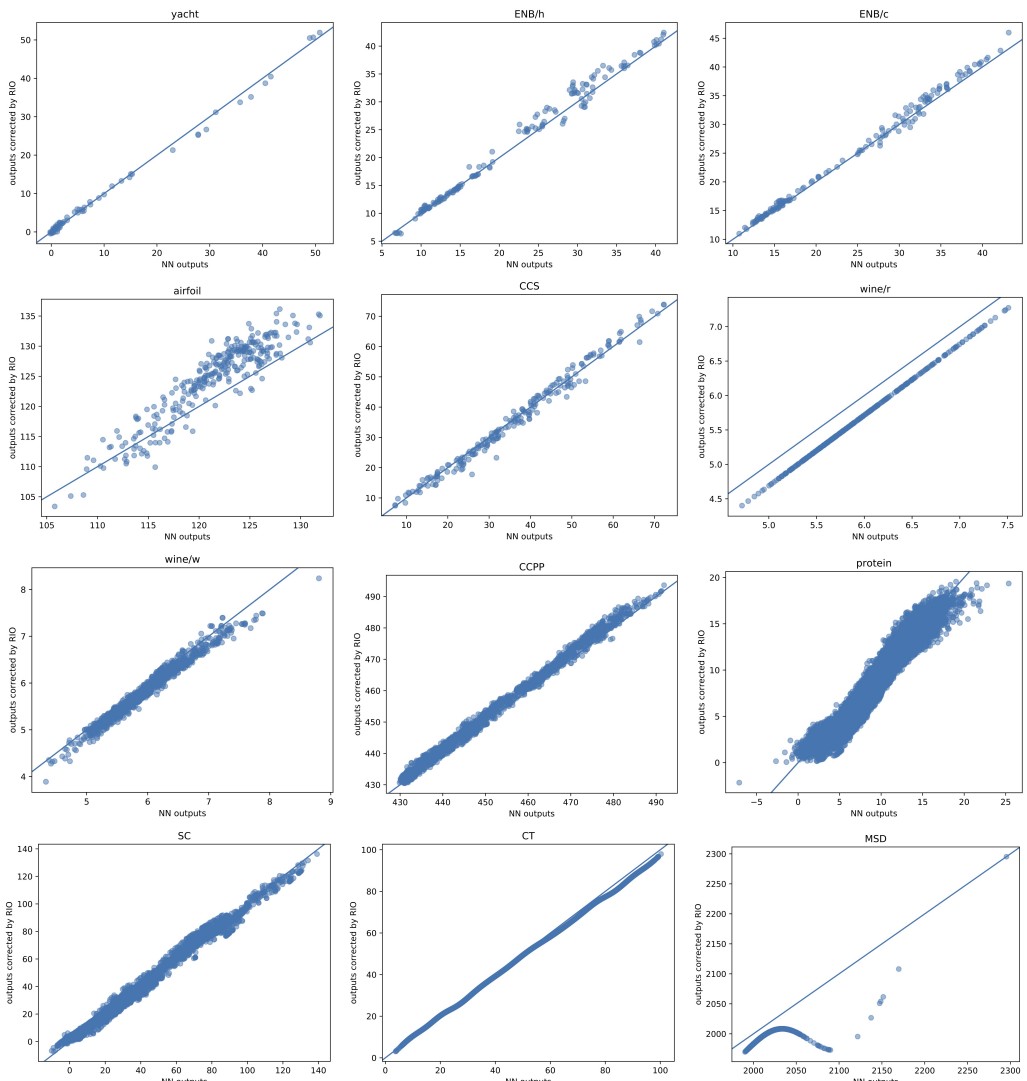

Figure 10: **Comparison between NN and RIO-corrected outputs.** Each dot represents a data point. The horizontal axis denote the original NN predictions, and the vertical axis the corresponding predictions after RIO corrections. The solid line indicates where NN predictions are same as RIO-corrected predictions (i.e., no change in output). This figure shows that RIO exhibits diverse behavior in how it calibrates output.

Table 5: Summary of experimental results (with ARD)

| Dataset n × d | Method | RMSE mean±std | NLPD mean±std | Noise Variance | Time (sec) | Dataset n × d | Method | RMSE mean±std | NLPD mean±std | Noise Variance | Time (sec) |
|---|---|---|---|---|---|---|---|---|---|---|---|
| yacht | NN | 2.35±1.16†‡ | - | - | 3.41 | ENB/h | NN | 1.01±0.38†‡ | - | - | 8.47 |
|  | RIO | 1.22±0.44 | 1.718±0.531 | 0.46 | 7.73 |  | RIO | **0.59±0.10** | **0.937±0.186** | 0.22 | 11.98 |
|  | R+I | 1.36±0.39†‡ | 1.893±0.482†‡ | 0.83 | 6.89 |  | R+I | 0.61±0.12†‡ | 0.971±0.197†‡ | 0.25 | 11.07 |
| 308 | R+O | 1.88±0.65†‡ | 2.208±0.470†‡ | 1.62 | 6.01 | 768 | R+O | 0.77±0.32†‡ | 1.150±0.328†‡ | 0.49 | 10.51 |
| × | Y+O | 1.88±0.62†‡ | 2.227±0.439†‡ | 1.82 | 9.56 | × | Y+O | 0.83±0.37†‡ | 1.226±0.345†‡ | 0.61 | 15.31 |
| 6 | Y+IO | 1.09±0.30†‡ | 1.591±0.425†‡ | 0.84 | 9.85 | 8 | Y+IO | 0.61±0.08†‡ | 0.974±0.164†‡ | 0.25 | 15.96 |
|  | SVGP | **0.91±0.21**†‡ | **1.359±0.208**†‡ | 0.77 | 8.93 |  | SVGP | 0.77±0.05†‡ | 1.156±0.049†‡ | 0.59 | 14.21 |
|  | NNGP | 12.40±1.45†‡ | 35.18±0.534†‡ | - | 7347 |  | NNGP | 4.97±0.29†‡ | 32.40±0.638†‡ | - | 7374 |
|  | ANP | 7.59±3.20†‡ | 1.793±0.887†‡ | - | 40.82 |  | ANP | 4.08±2.27†‡ | 2.475±0.559†‡ | - | 102.3 |
| ENB/c | NN | 1.88±0.46†‡ | - | - | 11.64 | airfoil | NN | 4.84±0.74†‡ | - | - | 8.83 |
|  | RIO | **1.42±0.18** | **1.825±0.155** | 1.31 | 15.27 |  | RIO | **2.49±0.15** | **2.349±0.048** | 6.56 | 17.92 |
|  | R+I | 1.45±0.17†‡ | 1.837±0.134‡ | 1.45 | 14.02 |  | R+I | 2.56±0.16†‡ | 2.376±0.054†‡ | 6.92 | 16.64 |
| 768 | R+O | 1.77±0.46†‡ | 2.021±0.232†‡ | 2.34 | 9.73 | 1505 | R+O | 4.14±0.27†‡ | 2.844±0.071†‡ | 16.31 | 10.97 |
| × | Y+O | 1.77±0.45†‡ | 2.020±0.231†‡ | 2.56 | 20.54 | × | Y+O | 4.39±1.64†‡ | 2.889±0.196†‡ | 20.08 | 22.90 |
| 8 | Y+IO | 1.46±0.22†‡ | 1.847±0.157†‡ | 1.42 | 21.65 | 5 | Y+IO | 3.73±0.35†‡ | 2.746±0.127†‡ | 14.88 | 25.08 |
|  | SVGP | 1.73±0.25†‡ | 1.969±0.130†‡ | 3.12 | 18.73 |  | SVGP | 3.36±0.23†‡ | 2.632±0.065†‡ | 11.11 | 21.80 |
|  | NNGP | 4.91±0.32†‡ | 30.14±0.886†‡ | - | 7704 |  | NNGP | 6.54±0.23†‡ | 33.60±0.420†‡ | - | 3355 |
|  | ANP | 4.81±2.15†‡ | 2.698±0.548†‡ | - | 64.11 |  | ANP | 21.17±30.72†‡ | 5.399±6.316†‡ | - | 231.7 |
| CCS | NN | 6.29±0.54†‡ | - | - | 6.53 | wine/r | NN | 0.689±0.037†‡ | - | - | 3.24 |
|  | RIO | 5.81±0.50 | **3.210±0.119** | 23.00 | 9.20 |  | RIO | 0.674±0.033 | 1.091±0.082 | 0.28 | 7.55 |
| 1030 | R+I | **5.80±0.49** | **3.206±0.118** | 23.19 | 8.80 | 1599 | R+I | 0.670±0.033†‡ | 1.085±0.084†‡ | 0.28 | 9.52 |
| × | R+O | 6.21±0.53†‡ | 3.286±0.121†‡ | 27.02 | 3.77 | × | R+O | 0.676±0.034†‡ | 1.093±0.083 | 0.29 | 5.18 |
| 8 | Y+O | 6.18±0.51†‡ | 3.278±0.114†‡ | 27.22 | 11.74 | 11 | Y+O | 0.675±0.033 | 1.088±0.081‡ | 0.29 | 13.56 |
|  | Y+IO | 5.91±0.46†‡ | 3.228±0.108†‡ | 24.50 | 12.09 |  | Y+IO | 0.672±0.033†‡ | 1.085±0.081†‡ | 0.29 | 14.23 |
|  | SVGP | 6.20±0.40†‡ | 3.233±0.060†‡ | 34.66 | 11.34 |  | SVGP | **0.640±0.028**†‡ | **0.969±0.043**†‡ | 0.39 | 13.32 |
| wine/w | NN | 0.725±0.026†‡ | - | - | 7.01 | CCPP | NN | 4.97±0.53†‡ | - | - | 10.11 |
|  | RIO | 0.707±0.017 | 1.093±0.035 | 0.38 | 11.8 |  | RIO | **3.99±0.13** | **2.796±0.026** | 15.82 | 26.27 |
| 4898 | R+I | **0.702±0.018**†‡ | 1.084±0.035†‡ | 0.38 | 14.76 | 9568 | R+I | **3.99±0.13** | 2.797±0.025†‡ | 15.87 | 24.47 |
| × | R+O | 0.711±0.019†‡ | 1.098±0.037†‡ | 0.39 | 6.46 | × | R+O | 4.33±0.13†‡ | 2.879±0.027†‡ | 18.51 | 9.99 |
| 11 | Y+O | 0.710±0.019†‡ | 1.096±0.037†‡ | 0.39 | 19.77 | 4 | Y+O | 8.94±44.78†‡ | 2.974±0.484†‡ | 2095 | 27.24 |
|  | Y+IO | 0.708±0.018† | 1.093±0.036 | 0.39 | 20.15 |  | Y+IO | 4.57±0.97†‡ | 2.968±0.255†‡ | 36.48 | 27.65 |
|  | SVGP | 0.714±0.017†‡ | **1.074±0.022**†‡ | 0.50 | 19.42 |  | SVGP | 8.80±44.83‡ | 2.917±0.468†‡ | 1455 | 26.61 |
| protein | NN | 4.23±0.08†‡ | - | - | 147.4 | SC | NN | 12.41±0.84†‡ | - | - | 73.11 |
|  | RIO | **4.08±0.05** | **2.826±0.013** | 15.75 | 149.6 |  | RIO | **11.24±0.33** | **3.844±0.030** | 104.6 | 99.91 |
| 45730 | R+I | 4.11±0.05†‡ | 2.834±0.013†‡ | 16.01 | 130.9 | 21263 | R+I | 11.27±0.32†‡ | 3.847±0.029†‡ | 105.5 | 95.33 |
| × | R+O | 4.15±0.06†‡ | 2.843±0.015†‡ | 16.31 | 98.88 | × | R+O | 11.68±0.42†‡ | 3.881±0.037†‡ | 113.9 | 47.61 |
| 9 | Y+O | 4.15±0.06†‡ | 2.843±0.015†‡ | 16.31 | 120.4 | 81 | Y+O | 11.68±0.42†‡ | 3.882±0.036†‡ | 114.2 | 72.08 |
|  | Y+IO | **4.08±0.05** | **2.826±0.013** | 15.75 | 148.6 |  | Y+IO | 11.29±0.38†‡ | 3.848±0.034†‡ | 105.9 | 103.4 |
|  | SVGP | 4.64±0.04†‡ | 2.955±0.008†‡ | 22.15 | 128.1 |  | SVGP | 14.12±0.27†‡ | 4.090±0.015†‡ | 217.8 | 95.78 |
| CT | NN | 1.12±0.29†‡ | - | - | 196.6 | MSD | NN | 12.54±1.16†‡ | - | - | 777.8 |
|  | RIO | **0.86±0.12** | **1.261±0.202** | 0.95 | 519.6 |  | RIO | **9.79±0.22** | **3.698±0.023** | 94.90 | 3059 |
| 53500 | R+I | 1.12±0.29†‡ | 1.505±0.260†‡ | 1.55 | 20.03 | 515345 | R+I | 12.54±1.16†‡ | 3.943±0.092†‡ | 156.6 | 91.01 |
| × | R+O | **0.86±0.12** | **1.261±0.202** | 0.95 | 159.4 | × | R+O | **9.82±0.19** | **3.701±0.020** | 96.11 | 2739 |
| 384 | Y+O | 0.98±0.77‡ | **1.303±0.336** | 2.02 | 180.6 | 90 | Y+O | 209.8±596.2‡ | 7.010±9.680‡ | 236.4 | 1491 |
|  | Y+IO | 0.88±0.13†‡ | **1.256±0.145** | 0.55 | 594.5 |  | Y+IO | 208.8±596.6‡ | 7.055±9.664‡ | 197.2 | 3347 |
|  | SVGP | 52.07±0.19†‡ | 5.372±0.004†‡ | 2712 | 29.69 |  | SVGP | 10.97±0.0†‡ | 3.981±0.0†‡ | 199.7 | 2590 |

The symbols † and ‡ indicate that the difference between the marked entry and RIO is statistically significant at the 5% significance level using paired $t$-test and Wilcoxon test, respectively. The best entries that are significantly better than all the others under at least one statistical test are marked in boldface (ties are allowed).

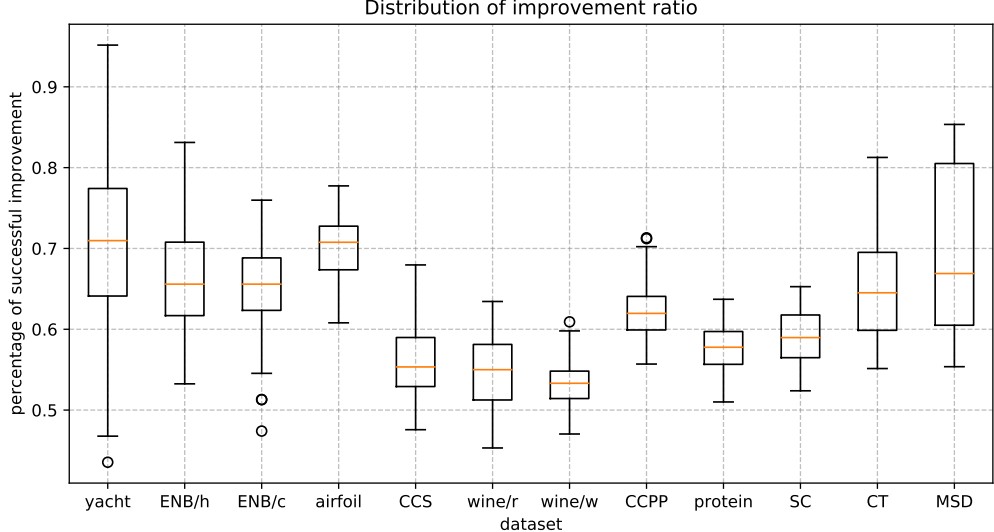

Figure 11: **Distribution of Impovement Ratio.** The box extends from the lower to upper quartile values of the data (each data point represents an independent experimental run), with a line at the median. The whiskers extend from the box to show the range of the data. Flier points are those past the end of the whiskers, indicating the outliers.

### D.5 EXPERIMENTAL RESULTS WITH 10,000 MAXIMUM OPTIMIZER IERATIONS

The experimental results shown in the main paper is based on the setup that all RIO variants are using L-BFGS-B optimizer with a maximun number of iterations as 1,000. To investigate the performance of RIO under larger computational budget, same experiments are run for all RIO variants with maximum number of optimizer iterations as 10,000 (all other experimental setups are identical with section D.1). Table 6 shows the summarized experimental results for 8 smallest datasets. According to the results, the rankings of the methods are very similar to those in Table 1 (in the main paper), and RIO still performs best in all metrics.

### D.6 RESULTS ON RANDOM FORESTS

In principle, RIO can be applied to any prediction model since it treats the pretrained model as a black box. To demonstrate this extensibility, RIO is applied to another classical prediction model — Random Forests (RF) (Ho, 1995; Breiman, 2001). The same experimental setups as in section D.1 are used. For RF, the number of estimators is set as 100 for all datasets. To avoid overfitting, the minimum number of samples required to be at a leaf node is 10 for all datasets, and the max depth of the tree is 7 for MSD and 5 for all other datasets. Table 7 shows the experimental results. From Table 7, RIO performs the best or equals the best method (based on statistical tests) in 9 out of 12 datasets in terms of both RMSE and NLPD. In addition, RIO significantly improves the performance of original RF in 11 out of 12 datasets. These results demonstrates the robustness and broad applicability of RIO.

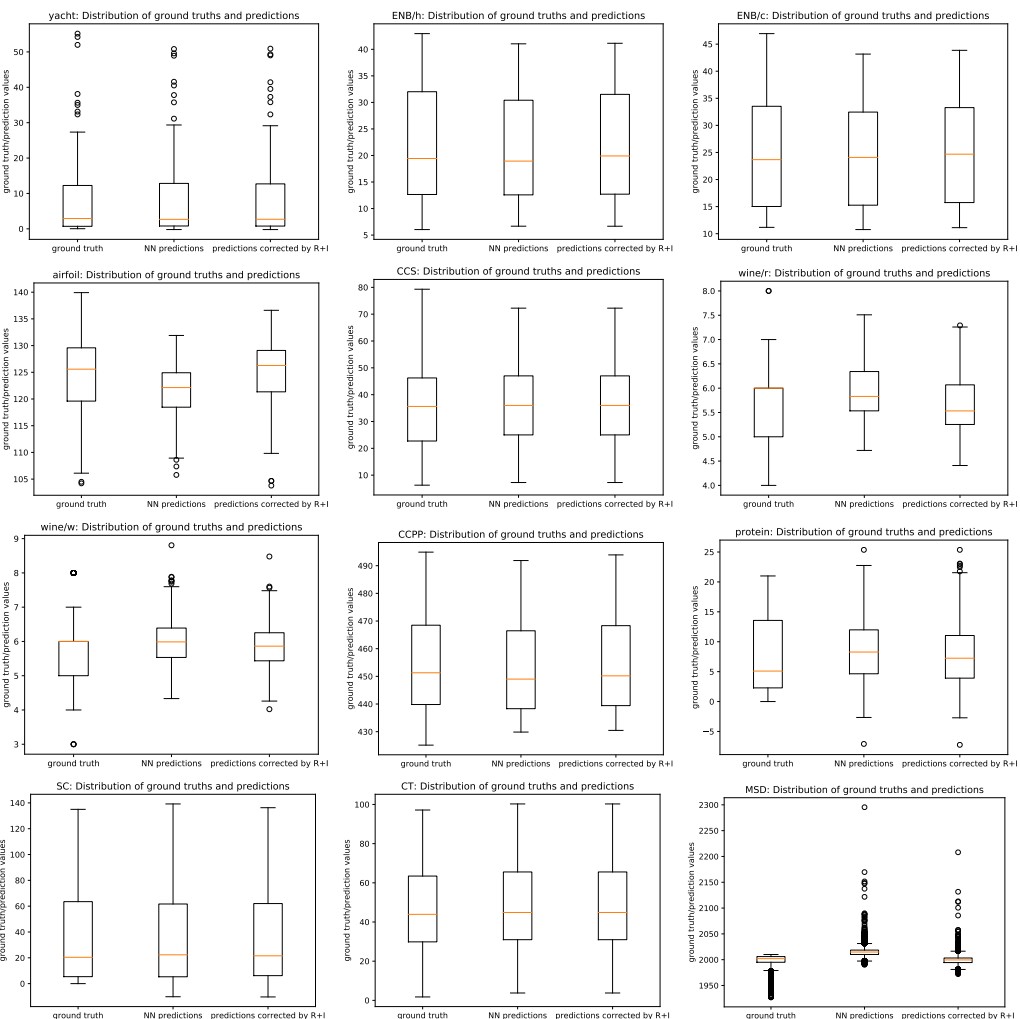

Figure 12: **Distribution of Ground Truths and NN/R+I-corrected predictions.** The box extends from the lower to upper quartile values of the data points, with a line at the median. The whiskers extend from the box to show the range of the data. Flier points are those past the end of the whiskers, indicating the outliers.

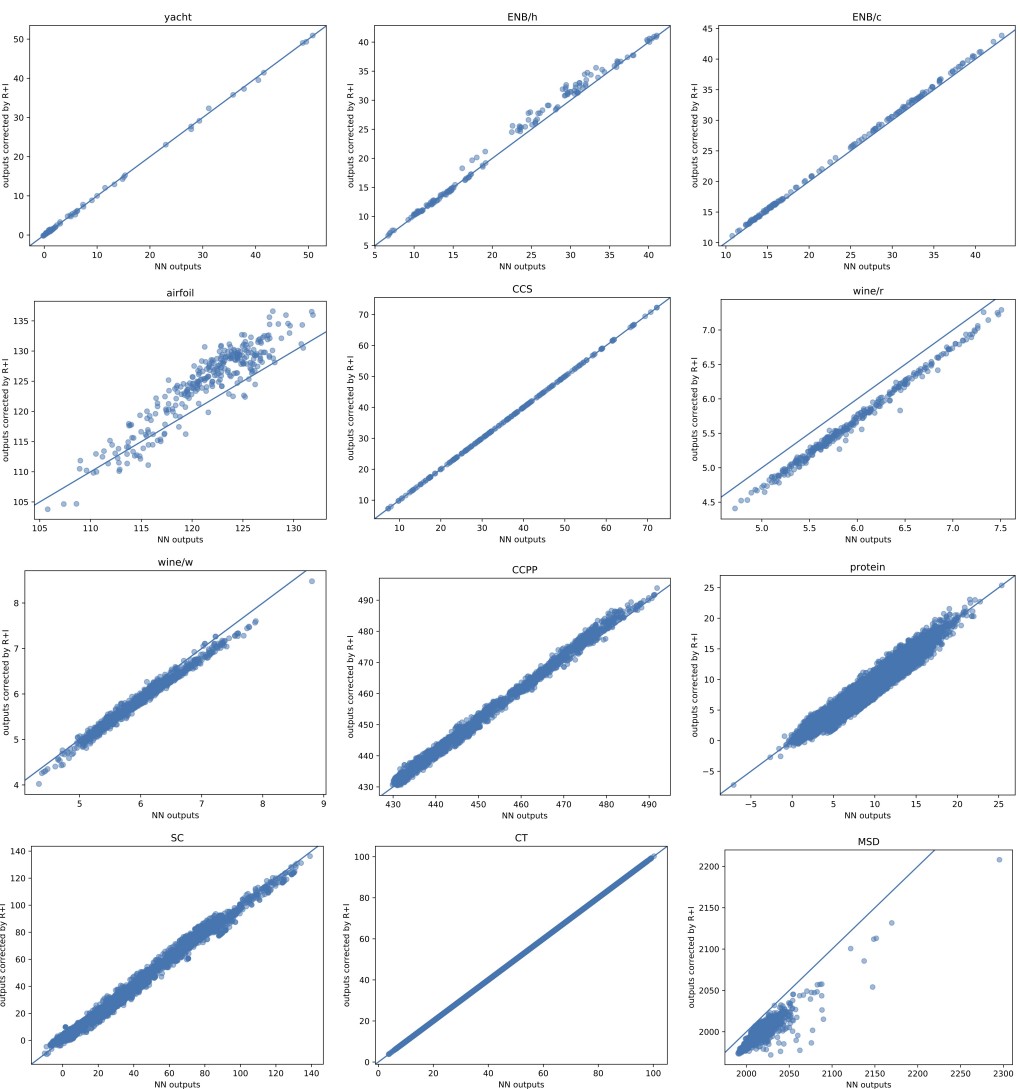

Figure 13: **Comparison between NN and R+I-corrected outputs.** Each dot represents a data point. The horizontal axis denote the original NN predictions, and the vertical axis the corresponding predictions after R+I corrections. The solid line indicates where NN predictions are same as R+I-corrected predictions (i.e., no change in output).

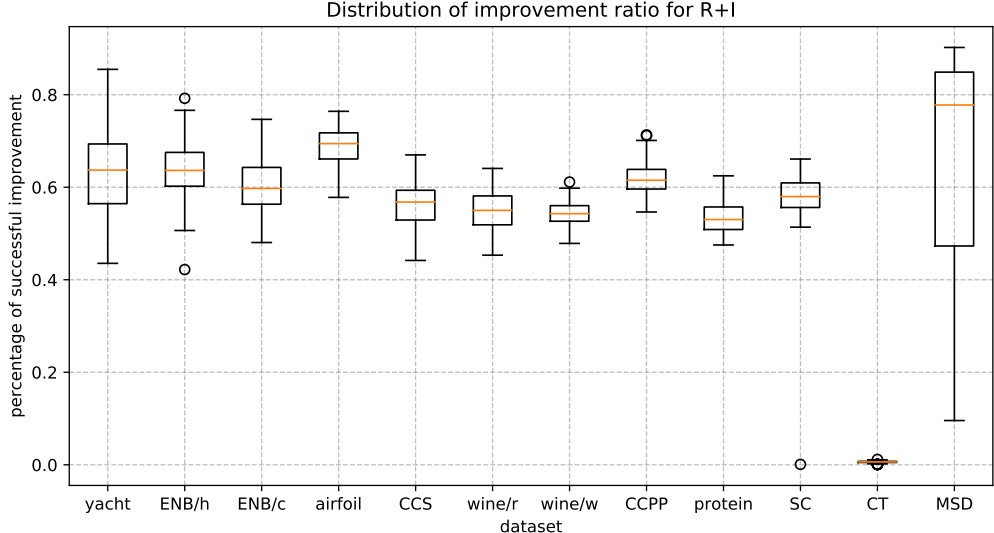

Figure 14: **Distribution of Impovement Ratio.** The box extends from the lower to upper quartile values of the data (each data point represents an independent experimental run), with a line at the median. The whiskers extend from the box to show the range of the data. Flier points are those past the end of the whiskers, indicating the outliers.

Table 6: Summary of experimental results (with 10,000 maximum optimizer iterations)

| Dataset n × d | Method | RMSE mean±std | NLPD mean±std | Noise Variance | Time (sec) | Dataset n × d | Method | RMSE mean±std | NLPD mean±std | Noise Variance | Time (sec) |
|---|---|---|---|---|---|---|---|---|---|---|---|
| yacht | NN | 2.20±0.93†‡ | - | - | 3.33 | ENB/h | NN | 0.94±0.37†‡ | - | - | 10.14 |
| | RIO | **1.40±0.50** | 1.883±0.568 | 0.74 | 25.67 | | RIO | **0.64±0.26** | **0.968±0.273** | 0.31 | 45.04 |
| | R+I | 1.93±0.65†‡ | 2.266±0.484†‡ | 2.19 | 5.59 | | R+I | 0.70±0.33†‡ | 1.043±0.317†‡ | 0.41 | 22.22 |
| 308 | R+O | 1.78±0.57†‡ | 2.176±0.525†‡ | 1.39 | 6.76 | 768 | R+O | 0.72±0.31†‡ | 1.084±0.309†‡ | 0.41 | 20.49 |
| × | Y+O | 1.78±0.56†‡ | 2.204±0.509†‡ | 1.60 | 23.99 | × | Y+O | 0.78±0.35†‡ | 1.163±0.328†‡ | 0.55 | 55.23 |
| 6 | Y+IO | **1.40±0.44** | 1.919±0.567 | 0.82 | 45.69 | 8 | Y+IO | 0.66±0.26†‡ | 1.013±0.280†‡ | 0.34 | 82.82 |
| | SVGP | 3.67±0.60†‡ | 2.689±0.111†‡ | 12.07 | 42.59 | | SVGP | 2.01±0.17†‡ | 2.145±0.071†‡ | 4.24 | 79.46 |
| | NNGP | 12.40±1.45†‡ | 35.18±0.534†‡ | - | 7374 | | NNGP | 4.97±0.29†‡ | 32.40±0.638†‡ | - | 7374 |
| | ANP | 7.59±3.20†‡ | **1.793±0.887†‡** | - | 40.82 | | ANP | 4.08±2.27†‡ | 2.475±0.559†‡ | - | 102.3 |
| ENB/c | NN | 1.87±0.42†‡ | - | - | 11.79 | airfoil | NN | 4.84±0.47†‡ | - | - | 8.96 |
| | RIO | **1.51±0.35** | **1.852±0.198** | 1.59 | 48.53 | | RIO | **3.06±0.20** | **2.551±0.058** | 9.44 | 104.0 |
| | R+I | 1.70±0.41†‡ | 1.98±0.21†‡ | 2.17 | 10.96 | | R+I | 3.13±0.21†‡ | 2.573±0.059†‡ | 9.91 | 73.22 |
| 768 | R+O | 1.75±0.41†‡ | 2.011±0.211†‡ | 2.21 | 10.85 | 1505 | R+O | 4.16±0.27†‡ | 2.848±0.068†‡ | 16.58 | 10.92 |
| × | Y+O | 1.75±0.41†‡ | 2.012±0.210†‡ | 2.27 | 39.52 | × | Y+O | 4.21±0.27†‡ | 2.862±0.082†‡ | 17.89 | 38.69 |
| 8 | Y+IO | 1.62±0.35†‡ | 1.936±0.197†‡ | 1.86 | 70.94 | 5 | Y+IO | 3.19±0.30†‡ | 2.583±0.087†‡ | 10.24 | 119.95 |
| | SVGP | 2.52±0.21†‡ | 2.363±0.072†‡ | 6.31 | 84.32 | | SVGP | 3.27±0.20†‡ | 2.608±0.056†‡ | 10.56 | 106.0 |
| | NNGP | 4.91±0.32†‡ | 30.14±0.886†‡ | - | 7704 | | NNGP | 6.54±0.23†‡ | 33.60±0.420†‡ | - | 3355 |
| | ANP | 4.81±2.15†‡ | 2.698±0.548†‡ | - | 64.11 | | ANP | 21.17±30.72†‡ | 5.399±6.316†‡ | - | 231.7 |
| CCS | NN | 6.25±0.49†‡ | - | - | 6.54 | wine/r | NN | 0.688±0.039†‡ | - | - | 3.26 |
| | RIO | **5.96±0.47** | **3.230±0.108** | 25.37 | 14.31 | | RIO | 0.671±0.033 | 1.088±0.087 | 0.28 | 20.12 |
| 1030 | R+I | 5.99±0.47†‡ | **3.235±0.107** | 26.04 | 4.91 | 1599 | R+I | 0.668±0.033†‡ | 1.080±0.085†‡ | 0.28 | 12.61 |
| × | R+O | 6.19±0.49†‡ | 3.280±0.112†‡ | 27.43 | 4.04 | × | R+O | 0.675±0.033†‡ | 1.094±0.088†‡ | 0.29 | 4.96 |
| 8 | Y+O | 6.18±0.48†‡ | 3.276±0.109†‡ | 27.65 | 17.55 | 11 | Y+O | 0.674±0.033†‡ | 1.089±0.086†‡ | 0.29 | 17.01 |
| | Y+IO | 6.03±0.47†‡ | 3.246±0.107†‡ | 26.24 | 41.89 | | Y+IO | 0.671±0.032 | 1.087±0.086 | 0.28 | 34.45 |
| | SVGP | 6.62±0.37†‡ | 3.297±0.045†‡ | 41.15 | 73.66 | | SVGP | **0.642±0.028†‡** | **0.973±0.042†‡** | 0.39 | 70.53 |
| wine/w | NN | 0.723±0.027†‡ | - | - | 8.51 | CCPP | NN | 4.94±0.49†‡ | - | - | 17.38 |
| | RIO | 0.704±0.018 | 1.088±0.034 | 0.38 | 49.96 | | RIO | **4.03±0.13** | **2.808±0.025** | 16.21 | 151.3 |
| 4898 | R+I | **0.700±0.017†‡** | 1.079±0.033†‡ | 0.38 | 26.43 | 9568 | R+I | 4.04±0.13†‡ | 2.810±0.026†‡ | 16.28 | 116.0 |
| × | R+O | 0.710±0.021†‡ | 1.095±0.037†‡ | 0.39 | 8.87 | × | R+O | 4.33±0.14†‡ | 2.880±0.029†‡ | 18.56 | 19.02 |
| 11 | Y+O | 0.710±0.020†‡ | 1.093±0.037†‡ | 0.39 | 29.97 | 4 | Y+O | 13.40±63.01†‡ | 3.012±0.663†‡ | 4161 | 100.8 |
| | Y+IO | 0.704±0.018‡ | 1.088±0.034 | 0.38 | 66.79 | | Y+IO | 4.71±1.51†‡ | 2.969±0.271†‡ | 33.58 | 267.1 |
| | SVGP | 0.713±0.016†‡ | **1.076±0.022†‡** | 0.5 | 158.1 | | SVGP | 4.25±0.13†‡ | 2.859±0.028†‡ | 17.94 | 334.6 |

The symbols † and ‡ indicate that the difference between the marked entry and RIO is statistically significant at the 5% significance level using paired $t$-test and Wilcoxon test, respectively. The best entries that are significantly better than all the others under at least one statistical test are marked in boldface (ties are allowed).

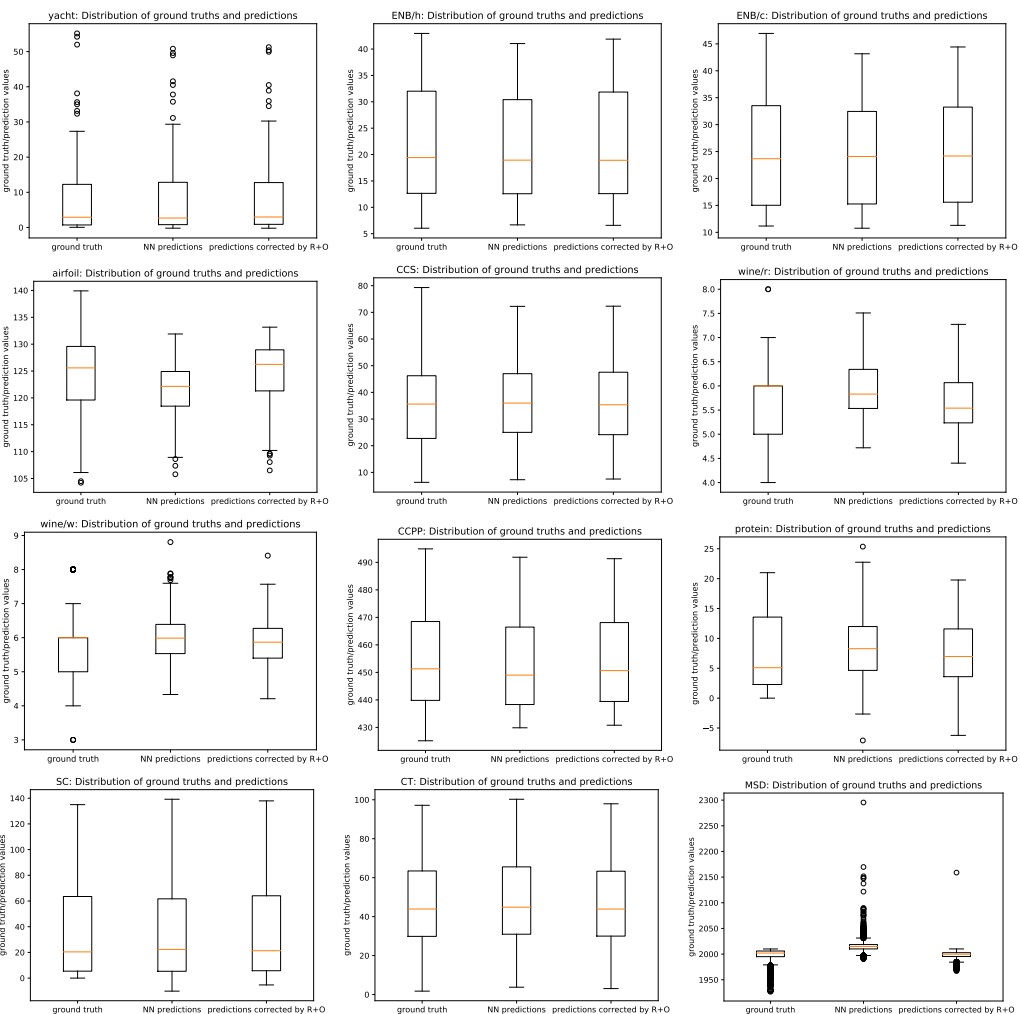

Figure 15: **Distribution of Ground Truths and NN/R+O-corrected predictions.** The box extends from the lower to upper quartile values of the data points, with a line at the median. The whiskers extend from the box to show the range of the data. Flier points are those past the end of the whiskers, indicating the outliers.

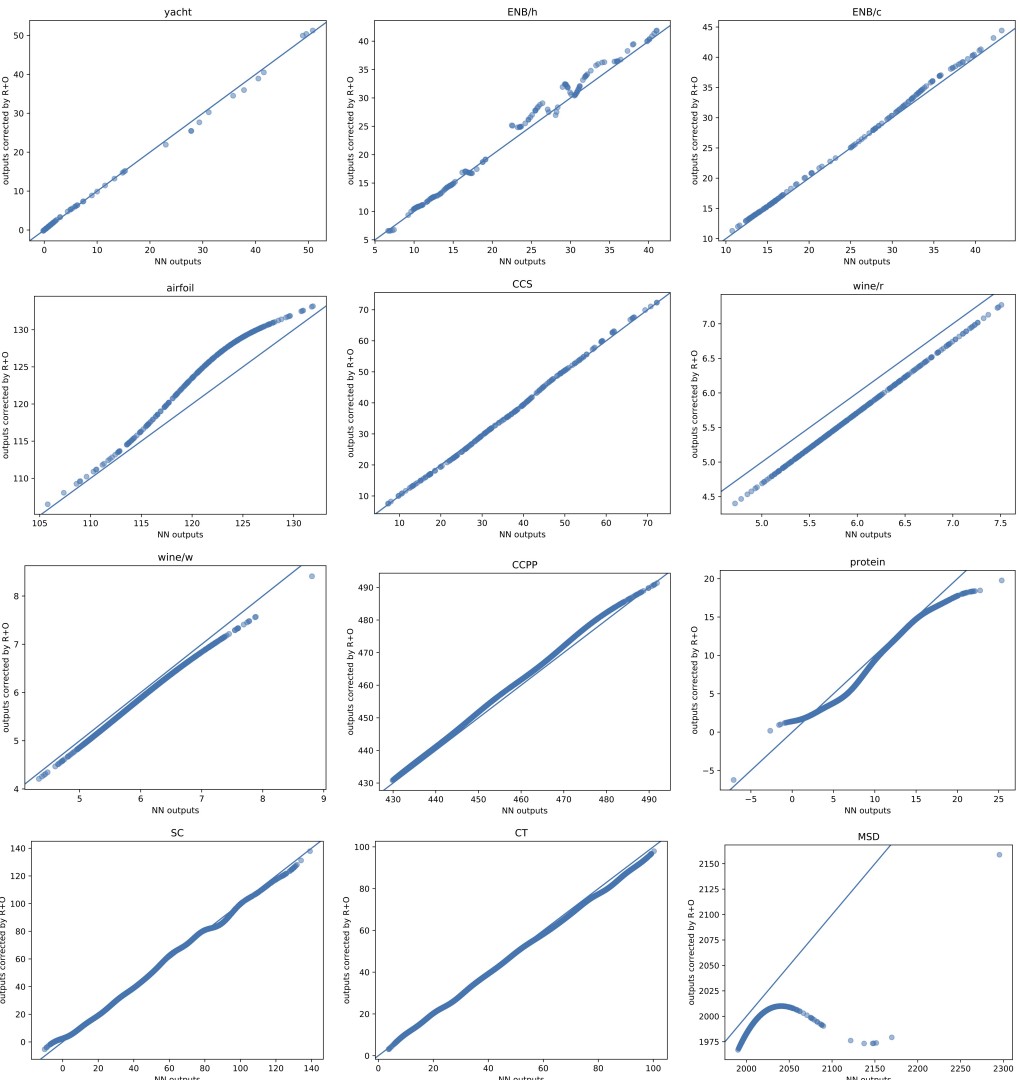

Figure 16: **Comparison between NN and R+O-corrected outputs.** Each dot represents a data point. The horizontal axis denote the original NN predictions, and the vertical axis the corresponding predictions after R+O corrections. The solid line indicates where NN predictions are same as R+O-corrected predictions (i.e., no change in output).

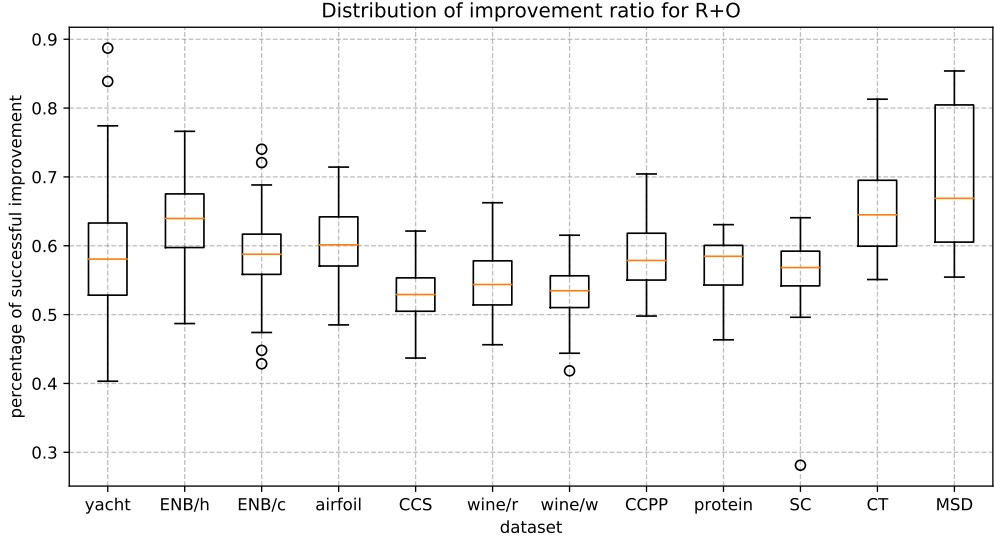

Figure 17: **Distribution of Impovement Ratio.** The box extends from the lower to upper quartile values of the data (each data point represents an independent experimental run), with a line at the median. The whiskers extend from the box to show the range of the data. Flier points are those past the end of the whiskers, indicating the outliers.

Table 7: Summary of experimental results with Random Forests (RF)

| Dataset n × d | Method | RMSE mean±std | NLPD mean±std | Noise Variance | Time (sec) | Dataset n × d | Method | RMSE mean±std | NLPD mean±std | Noise Variance | Time (sec) |
|---|---|---|---|---|---|---|---|---|---|---|---|
| yacht | RF | 1.95±0.59†‡ | - | - | 0.33 | ENB/h | RF | 1.44±0.16†‡ | - | - | 0.35 |
| | RIO | **0.80±0.18** | **1.177±0.187** | 0.49 | 14.93 | | RIO | **0.79±0.13** | **1.150±0.138** | 0.61 | 16.66 |
| | R+I | 1.62±0.67†‡ | 1.833±0.453†‡ | 1.91 | 12.41 | | R+I | 0.91±0.12†‡ | 1.306±0.128†‡ | 0.74 | 14.13 |
| 252 | R+O | 1.29±0.33†‡ | 1.658±0.267†‡ | 1.31 | 12.00 | 768 | R+O | 1.02±0.20†‡ | 1.401±0.187†‡ | 0.92 | 13.81 |
| × | Y+O | 1.47±0.33†‡ | 1.841±0.243†‡ | 2.29 | 16.43 | × | Y+O | 1.43±0.17†‡ | 1.792±0.146†‡ | 1.88 | 17.57 |
| 6 | Y+IO | 0.87±0.20†‡ | 1.268±0.193†‡ | 0.65 | 18.40 | 8 | Y+IO | 0.88±0.11†‡ | 1.280±0.111†‡ | 0.73 | 20.51 |
| | SVGP | 4.41±0.60†‡ | 2.886±0.096†‡ | 18.55 | 15.44 | | SVGP | 2.13±0.18†‡ | 2.199±0.074†‡ | 4.70 | 16.90 |
| | NNGP | 12.40±1.45†‡ | 35.18±0.534†‡ | - | 7347 | | NNGP | 4.97±0.29†‡ | 32.40±0.638†‡ | - | 7374 |
| | ANP | 7.59±3.20†‡ | 1.793±0.887†‡ | - | 40.82 | | ANP | 4.08±2.27†‡ | 2.475±0.559†‡ | - | 102.3 |
| ENB/c | RF | 1.93±0.15†‡ | - | - | 0.31 | airfoil | RF | 3.68±0.17†‡ | - | - | 0.36 |
| | RIO | **1.55±0.15** | **1.854±0.087** | 2.13 | 16.31 | | RIO | **2.90±0.17** | **2.487±0.063** | 7.37 | 19.80 |
| | R+I | **1.53±0.12** | **1.845±0.072** | 2.08 | 12.95 | | R+I | 2.95±0.17†‡ | 2.505±0.059†‡ | 7.88 | 17.19 |
| 768 | R+O | 1.85±0.14†‡ | 2.037±0.089†‡ | 2.93 | 10.62 | 1505 | R+O | 3.54±0.17†‡ | 2.693±0.059†‡ | 10.50 | 8.26 |
| × | Y+O | 1.92±0.15†‡ | 2.081±0.091†‡ | 3.20 | 19.82 | × | Y+O | 3.58±0.18†‡ | 2.714±0.085†‡ | 12.05 | 21.22 |
| 8 | Y+IO | 1.85±0.14†‡ | 2.039±0.087†‡ | 2.99 | 21.50 | 5 | Y+IO | 3.31±0.41†‡ | 2.623±0.131†‡ | 10.44 | 23.61 |
| | SVGP | 2.63±0.23†‡ | 2.403±0.079†‡ | 6.81 | 18.63 | | SVGP | 3.59±0.20†‡ | 2.698±0.054†‡ | 12.66 | 20.52 |
| | NNGP | 4.91±0.32†‡ | 30.14±0.886†‡ | - | 7704 | | NNGP | 6.54±0.23†‡ | 33.60±0.420†‡ | - | 3355 |
| | ANP | 4.81±2.15†‡ | 2.698±0.548†‡ | - | 64.11 | | ANP | 21.17±30.72†‡ | 5.399±6.316†‡ | - | 231.7 |
| CCS | RF | 7.17±0.43†‡ | - | - | 0.49 | wine/r | RF | **0.625±0.028†‡** | - | - | 0.75 |
| | RIO | **5.84±0.34** | **3.180±0.068** | 24.84 | 17.72 | | RIO | 0.642±0.028 | 1.019±0.070 | 0.27 | 21.13 |
| 1030 | R+I | 5.89±0.35†‡ | 3.190±0.069†‡ | 25.34 | 14.62 | 1599 | R+I | **0.625±0.028†‡** | **0.966±0.059†‡** | 0.31 | 7.57 |
| × | R+O | 6.90±0.41†‡ | 3.380±0.079†‡ | 34.20 | 9.77 | × | R+O | 0.627±0.028†‡ | 0.974±0.061†‡ | 0.30 | 9.26 |
| 8 | Y+O | 7.03±0.78†‡ | 3.401±0.154†‡ | 35.75 | 18.72 | 11 | Y+O | 0.628±0.028†‡ | 0.975±0.062†‡ | 0.30 | 20.74 |
| | Y+IO | 5.88±0.33†‡ | 3.192±0.068†‡ | 25.78 | 22.01 | | Y+IO | 0.635±0.031†‡ | 0.999±0.074†‡ | 0.28 | 24.88 |
| | SVGP | 6.88±0.40†‡ | 3.337±0.048†‡ | 44.62 | 18.09 | | SVGP | 0.642±0.028 | 0.974±0.042†‡ | 0.40 | 20.41 |
| wine/w | RF | 0.714±0.016†‡ | - | - | 1.17 | CCPP | RF | 4.21±0.13†‡ | - | - | 1.47 |
| | RIO | 0.702±0.015 | 1.068±0.025 | 0.43 | 36.60 | | RIO | **4.02±0.14** | **2.801±0.029** | 15.18 | 44.81 |
| 4898 | R+I | **0.697±0.016†‡** | **1.059±0.025†‡** | 0.44 | 33.17 | 9568 | R+I | 4.03±0.14†‡ | 2.803±0.029†‡ | 15.22 | 39.09 |
| × | R+O | 0.715±0.016†‡ | 1.087±0.026†‡ | 0.46 | 13.70 | × | R+O | 4.17±0.13†‡ | 2.839±0.028†‡ | 16.19 | 17.67 |
| 11 | Y+O | 0.715±0.016†‡ | 1.087±0.026†‡ | 0.46 | 34.16 | 4 | Y+O | 13.26±63.05‡ | 3.274±3.329†‡ | 2095 | 44.63 |
| | Y+IO | 0.710±0.017†‡ | 1.080±0.028†‡ | 0.45 | 41.52 | | Y+IO | 4.63±0.95†‡ | 2.975±0.253†‡ | 35.54 | 45.63 |
| | SVGP | 0.719±0.018†‡ | 1.081±0.022†‡ | 0.50 | 36.62 | | SVGP | 4.36±0.13†‡ | 2.887±0.026†‡ | 18.94 | 48.83 |
| protein | RF | 5.05±0.03†‡ | - | - | 19.76 | SC | RF | 15.26±0.25†‡ | - | - | 56.98 |
| | RIO | **4.55±0.03** | **2.935±0.007** | 20.92 | 298.6 | | RIO | **14.18±0.75** | **4.072±0.051** | 196.1 | 170.6 |
| 45730 | R+I | 4.57±0.04†‡ | 2.939±0.008†‡ | 21.06 | 278.1 | 21263 | R+I | 14.38±0.76‡ | 4.089±0.047†‡ | 204.1 | 145.0 |
| × | R+O | 5.02±0.04†‡ | 3.033±0.007†‡ | 24.79 | 242.9 | × | R+O | 14.98±0.40†‡ | 4.126±0.034†‡ | 213.9 | 72.33 |
| 9 | Y+O | 5.04±0.04†‡ | 3.037±0.007†‡ | 25.02 | 206.4 | 81 | Y+O | 14.80±0.30†‡ | 4.121±0.059†‡ | 204.8 | 167.6 |
| | Y+IO | 4.56±0.03†‡ | 2.937±0.007†‡ | 21.00 | 300.2 | | Y+IO | **14.39±0.29** | **4.079±0.020** | 196.9 | 234.4 |
| | SVGP | 4.68±0.04†‡ | 2.963±0.008†‡ | 22.54 | 281.1 | | SVGP | 14.66±0.25†‡ | 4.135±0.013†‡ | 239.2 | 221.3 |
| CT | RF | 9.77±0.16†‡ | - | - | 302.0 | MSD | RF | 9.93±0.01†‡ | - | - | 2112 |
| | RIO | **9.32±0.49** | **3.651±0.053** | 86.67 | 1024 | | RIO | 9.90±0.01 | 3.711±0.001 | 97.59 | 2133 |
| 53500 | R+I | 9.77±0.16†‡ | 3.698±0.017†‡ | 94.04 | 47.04 | 515345 | R+I | 9.93±0.01†‡ | 3.714±0.001†‡ | 98.11 | 136.8 |
| × | R+O | 9.35±0.46 | **3.654±0.049** | 86.94 | 228.1 | × | R+O | 9.92±0.01†‡ | 3.714±0.001†‡ | 97.78 | 1747 |
| 384 | Y+O | 9.46±0.36†‡ | 3.668±0.042†‡ | 88.00 | 448.7 | 90 | Y+O | 9.94±0.01†‡ | 3.715±0.001†‡ | 98.29 | 2198 |
| | Y+IO | 9.58±0.66†‡ | 3.676±0.094†‡ | 91.09 | 1439 | | Y+IO | 9.94±0.01†‡ | 3.715±0.001†‡ | 96.73 | 5638 |
| | SVGP | 52.07±0.17†‡ | 5.372±0.003†‡ | 2712 | 61.70 | | SVGP | **9.57±0.00†‡** | **3.677±0.000†‡** | 92.21 | 2276 |

The symbols † and ‡ indicate that the difference between the marked entry and RIO is statistically significant at the 5% significance level using paired *t*-test and Wilcoxon test, respectively. The best entries that are significantly better than all the others under at least one statistical test are marked in boldface (ties are allowed).

