# OpenReview forum: "Quantifying Point-Prediction Uncertainty in Neural Networks via Residual Estimation with an I/O Kernel"
_ICLR.cc/2020/Conference — Accept (Poster)_

### Official Review · AnonReviewer2 · 2019-10-18
**Official Blind Review #2**

**Rating:** 6

**Review:**

This paper proposes a new framework (RIO) to estimate uncertainty in pretrained neural networks. For this purpose, RIO employs Gaussian Processes whose kernels are calculated by kernel functions of input and output samples and the corresponding target values.

- The proposed approach is interesting and the initial results are promising. However, there are various major and minor problems with the paper:

- The proposed method can be applied to any machine learning algorithm. It is not clear why you focus on employment of the proposed method for vanilla NNs.

- Have you applied RIO for other learning algorithms as well?

- Could you please explain more precisely, how you utilize which particular properties of NNs in RIO, and/or how RIO helps quantification and improvement of uncertainty of NNs particularly?

- Following equation (7), you claim that “In other words, RIO not only adds uncertainty estimation to a standard NN—it also makes its predictions more accurate, without any modification to its architecture or training”. Could you please verify and justify how RIO makes predictions of NNs more accurate? In this statement, I guess that you consider the results given in Theorem 2.6. However, you should not that the error functions given in Theorem 2.6 are calculated in a cascaded manner, i.e., by applying a GP at the output of a NN.

- The main proposal of the paper is that RIO makes it possible to estimate uncertainty in any pretrained standard NN. In order to verify that proposal, you should improve the experiments, esp. using larger datasets with larger neural networks, including deep neural networks.

After Rebuttal:

I read the comments of the other reviewers and response of the authors. Most of my questions were addressed in the rebuttal, and the paper was improved. However, there is still room to improve the paper with additional analysis using state-of-the-art algorithms on benchmark datasets, and to improve presentation of the work. Therefore, I improve my rating to Weak Accept.

**Experience Assessment:**

I have published in this field for several years.

**Review Assessment: Checking Correctness Of Derivations And Theory:**

I carefully checked the derivations and theory.

**Review Assessment: Checking Correctness Of Experiments:**

I carefully checked the experiments.

**Review Assessment: Thoroughness In Paper Reading:**

I read the paper at least twice and used my best judgement in assessing the paper.

---

> ### Author Response · Authors · 2019-11-15
> **Response to Reviewer #2 (1 out of 2)**
>
> Thank you for your constructive comments. Please see our responses to each of your concerns below:
>
> Q1: “The proposed method can be applied to any machine learning algorithm. It is not clear why you focus on employment of the proposed method for vanilla NNs.”
>
> A1: RIO can indeed be applied to other machine learning algorithms, but we believe vanilla NNs are a good choice for this paper for two reasons: (1) As the first paper on RIO, it makes sense to focus it on the analysis and demonstration of RIO’s abilities without the complexity of multiple platforms, and (2) vanilla NNs are very common model used by practitioners, making the results relevant to many people. We have discussed the motivation and reasons for which we choose standard NN as the focus in the main paper. However, since it is insightful to also test the generality of RIO, we have added a whole set of experiments that instead use random forest models, as described in A2 below.
>
> In more details:
> 1. Since this is the very first paper that introduces RIO, focusing on one widely used model allows us to do a thorough and deep investigation into the new approach, both theoretically and empirically. These detailed analysis and results should be very informative for practitioners who are using vanilla NNs. Including different approaches may lose this focus and depth.
> 2. Vanilla NN is arguably the most commonly used model among practitioners for making point predictions, but it also creates a lot of inconvenience and risks due to the lack of uncertainty information. Our target is to develop a tool that is practical and useful for the practitioner community, so choosing vanilla NN to demonstrate the effectiveness of RIO would be most appropriate as the first milestone.
>
> Q2: “Have you applied RIO for other learning algorithms as well?”
>
> A2: We have added the experimental results on Random Forests for all RIO variants and all datasets. Please see Table 7 in Appendix D.6 for full details of the experiments and results. To summarize, RIO performs the best or equals the best method (based on statistical tests) in 9 out of 12 datasets in terms of both RMSE and NLPD. In addition, RIO significantly improves the performance of original RF in 11 out of 12 datasets. These empirical results verifies the robustness and broad applicability of RIO. Full details of the results are included in the appendix, and we also referred to this as a concrete example when we discuss the extensibility of RIO in future work.
>
> Q3: “Could you please explain more precisely, how you utilize which particular properties of NNs in RIO, and/or how RIO helps quantification and improvement of uncertainty of NNs particularly?”
>
> A3: We use the expressivity of NNs, which means that they can learn complex structure that a GP would treat as noise. This point has been clarified in the revised version of Section 2.2. Similarly, RIO is particularly well-suited for NNs, because their expressivity makes it difficult to quantify their uncertainty with simpler analytical methods. However, RIO can be easily extended to other kinds of regression models as well, e.g., the new experiments in Appendix D.6 show that they work with random forests.

---

> > ### Author Response · Authors · 2019-11-15
> > **Response to Reviewer #2 (2 out of 2)**
> >
> > Q4: “Following equation (7), you claim that “In other words, RIO not only adds uncertainty estimation to a standard NN—it also makes its predictions more accurate, without any modification to its architecture or training”. Could you please verify and justify how RIO makes predictions of NNs more accurate? In this statement, I guess that you consider the results given in Theorem 2.6. However, you should not that the error functions given in Theorem 2.6 are calculated in a cascaded manner, i.e., by applying a GP at the output of a NN.”
> >
> > A4: We recognize that our statement “makes its predictions more accurate” leads to a confusion here. RIO is designed as a supporting tool that can be applied on top of a pre-trained NN, so what we mean here is that RIO can calibrate/correct the output of that pre-trained NN --- it does not change the performance of the pre-trained network itself. The error functions given in Theorem 2.6 correctly reflects the standard usage of RIO, i.e., a NN is pre-trained, then RIO is applied to it afterwards. To avoid confusion, we have modified the statement to “it also provides a way to calibrate NN predictions”.
> >
> > Q5: “The main proposal of the paper is that RIO makes it possible to estimate uncertainty in any pretrained standard NN. In order to verify that proposal, you should improve the experiments, esp. using larger datasets with larger neural networks, including deep neural networks.”
> >
> > A5: Thanks for this constructive comment. We have added new experiments that show RIO's off-the-shelf applicability to modern deep convolutional architectures on large datasets. It was applied to a recent pre-trained NN for age estimation based on DenseNet [1][2], which has 121 layers. The dataset is IMDB, which is the largest open source dataset of face images with age labels for training [3]. The pretrained NN and all data preprocessing were taken exactly from the official code release. RIO substantially improves upon the prediction errors of the pre-trained NN, outperforms SVGP in terms of both prediction error and uncertainty estimation, and yields realistic confidence intervals (See Table 3 in the main paper for more details). We have added these results into the main paper.
> >
> > [1] Tsun-Yi Yang, Yi-Hsuan Huang, Yen-Yu Lin, Pi-Cheng Hsiu, and Yung-Yu Chuang. “SSR-net: A compact soft stagewise regression network for age estimation”. In Proc. of IJCAI, pp. 1078–1084, 2018.
> > [2] Gao Huang, Zhuang Liu, Laurens Van Der Maaten, and Kilian Q Weinberger. “Densely connected convolutional networks”. In Proc. of CVPR, pp. 4700–4708, 2017.
> > [3] https://data.vision.ee.ethz.ch/cvl/rrothe/imdb-wiki/

---

### Official Review · AnonReviewer1 · 2019-10-28
**Official Blind Review #1**

**Rating:** 8

**Review:**

This paper solves an interesting scientific and applied problem: can we construct an algorithm to predict uncertainties without re-training/modifying existing neural network training algos? The authors propose a novel technique (called RIO) which leverages existing neural network but use both the input as well as the output of the neural net as an input to a GP which regresses on the residual error of the neural network. The authors describe the theoretical foundations as well as show empirical results on multiple datasets.

My thoughts on the paper:
- The paper is well written and from section 2.1 it is clear how one could re-produce their method.
- The theoretical section 2.2 feels a bit rushed, I think it would be worth sharing the high level intuition behind some of the theory first before going into the details.
- Section 2.4 could be more explicit about what "large scale" means. I.o.w. from a practical point of view, the method is only limited by approximate inference for Gaussian processes. Anno 2019 this is ...
- The empirical section is particularly strong and contains a wide variety of experiments with detailed analysis.
As a result, I think this is a good piece of scientific work that could be interesting to the wider community.

Although I did not re-run the results, the authors do share full source code for their results.

**Experience Assessment:**

I have read many papers in this area.

**Review Assessment: Checking Correctness Of Derivations And Theory:**

I assessed the sensibility of the derivations and theory.

**Review Assessment: Checking Correctness Of Experiments:**

I assessed the sensibility of the experiments.

**Review Assessment: Thoroughness In Paper Reading:**

I read the paper at least twice and used my best judgement in assessing the paper.

---

> ### Author Response · Authors · 2019-11-15
> **Response to Review #1**
>
> Many thanks for your recognition of our work! We really appreciate your encouraging words about our contributions. Please see below for our responses to your concerns:
>
> Q1: “The theoretical section 2.2 feels a bit rushed, I think it would be worth sharing the high level intuition behind some of the theory first before going into the details.”
>
> A1: Thanks for this constructive suggestion. We have rewritten Section 2.2 to make the high level intuition and motivation more clear. A big picture summary was added to the beginning of section, followed by an intuitive discussion of the approach. Prose has also been added to improve the flow of the section and clarify the predictions and conclusions drawn from the theoretical model.
>
> Q2: “Section 2.4 could be more explicit about what "large scale" means. I.o.w. from a practical point of view, the method is only limited by approximate inference for Gaussian processes. Anno 2019 this is ...”
>
> A2: Thanks for bringing up this point. Yes, the scalability of RIO is only limited by the approximate GP method. In order to quantitatively define what is a “large scale” dataset, we analyzed existing public regression datasets (as of November 2019). Based on the distribution of their sizes, a regression dataset can be considered “large scale” (~top 10% in size) if the product of its number of data points and number of features is larger than 1 million. We have added a clarification in Section 2.4 to make the definition of “large scale” more explicit. Among the datasets tested in this paper, 3 of them (“SC”, “CT”, “MSD”) fulfill this criterion, and they are ~1.7 million, ~20 million and ~46 million, respectively. RIO shows strong performance in all three “large scale” datasets, so the scalability of RIO is demonstrated.

---

### Official Review · AnonReviewer4 · 2019-10-30
**Official Blind Review #4**

**Rating:** 6

**Review:**

# Summary

The authors propose a method for post-hoc correction and predictive variance estimation for neural network models. The method fits a GP to the model residuals, and learns a composite kernel that combines two kernels defined on the input space and the model’s output space (called RIO, R for residual, and IO for the input-output kernel). The authors suggest that residual means and variances at test points can then be calculated explicitly using predictive distributions from the GP. The authors run a large panel of experiments across a number of datasets, and compare to a number of methods that draw connections between neural networks and GP’s. In addition, the full method is compared to a number of methods that utilize only some components of the full RIO method. In these experiments, the RIO method generally shows strong performance in both RMSE and NLPD compared to these baselines.

# Feedback

Overall, this is a neat method. It has the flavor of a number of other composite ML methods that have worked well in the past---e.g., boosting and platt scaling---but is different enough to stand on its own. The experimental results are quite promising.

However, I am torn about the paper, because the theoretical discussion of the method is quite convoluted and seems either irrelevant or incorrect. I wish that the authors had spent more time with small demonstrations of what the procedure does in some simple settings. This would give practitioners considering the method far more intuition about when they would expect it to work and fail than the current theoretical discussion.

## Uncertainty Discussion is Lacking

The motivation and discussion sell this method as an uncertainty quantification method, but almost all of the theoretical development revolves around prediction correction. The methods properties as an uncertainty quantification tool are underdeveloped.

The only theoretical point made about uncertainty estimation is Theorem 2.7, which states that the scalar variance of the GP “nugget” is positively correlated with the variance of the NN’s residuals. Providing a scalar summary of noise is not particularly compelling for a method advertised as a point-prediction uncertainty quantification method. In addition, it is not clear what probability distribution the “correlation” is defined over. The argument made in the proof seems quite obvious: if a GP is used to model a noisier process (i.e., residuals with a larger variance), it will in some cases classify that variability as independent noise.

If the authors wanted to focus on the properties of their method as an uncertainty quantification tool, they could discuss the assumptions underlying the GP error estimates, and when they would be likely to diverge from practical properties like predictive interval coverage. For example, because the base NN predictor is treated as fixed, it seems that this method ignore uncertainty that stems from the NN fit due to random initialization. Likewise, it seems that this method would not quantify uncertainty from resampling the data and obtaining a new NN predictor. )The coverage experiments in the appendix seem to confirm this -- generally, the predictive intervals generally under-cover the predicted values.) It’s fine if the method doesn’t quantify these types of uncertainty, but discussion of these types of issues would be far more welcome than the current convoluted theory in Section 2. This discussion might not yield theorems, but it would give practitioners useful guidelines for deciding whether the particular scheme would likely work for this application.

## Problems with the Error Correction Argument

The theory section, especially 2.2, was very difficult to parse. First, as a matter of style, a sequence of Lemma and Theorem statements are given without defining most of the notation used therein, and with almost no prose providing context or intuition. In the buildup to the theorems, it is also unclear which assertions about the decompositions of y_i are assumptions about the true data generating process, and which assertions are specifications of a particular GP model.

The substance also has some issues. I think the intention in this section is to get to a rather simple variance decomposition of the labels y. The question is how much variation in y or the residual is represented in the posterior predictive mean of a particular GP. It seems reasonable that in some cases, the structure in the residual may be more amenable to modeling with a stationary GP than the structure in the raw labels y. It is not clear that all of the theoretical complexity here is necessary to make this point.

Instead, the authors make a convoluted argument that attempts to establish that the errors from the NN + GP approach will be smaller under very general circumstances. The argument is phrased somewhat ambiguously (it is not clear exactly what is being assumed, and what is corresponds to the specification of a working model), but depending on how one reads this section, the argument makes statements that are either too broad to be correct, or too narrow to be relevant.

The argument decomposes for the raw labels and the residuals into pieces that a GP can “capture” or “represent”, and parts that it cannot. The two equations are:

y_i = f(x_i) + g(x_i) + \xi_i
R_i = (y_i - h_NN(x_i)) = r_f(x_i) + r_g(x_i) + \xi_i

f(.) and r_f(.) represent the portions of the label and residual processes, respectively, that the GP "captures". It is assumed that the GP will model this portion correctly, and leave the “epsilon-indistinguishable” portion g(.) or r_g(.) untouched. The argument then assumes that f(.) and r_f(.) will have proportional kernels, and so it is possible to show that the predictions of residuals based on r_f(.) will have smaller predictive variance than predictions based on f(.) as long as the variation represented by r_g(.) is smaller than the variation represented by g(.).

On its face, this argument raises some red flags. Because h_NN(.) is allowed to be an arbitrary function, the argument here should be symmetric. Why can’t we also get a guaranteed variance reduction by adding h_NN(.) to y rather than subtracting it? Perhaps some of this is captured in the parameter \delta, which quantifies the reduction in variation represented in r_g(.) vs g(.), but the argument that the kernel of r_f(.) can be no larger than the kernel f(.) in terms of trace (that is, the proportionality constant \alpha is not greater than 1) does not make sense. If h_NN(x_i) is simply -f(x_i), then these arguments would not go through. At the very least, conditions need to be articulated about the properties of h_NN(.).

Some of the strangeness comes from the fact that this is a poor model of most prediction problems, where the main issue with fitting a GP is not “indistinguishability”, but misspecification. Consider a process y_i that is non-stationary; say g(.) has a linear trend in some component of x. A GP with a stationary covariance kernel fit to this process (such as RBF) will attempt to explain the variation due to the linear trend with a variance kernel that encodes long-range dependence. On the other hand, if this trend were removed by a base model like an NN, the residuals would have a very different structure (perhaps they would be stationary), and in this case, the GP would fit the data with a very different covariance kernel.

Unfortunately, it does not seem like the formalism here can express a notion of misspecification at all. In the theory, it is assumed that the GP will only model the portion of the labels y_i for which it is property specified (in this case, f(.)). This generally does not occur in practice, as in the example above. It might be possible for this to apply in some circumstances, but the authors give no conditions (e.g., that the process y_i be stationary). Based on this assumption, the authors assert that the fitted GP to f(.) and r_f(.) will have the same covariance kernel parameters up to some proportionality constant \alpha. Much of their theoretical argument depends on this proportionality. But this proportionality cannot apply in general, and again, no conditions are given for when we might expect this to hold.

It would be far more compelling if the authors proposed the very standard approach to modeling data via covariance kernels, where one first models non-stationary portions of the data with a base model, then models the correlation in the residuals with something like a GP. This is the bread-and-butter approach in, say, timeseries analysis (see, e.g., the Shumway and Stoffer textbook https://www.stat.pitt.edu/stoffer/tsa4/tsa4.htm), and the approach in this paper could be framed similarly.

## Demos I Wish I Had Seen

I wish the authors had presented some demonstrations of what the GP does to the fitted values of an NN. Giving a demonstration of how the output kernel modifies predicted values, for example, would give some nice intuition the value added by this portion. I suspect that this step essentially performs something like Platt scaling, but for continuous outcomes, by shrinking predictions together so that they better match the overall distribution of observed labels. Perhaps the mechanism is different. At any rate, it would be useful to understand where the information gain is coming from, and this would be far better expressed concretely in terms of a toy data example than the theoretical arguments that are given.

## Coverage Experiments

I wish the coverage experiments evaluating predictive intervals were included in the main text. As far as uncertainty quantification evaluations go, coverage is one of the few assessments that does not rely on the model itself (unlike NPLD, which uses the model’s own log-likelihood), and can be phrased as a concrete performance guarantee.

Here, the goal for predictive intervals is to cover the true prediction value _at least as often_ as the nominal rate (95% intervals should cover the truth _at least_ 95% of the time), not merely that coverage be “close” to the nominal rate. This asymmetric evaluation gives you a concrete guarantee that the uncertainty estimate is conservative. The coverage experiments show that this method quite systematically under-covers compared to the end-to-end SVGP method, which generally satisfies this coverage property. I think this is important information to include about the model, and generally I think this behavior results from the fact that uncertainty is not propagated from the NN fit. This should be presented clearly in the main text.

**Experience Assessment:**

I have published one or two papers in this area.

**Review Assessment: Checking Correctness Of Derivations And Theory:**

I carefully checked the derivations and theory.

**Review Assessment: Checking Correctness Of Experiments:**

I assessed the sensibility of the experiments.

**Review Assessment: Thoroughness In Paper Reading:**

I read the paper thoroughly.

---

> ### Author Response · Authors · 2019-11-15
> **Response to Review #4 (1 out of 4)**
>
> Thanks for this thorough and detailed review of our work, particularly with regards to the theory. The overarching concern was that the motivation, details, and implications of the theory were unclear, and it would be more compelling if the detailed behaviors of RIO can be demonstrated using concrete examples. To address these concerns, we have added more concrete empirical demonstrations of the detailed behaviors of RIO, regarding both output correction and confidence interval coverage. We have also rewritten Section 2.2 in the newly uploaded version of the paper, aiming to clarify the assumptions, the motivation of each step, and the conclusions drawn. We believe this update addresses the overarching concern, and addresses many of the specific comments in the process. We will respond to your concerns regarding “## Demos I Wish I Had Seen” and “## Coverage Experiments” first, then reply to all your concerns related to theory.
>
> Q: comments within “## Demos I Wish I Had Seen” section
>
> A: Thanks for these very insightful comments. We have added two empirical studies to analyze what RIO actually does during the correction of NN outputs. The first one demonstrates that RIO doesn't always shrink predictions together, but instead performs different calibrations on each point in order to move the predictions closer to the ground truth. The second one demonstrates that most of these calibrations are indeed improvements. We have included detailed description of these studies in the appendix, and mention the conclusions in the main text.
>
> More details on the first study: In the first empirical analysis, we randomly pick a run for each tested dataset, and plot the distributions of ground truth labels(outcomes), original NN predictions and predictions corrected after RIO. The results are summarized in Figure 9 of Appendix D.3. Based on the results, it is clear that RIO is not simply shrinking predictions together. Instead, RIO tends to calibrate each NN prediction accordingly. The distribution of outputs after RIO calibration may be a shift, or shrinkage, or expansion, or even more complex modifications of the original NN predictions, depending on how different are NN predictions from ground truths. As a result, the distribution of RIO calibrated outputs are always closer to the distribution of ground truths. One interesting behavior can be observed for “protein” dataset (row 3, rightmost plot): after applying RIO, the range of whiskers shrunk and the outliers disappeared, but the box (indicating 25 to 75 percentile of the data) expanded. This behavior shows that RIO is actually trying to calibrate each point differently. To provide more details, the point-wise comparisons between NN outputs and RIO-corrected outputs for the same experimental runs as in Figure 9 are shown in Figure 10 of Appendix D.3. From Figure 10, RIO shows different calibration behaviors accordingly. If we compare the plots in Figure 10 to the corresponding ones in Figure 9 (they are for the same run on the same dataset), it is clear that all these different calibration behaviors actually make sense, and they are generally leading to more accurate predictions of ground truths.
>
> More details on the second study: In the second empirical study, we define a new performance metric called “improvement ratio” (IR), which is the ratio between number of successful corrections (successfully reducing the error) and total number of data points. For each run on each dataset, we calculate this IR value, and the distribution of IR values over 100 independent runs (random dataset split except for MSD, random NN initialization and training) on each dataset is plotted in Figure 11 of Appendix D.3. According to the results, the IR values for RIO are above 0.5 in most cases. For 7 datasets, IR values are above 0.5 in all 100 independent runs. For some runs in “yacht”, “ENB/h”, “CT”, and “MSD”, the IR values are above 0.8 or even above 0.9. All these observations show that RIO is making meaningful corrections instead of random perturbations. Results in Figure 11 also provides useful information for practitioners: Although not all RIO calibrations improve the result, most of them do.
>
> Q: comments within “## Coverage Experiments” section
>
> A: Confidence interval (CI) coverage indeed is a concrete and straightforward performance metric, which is why we included the 95%/90%/68% CI coverages for all algorithms in all datasets in the Appendix. However, after a deeper investigation, including an additional experiment (Figure 6, 7 and 8 in Appendix D.2), we found this performance metric to be noisy and potentially misleading. Drawing conclusions from it requires lengthy qualifications; given the page limits of the main text, we believe such discussions are better presented in the appendix. In contrast, NLPD loss is more reliable, which is why it is the primary measure in this paper. This choice is now explained in the main text, and a justification given in the appendix.

---

> > ### Author Response · Authors · 2019-11-15
> > **Response to Review #4 (2 out of 4)**
> >
> > More details regarding the unreliability of CI coverage metric:
> > 1. The plots for dataset “yacht”, “ENB/c”, “protein”, “SC”, and “CT” show that the RIO variants are more optimistic than SVGP in 95% and 90% CI coverage, but becomes more conservative than SVGP in 68% CI coverage. If one approach is really more conservative than the other one, then it should have wider CI coverage at different confidence levels consistently. This phenomenon alerts us that the empirical CI coverage may mislead the comparison.
> > 2. In addition, for “CT” dataset, SVGP has an extremely high RMSE of ~52 while RIO variants only have RMSEs of ~1. However, SVGP still shows acceptable 90% and 68% CI coverage, and even has over-confident coverage for 68% CI. After investigation, what actually happened is that SVGP was not able to extract any useful information from the high-dimensional input space, so it treated all the outcomes as simply noise. As a result, SVGP shows a very large RMSE compared to other algorithms, and the mean of its predicted outcome distribution is always around 0. Since SVGP treats everything as noise, the estimated noise variance is very high, and the estimated 95% CI based on this noise variance is overly high and covers all the test outcomes in most cases. When the estimated 90% CI is evaluated, the big error in mean estimation and big error in noise variance estimation cancel most part of each other by chance, i.e., the estimated 90% CI is mistakenly shifted by erroneous mean then the overly wide noise variance fortunately covers slightly more than 90% test outcomes. Similar thing happens to the estimated 68% CI, but now the error in noise variance cannot fully cover the error in mean, so the coverage percentages are below 68%, indicating over-confidence. This investigation shows how noisy the empirical CI coverage may be.
> > 3. To have a clearer picture about what RIO and SVGP do regarding CI coverage, we added an experiment that shows the distribution of CI coverages for all confidence levels (from 1% to 99%), and plot the results for RIO and SVGP in the same figure (Figure 6, 7, and 8 in Appendix D.2). From Figure 6, 7 and 8, it can be seen that SVGP also shows more “optimistic” CI coverages in many cases (“airfoil”, “CCPP”, “protein”, “CT”, and confidence levels below 70% in “Yacht”). One interesting observation is that SVGP tends to be more “conservative” for high confidence levels (>90%), even in cases where they are “optimistic” for low confidence levels. After investigation, this is because SVGP normally has an overly high noise variance estimation (also comes with a higher prediction RMSE in most cases), so it has a higher chance to cover more points when the increase in CI width (influenced by noise variance) surpasses the erroneous shift of mean (depending on prediction errors). This can explain why the original 95% and 90% CI coverage plots may suggest that SVGP is more “conservative”. In summary, we can not easily draw solid conclusions from these CI coverage metrics regarding the ability of approaches in predictive uncertainty estimation. A method that simply learns the distribution of the labels would perform well in CI coverage metric, but it cannot make any meaningful point-wise prediction.
> >
> > Details on using NLPD: We agree that being conservative is better than over-confident in real-world applications, but we also think a more accurate uncertainty estimation makes sense —- it will provide more useful information for decision making. A good example would be SVGP in “CT” dataset, its 95% CI covers 100% of the testing outcomes, which is very conservative. However, no useful information can be extracted from this extremely wide CI: it simply covers everything. We think a good balance between being accurate and being conservative is important. This is why we choose the NLPD metric to measure the performance in uncertainty estimation. According to [1], “The NLPD loss favours conservative models” (see Fig. 7 in [1]), but it also penalizes both over- and under-confident predictions. It is widely used in literature as a reasonable measure of Bayesian models [2]. During our testing, NLPD indeed returns more reliable evaluations of the uncertainty estimation (one good example would be that SVGP has a very high NLPD loss in “CT” dataset). Based on all the above considerations, we prefer NLPD as the performance metrics for uncertainty estimation.
> >
> > [1] Joaquin Quin ̃onero-Candela, Carl Edward Rasmussen, Fabian Sinz, Olivier Bousquet, and Bernhard Scholkopf. “Evaluating predictive uncertainty challenge”. In Machine Learning Challenges. Evaluating Predictive Uncertainty, Visual Object Classification, and Recognising Tectual Entailment, pp. 1–27, Berlin, Heidelberg, 2006. Springer Berlin Heidelberg.
> > [2] Andrew Gelman, Jessica Hwang, and Aki Vehtari. “Understanding predictive information criteria for bayesian models”. Statistics and Computing , 24(6):997–1016, Nov 2014.

---

> > > ### Author Response · Authors · 2019-11-15
> > > **Response to Review #4 (3 out of 4)**
> > >
> > > As described in the Main Response at the top of the first part of the response, this part of the response contains responses to specific comments regarding the theory in Section 2.2. Here, each original review comment has been briefly summarized to make the response more concise.
> > >
> > > Q. On concern that the discussion focuses on too much on prediction and too little on uncertainty:
> > >
> > > A: The connection of prediction correction to uncertainty quantification is now clarified in Section 2.2. First, it is important to focus on prediction correction because it implies that uncertainty can be quantified without sacrificing prediction performance, which is a drawback of many existing methods. Second, when prediction error is improved by reducing spurious noise, as in Section 2.2, the uncertainty quantification immediately becomes more precise. More discussion of uncertainty estimation has been added in the revised Section 2.2.
> > >
> > > Q. On concern that Theorem 2.7 does not contribute much:
> > >
> > > A: While the argument made in the proof is straightforward, it makes a valuable observation about the global structure of point-prediction uncertainty. The practical property that GP variance is positively correlated with NN residuals motivates quantifying point-prediction uncertainty in the first place: unstable NNs are less certain than stable NNs, and this fact must be captured by the method.
> > >
> > > Q. On request to discuss the assumptions underlying the GP error estimates and the uncertainty resulting from random initialization and resampling.
> > >
> > > A: Investigating additional sources of error and uncertainty is an interesting direction of future work. Indeed, the main extent of the theory in the paper is to show that using a GP to augment a pretrained neural network can be a reasonable thing to do. This theory is sufficient to motivate the method and make predictions that are borne out in experiments. Though perhaps straightforward, the theory does confirm some useful guidelines for whether the approach would work in a practitioner’s application, e.g., that they have access to an NN that outperforms GP in their application. More detailed guidelines for when to use RIO are gleaned from the extensive experimental results in Section 3 and in the Appendix.
> > >
> > > Q. On difficulty to follow section 2.2:
> > >
> > > A: We have rewritten Section 2.2 to make the intuition, assumptions, and notation clear.
> > >
> > > Q. On concern that the theoretical complexity is unnecessary to make the main point that residuals are more amenable to modeling than the raw labels.
> > >
> > > A: Indeed, this is the main point, and not much theoretical complexity is required to establish it at a high level. Section 2.2. has been rewritten to reflect this perspective. The goal of the more involved theory is to provide a concrete instantiation of the point, i.e., to identify a class of scenarios in which the high-level motivation produces the desired behavior.

---

> > > > ### Author Response · Authors · 2019-11-15
> > > > **Response to Review #4 (4 out of 4)**
> > > >
> > > > Q. On concern that the arguments for lower NN+GP errors are too narrow to be useful:
> > > >
> > > > A: Section 2.2 has been rewritten to clarify this concern. The goal is not to establish that the errors will be smaller under very general circumstances, but to identify a class of scenarios in which the desired behavior is achieved. This class of scenarios also has value in the fact that it leads to conclusions that can be corroborated by real world experiments. In fact, the theoretical conclusions in Section 2.2 are indeed in accordance with the experimental results. In particular, experiments confirm that RIO always outperforms the pretrained NN (Table 1), outperforms GP alone when the pretrained NN is well-trained (Table 1), and reductions in apparent noise correspond to lower prediction error (Table 2).
> > > >
> > > > Q. On request to clarify the conditions on properties of h_NN, and in particular, the reason why \alpha is not greater than one.
> > > >
> > > > A: A central assumption is that any spurious complexity introduced by the NN is collected in r_g, and this assumption is expressed in the constraint that \alpha is not greater than one. This is now made clear in the rewritten version of Section 2.2.
> > > >
> > > > Q. On concern that in most prediction problems the main issue for GP is not indistinguishability but misspecification.
> > > >
> > > > A: Although indistinguishability may not be the main issue in all prediction problems, we find that it is an important issue for many problems in practice, especially those for which NN outperforms GP, since, unlike many models, NNs are expressive enough to learn structure that GP would discount as noise. This is evidenced by the substantially higher noise estimates for SVGP than for RIO in Table 1, and the fact that in the CT dataset (Table 1) and IMDB results (Table 3), SVGP is unable to learn any useful structure, estimating nearly all of the variance as noise. We have clarified the connection between misspecification and indistinguishability at the beginning of the rewritten version of Section 2.2.
> > > >
> > > > Q. On concern that the formalism cannot express misspecification because it is based on assuming that the GP fitted to f(.) and r_f(.) have proportional covariance kernel parameters.
> > > >
> > > > A: Proportionality of the covariance kernels is indeed a simplifying assumption. It is useful because it allows us to focus on the interaction of signal variance and noise variance, leading to insights that are verified in experiments[TBD].
> > > >
> > > > Q. On the suggestion that the analysis could be more compelling if it were based on the standard approach of decomposing a covariance kernel into its stationary and non-stationary components.
> > > >
> > > > A: This is indeed a compelling idea in general. However, in practice, there is nothing inherent to RIO that assumes the input and output kernels of GP are stationary, and both the input kernel and the output kernel can contain both stationary and non-stationary components. The “misspecification” issue is a problem-specific kernel tuning problem, which is general to all GP-related approaches. Therefore, we simplify this aspect in order to highlight properties specific to RIO in particular.

---

> > ### Comment · AnonReviewer4 · 2019-11-15
> > **Response to all responses**
> >
> > Thanks for these thorough responses. They give a clearer picture of your method. As I stated before, I think the method is a very good idea, but that the theory discussion (still) does more to obfuscate the method than to motivate it.
> >
> > ## Demos
> >
> > These experiments are helpful.
> >
> > Would you be able to include plots of what _just_ the input kernel or _just_ the output kernel would do? I'm particularly interested in how the output kernel adds value here.
> >
> > ## Coverage experiments
> >
> > I appreciate this discussion, and agree that coverage is not the end-all. It is useful to have the context about what the GP method is doing.
> >
> > ## Prediction vs Uncertainty
> >
> > I take the point that prediction and uncertainty are clearly related. I still wish there were more discussion about what kind of uncertainty is being quantified, and what uncertainty is not quantified. For example, as you stated, for a completely overfit NN, your uncertainty estimate would be zero. Perhaps this is what you're getting at with Theorem 2.2, but I would find it more compelling if were described, e.g., in terms of uncertainty implied by mistakes made in the training set.
> >
> > ## Theory
> >
> > I appreciate that Section 2.2 has been rewritten, and it is indeed much clearer now.
> >
> > I still feel like there's too much going on in this theory section. The main point is that the decomposition of the residual function may be more favorable to model with a GP than the decomposition of the label function. I like your characterization of $f(.)$ and $g(.)$ in terms of _apparent_ signal and noise, and it would be straightforward to extend this $r_f(.)$ and $r_g(.)$, and simply state that the expected error depends on $\|g\|$ and $\|r_g\|$ in each case, so if $\|r_g\|$ is smaller than $\|g\|$, then you're going to do better.
> >
> > The additional assumptions made here to construct a case where this condition happens to hold are only confusing, and I would bet that these assumptions do not hold in the experiments. (I do appreciate that these are now articulated clearly as assumptions). For example, in the CT experiment, I assume you fit SVGP using an RBF kernel (which is stationary): did you check whether the kernel from fitting the GP alone is proportional to the kernel from fitting the GP after the NN? I predict that the length scale of the kernel from fitting the GP alone is much larger than the length scale of the kernel from fitting the GP after the NN. I imagine in most cases where the noise variance changes dramatically between SVGP and RIO that the input kernels look very different.
> >
> > Even though the qualitative behavior happens to match the predictions of the theory, to me, this theory does not capture why you see the behavior that you do. As I stated in the original review, given that most of these GP baselines are using stationary kernels, differences in misspecification between the raw labels vs the residuals seems like a much more plausible explanation for RIO's success.

---

> > > ### Author Response · Authors · 2019-11-15
> > > **Response to "Response to all Responses" of Reviewer #4**
> > >
> > > Thanks for the quick response! We were happy to be able to supply some last minute feedback below.
> > >
> > > 1.  On “## Demos” comment
> > >
> > > Good suggestion! We have generated the plots for “R+I” (predicting residuals with only input kernel) and “R+O” (predicting residuals with only output kernel) in the same way as we did for RIO in Figure 9, 10 and 11. In the updated version of the paper, please see Figures 12, 13 and 14 in Appendix D. 3 regarding “R+I”, and Figures 15, 16 and 17 in Appendix D.3 regarding “R+O”. From the results, the output kernel indeed helps a lot in problems where input kernel does not work well (“CT” and “MSD”), and it also shows more robust performance in terms of improvement ratio (IR) in most datasets. However, it is still generally worse than full RIO. The conclusion is that output kernel is very helpful, but combining input kernel with output kernel is the best.
> > >
> > > More details and clarification for the results:
> > >         1). “R+I” shows an extremely low IR in “CT” dataset (Figure 14), after investigation, this is because the input kernel itself is not able to learn anything from the complex high-dimensional input space, so it treats everything as noise. As a result, it keeps the NN output unchanged during correction in most cases. Applying output kernel instead solves the issue.
> > >         2). After comparing Figure 10, 13 and 16, it can be observed that the behaviors of RIO are either a mixture or selection between R+I and R+O. This means RIO with I/O kernel is able to choose the best kernel among these two or combines both if needed.
> > >
> > > 2. On “## Prediction vs Uncertainty” comment
> > >
> > > Good suggestion, thanks. RIO focuses on the pretrained NN’s internal uncertainty about its predictions, i.e., indeed the uncertainty implied by mistakes made on the training set, like you suggest. We will make this perspective clear in the final version of the paper, and discuss how it relates to other forms of uncertainty.
> > >
> > > 3. On “## Theory” comment
> > >
> > > Thanks for the suggestion to simply state the extension to $r_f(.)$ and $r_g(.)$ in the high-level portion of the theory. We agree that this makes the intuition even more clear, and will add it to the final version of the paper.
> > >
> > > We agreed that the current theory does not cover all the aspects that leads to RIO’s success. We think the improvement of RIO results from both “indistinguishability” and “misspecification”. We will discuss this point and make it clear in the final version of the paper. However, we find that the current theory also captures part of the motivations of the approach, and provides a novel perspective, as well as insights on the connection between NN and GP. We can move part of the theory into the appendix, and move more detailed empirical studies into the main paper.
> > >
> > > Thanks for all your help—we believe the paper has much improved as a result!

---

### Official Review · AnonReviewer3 · 2019-11-02
**Official Blind Review #3**

**Rating:** 6

**Review:**

The paper focuses on the model inference of neural networks (NN). The authors propose to use NN for the model, and fit the prediction residuals with a Gaussian process with input/output (IO) kernel. This kernel considers both input x and output y. The authors show that the NN+GP scheme has lower generalization error compared with solely using GP or NN to fit the model. Also, the IO kernel generalizes better than input kernel I, and output kernel O, in Gaussian process modeling. In experiments, the authors evaluate various methods in terms of several metrics to show that the proposed procedure gives better uncertainty estimation and more accurate point estimation.

In general it is a good paper, with good applications. The motivation is clear. The key idea of this paper is pretty common in statistical inference.
1.	In more practical settings we cannot assume that NN is always trained well. In this case, does the proposed method perform much worse than GP?
2.	Is this the only proposal for fitting the residuals for uncertainty estimation? Is there any other similar approach? I would like to see more discussions on other related methods and how the idea is different.
3.	Summarizing the whole procedure in an algorithm could make things clearer.


**Experience Assessment:**

I have read many papers in this area.

**Review Assessment: Checking Correctness Of Derivations And Theory:**

I assessed the sensibility of the derivations and theory.

**Review Assessment: Checking Correctness Of Experiments:**

I assessed the sensibility of the experiments.

**Review Assessment: Thoroughness In Paper Reading:**

I read the paper at least twice and used my best judgement in assessing the paper.

---

> ### Author Response · Authors · 2019-11-15
> **Response to Review #3**
>
> Thank you for your positive evaluation of our work. Please see our point-to-point responses to your comments:
> Q1: “In more practical settings we cannot assume that NN is always trained well. In this case, does the proposed method perform much worse than GP?”
>
> A1: No. The study did actually include several cases where the original NN performs poorly, and RIO still performed better than or comparably to GP. This result was not emphasized in the original paper; we have made it clear in the revised version.
>
> In more details: We paid special attention to evaluating the robustness of RIO during the design of the experiments. As stated in the experimental setups, for each dataset (except for MSD, in which the dataset split is strictly predefined by the provider), 100 independent runs are conducted. During each run, the dataset is randomly split, and the NN is randomly initialized. Moreover, we are using a standard training procedure without over-tuning that are commonly used by practitioners. All these steps enable us to cover different training situations and generate NNs with different qualities. The performance distributions of original NN and corresponding RIO/GP are plotted in Figure 3 in the original paper. From Figure 3, the trained NNs show diverse performance in terms of prediction RMSE (horizontal axis), and RIO is able to consistently improve the performance of NN into a level that is better or comparable to GP even though the original performance of NN is poor. For the datasets in which original NN performs much worse than GP, namely “airfoil”, “CCPP” and “MSD”, the performance after applying RIO becomes better than GP instead. More specifically, for “airfoil”, the original RMSEs of some NNs are above 5.5 (dots on the right side) while corresponding GP in the same runs only have RMSEs below 4.0, applying RIO to these NNs achieves RMSEs below 3.5. Similar pattern can be observed in “CCPP”. For “MSD”, although the original RMSEs of NN are above 17.0 in some cases, comparing to ~9.6 for GP, RIO is able to reduce these RMSEs to similar level (~9.8) as GP or even better (~9.4). These experimental results demonstrate that RIO is still robust even in the cases where NN are not well trained.
>
> Q2: “Is this the only proposal for fitting the residuals for uncertainty estimation? Is there any other similar approach? I would like to see more discussions on other related methods and how the idea is different.”
>
> A2: We have done a thorough literature review, and to the best of our knowledge, this is the only work that fits residuals for uncertainty estimation. The unique characteristic of RIO is that it is designed as a supporting tool to augment pre-trained NNs. In contrast, all other existing methods are designed as independent models that need to be trained from scratch. Considering the popularity of NNs among practitioners and the lack of uncertainty information in NN predictions, we do think a tool that can be directly applied on top of pre-trained NNs provides practical value. We have expanded the discussions in “Introduction” section and “Related Work” section to emphasize this point.
>
> Q3: “Summarizing the whole procedure in an algorithm could make things clearer.”
>
> A3: Thanks for your constructive comment. We have added an algorithm in the Appendix (Algorithm 1 in Section C) that describes the procedure of RIO. We have added a note in the main paper to refer interested readers to the algorithm.

---

### Author Response · Authors · 2019-11-15
**Summary of Main Revisions**

We want to thank all the reviewers for their effort in reading the paper and providing constructive comments. We have addressed all the main concerns, and have updated the paper accordingly. The main updates are as follows:

-New experiments:
       -Application of RIO to age estimation with pretrained DenseNet
       -Application of RIO with Random Forest regressors
-New empirical analysis:
       -Detailed analysis on the behavior of RIO in error correction
       -Detailed analysis on confidence interval coverage
-Clarification of theory:
       -Section 2.2 has been rewritten to clarify the high level intuition, the motivation of each step, and the conclusions drawn.

We have also responded to the specific comments of each reviewer in replies to their reviews.

---

### Decision · Program_Chairs · 2019-12-19

**Decision:**

Accept (Poster)

**Comment:**

This paper presents a method to model uncertainty in deep learning regressors by applying a post-hoc procedure.  Specifically, the authors model the residuals of neural networks using Gaussian processes, which provide a principled Bayesian estimate of uncertainty.  The reviewers were initially mixed and a fourth reviewer was brought in for an additional perspective.  The reviewers found that the paper was well written, well motivated and found the methodology sensible and experiments compelling.  AnonReviewer4 raised issues with the theoretical exposition of the paper (going so far as to suggest that moving the theory into the supplementary and using the reclaimed space for additional clarifications would make the paper stronger).  The reviewers found the author response compelling and as a result the reviewers have come to a consensus to accept.  Thus the recommendation is to accept the paper.

Please do take the reviewer feedback into account in preparing the camera ready version.  In particular, please do address the remaining concerns from AnonReviewer4 regarding the theoretical portion of the paper.  It seems that the methodological and empirical portions of the paper are strong enough to stand on their own (and therefore the recommendation for an accept).  Adding theory just for the sake of having theory seems to detract from the message (particularly if it is irrelevant or incorrect as initially pointed out by the reviewer).